# A liver immune rheostat regulates CD8 T cell immunity in chronic HBV infection

Chronic hepatitis B virus (HBV) infection affects 300 million patients worldwide[1,2], in whom virus-specific CD8 T cells by still ill-defined mechanisms lose their function and cannot eliminate HBV-infected hepatocytes[3–7]. Here we demonstrate that a liver immune rheostat renders virus-specific CD8 T cells refractory to activation and leads to their loss of effector functions. In preclinical models of persistent infection with hepatotropic viruses such as HBV, dysfunctional virus-specific CXCR6[+] CD8 T cells accumulated in the liver and, as a characteristic hallmark, showed enhanced transcriptional activity of cAMP-responsive element modulator (CREM) distinct from T cell exhaustion. In patients with chronic hepatitis B, circulating and intrahepatic HBV-specific CXCR6[+] CD8 T cells with enhanced *CREM* expression and transcriptional activity were detected at a frequency of 12–22% of HBV-specific CD8 T cells. Knocking out the inhibitory *CREM/ICER* isoform in T cells, however, failed to rescue T cell immunity. This indicates that CREM activity was a consequence, rather than the cause, of loss in T cell function, further supported by the observation of enhanced phosphorylation of protein kinase A (PKA) which is upstream of CREM. Indeed, we found that enhanced cAMP–PKA-signalling from increased T cell adenylyl cyclase activity augmented CREM activity and curbed T cell activation and effector function in persistent hepatic infection. Mechanistically, CD8 T cells recognizing their antigen on hepatocytes established close and extensive contact with liver sinusoidal endothelial cells, thereby enhancing adenylyl cyclase–cAMP–PKA signalling in T cells. In these hepatic CD8 T cells, which recognize their antigen on hepatocytes, phosphorylation of key signalling kinases of the T cell receptor signalling pathway was impaired, which rendered them refractory to activation. Thus, close contact with liver sinusoidal endothelial cells curbs the activation and effector function of HBV-specific CD8 T cells that target hepatocytes expressing viral antigens by means of the adenylyl cyclase–cAMP–PKA axis in an immune rheostat-like fashion.

CD8 T cells are key in the control of hepatitis B virus (HBV) infection of the liver and kill infected hepatocytes[3] but, during chronic infection, virus-specific CD8 T cells are dysfunctional and fail to eliminate infected hepatocytes. Spontaneous regain of immune control of infection in a few patients with chronic hepatitis B indicates that loss of virus-specific T cell function in these patients is reversible[8,9] and is not necessarily epigenetically programmed as observed for exhausted virus-specific T cells[10]. Attempts to strengthen virus-specific immunity by immune therapies, such as therapeutic vaccination, are considered promising approaches to restore virus-specific CD8 T cell function in patients with chronic hepatitis B[11–15]. It remains largely unclear, however, what causes the loss of virus-specific CD8 T cell function in the liver during persistent hepatocyte infection.

### CREM-expressing CXCR6[+] CD8 T cells in persistent HBV

It is difficult to study the mechanisms curbing antiviral T cell immunity during chronic hepatitis B because of the scarcity of virus-specific CD8 T cells[16–18]. Therefore, we established a model of persistent infection compared to acute-resolved infection with viruses that target and replicate specifically in hepatocytes. We generated two hepatotropic recombinant adenoviruses encoding ovalbumin, green fluorescence protein (GFP) and luciferase (GOL)[19]. These adenoviruses differed in their promoters driving viral gene expression and the outcome of infection, a cytomegalovirus promoter (Ad–CMV–GOL) leading to acute resolved infection with transient liver damage compared to a hepatocyte-specific transthyretin promoter (Ad–TTR–GOL) leading to persistent infection with continuous low-level liver damage (Fig. 1a and Extended Data Fig. 1a–f). Ad–TTR–GOL, therefore, shares salient features with HBV; that is, hepatotropic infection, hepatocyte-restricted gene expression and development of persistent infection. To follow and characterize antigen-specific CD8 T cells, we transferred 100 naive ovalbumin-specific T cell receptor (TCR)-transgenic CD8 T cells the day before infection which were identified through the expression of a congenic marker (CD45.1) (Extended Data Fig. 1g). In hepatic antigen-specific CD8 T cells after resolved infection, phenotypic profiling showed mutually exclusive expression of the chemokine receptors CXCR6 and CX₃CR1, whereas in spleen only CX₃CR1[+] cells were detected (Fig. 1b–d). The antigen-specific CXCR6[+] CD8 T cells co-expressed CD69 and GzmB (Fig. 1b–f), consistent with induction of liver-resident

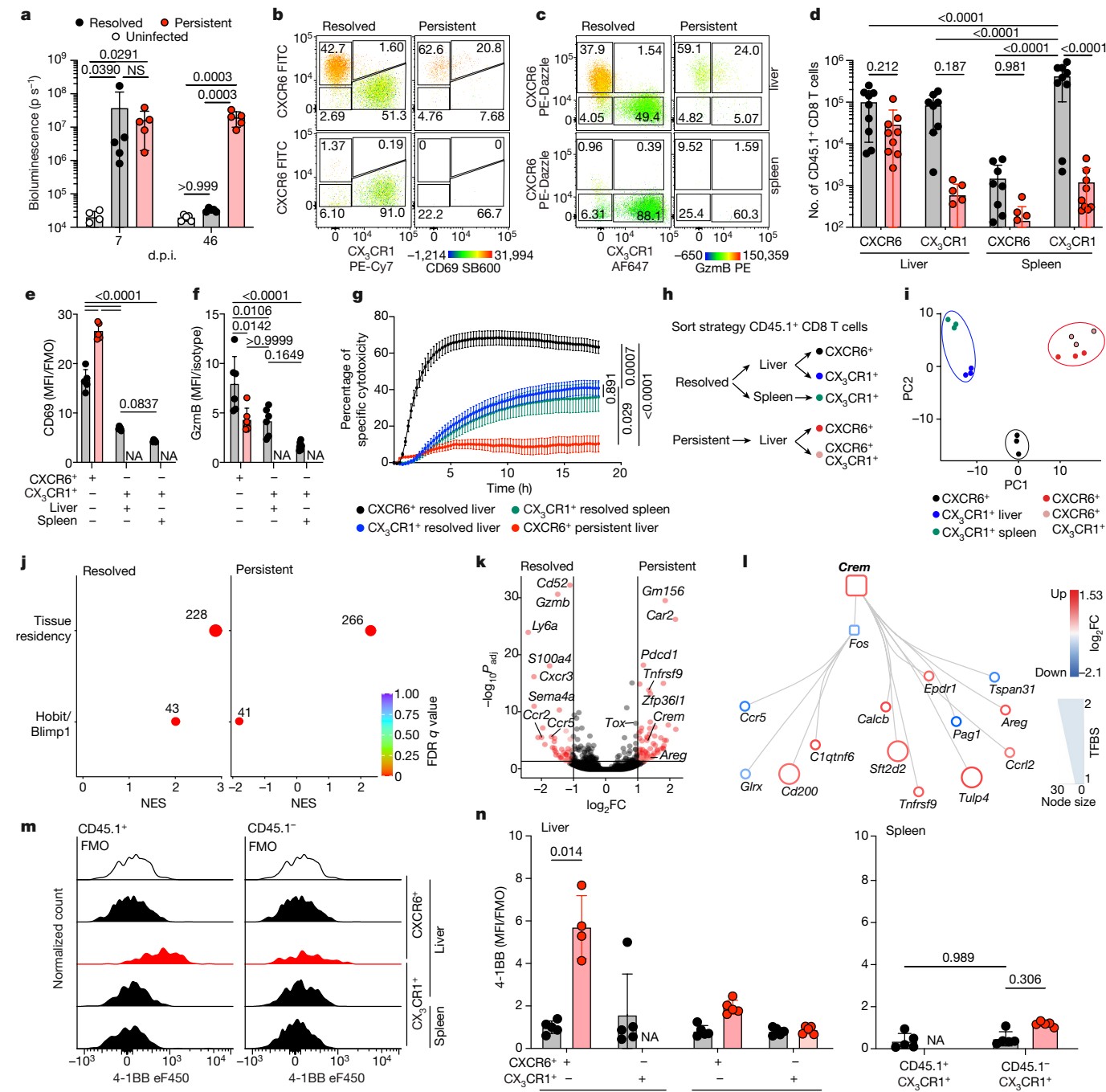

**Fig. 1 | Dysfunctional hepatic virus-specific CXCR6+ CD8 T cells characterized by enhanced CREM activity during persistent hepatotropic infection.**
**a**, Liver bioluminescence in vivo imaging of Ad–CMV–GOL (resolved), Ad–TTR–GOL (persistent) infected or uninfected mice. *P* values determined by one-way analysis of variance (ANOVA) with Tukey's multiple comparisons per timepoint (*n* = 5). **b,c**, Expression of CXCR6, CX₃CR1 and either CD69 (**b**) or GzmB (**c**) by antigen-specific CD45.1+ CD8 T cells in liver and spleen at 45 days post infection (d.p.i.). **d–f**, Quantification of CXCR6 and CX₃CR1 (**d**), CD69 (**e**) and GzmB (**f**) expression data from **b** and **c**. *P* values determined by two-way-ANOVA with Tukey's multiple comparison for adjusted *P* value ($P_{adj}$) (*n* = 8 (**d**); *n* = 5 (**e**); *n* = 5 (**f**)). **g**, Real-time specific cytotoxicity of CD45.1+ CD8 T cells against OVA₂₅₇₋₂₆₄ peptide-loaded hepatocytes. *P* values determined by one-way ANOVA with Tukey's multiple comparison of area under the curve (AUC) for $P_{adj}$ (*n* ≥ 3). **h**, Scheme of CD45.1+ CD8 T cell FACSorting for RNA-seq analysis. **i**, Principal component (PC) analysis of RNA-seq results (*n* = 3). **j**, GSEA in liver CD45.1+CXCR6+

CD8 T cells from resolved (left) and persistent (right) infection for a tissue-residency signature and Hobit- and Blimp1-dependent genes (permutation test with Benjamini–Hochberg false discovery rate (FDR)). NES, normalized enrichment score, **k**, Differentially expressed genes (DEGs) in liver CD45.1+CXCR6+ CD8 T cells during persistent infection or after resolved infection (red, $P_{adj}$ < 1.31 (*P* < 0.05 Wald test with Benjamini–Hochberg's correction) and log₂-transformed fold change (FC) > 1 or >−1, *n* = 3). **l**, Transcription factor network analysis comparing CD45.1+CXCR6+ CD8 T cells in persistent and resolved infection (*n* = 3). TFBS, transcription factor-binding site. **m,n**, 4-1BB expression by virus-specific CD45+ CD8 T cells compared to bulk CD45.1− CD8 T cells at 45 d.p.i. (**m**) and quantification (**n**). *P* values determined by two-way ANOVA with Tukey's multiple comparison for $P_{adj}$ (*n* = 5). In **a–g,m,n**, one out of two or more independent experiments shown; NS, not significant, *P* ≥ 0.05, *P* < 0.05, **P* < 0.01, ****P* < 0.001, *****P* < 0.0001. Data are mean and s.d. FMO, fluorescence minus one; MFI, geometric mean fluorescence intensity; NA, not analysed.

memory or effector-memory CD8 T cells, which are characterized by CXCR6 or $CX_3CR1$ expression, respectively[20–22]. Both CXCR6$^+$CD8 T cells and $CX_3CR1^+$CD8 T cells from the liver and spleen after resolved infection efficiently eliminated target cells and produced interferon-γ (IFNγ) and tumour necrosis factor (TNF) ex vivo after cognate stimulation (Fig. 1g and Extended Data Fig. 1h,i). By contrast, during persistent infection, antigen-specific CD8 T cells were mainly found in the liver and expressed CXCR6 and CD69 but lost GzmB expression, whereas specific effector $CX_3CR1^+$CD8 T cells were scarce in both liver and spleen (Fig. 1b–f). These CXCR6$^+$CD8 T cells expressed high concentrations of co-inhibitory molecules (PD-1, TIGIT, TIM-3 and LAG3) and the transcription factor TOX (Extended Data Fig. 1j,k), lacked antigen-specific cytotoxicity and failed to produce cytokines after antigen stimulation (Fig. 1g and Extended Data Fig. 1h,i), which is reminiscent of T cell exhaustion observed during persistent lymphocytic choriomeningitis virus (LCMV) infection. Together, antigen-specific CD8 T cells during persistent hepatotropic infection were retained in the liver and lost GzmB expression as well as their cytotoxic effector function, which raised the question of which transcriptional programmes are responsible for this loss of function.

We performed RNA sequencing (RNA-seq) analysis of FACS-sorted antigen-specific CXCR6$^+$ and $CX_3CR1^+$CD45.1$^+$CD8 T cells from the liver and spleen after resolved hepatotropic infection and during persistent infection and identified distinct transcriptional profiles (Fig. 1h,i). CD69$^+$CXCR6$^+$CD8 T cells after acute resolved infection were characterized by a tissue-residency gene expression profile together with regulation by the tissue-residency-mediating transcription factors Hobit and Blimp1 (Fig. 1j and Extended Data Fig. 2a). By contrast, although CD69$^+$CXCR6$^+$CD8 T cells during persistent infection showed enhanced expression of genes associated with tissue residency, they did not show induction of transcriptional targets of Hobit and Blimp1 (Fig. 1j, Extended Data Fig. 2a and Supplementary Tables 6 and 7). Interestingly, CXCR6$^+$CX$_3$CR1$^+$CD8 T cells during persistent infection shared an almost identical gene expression pattern with CXCR6$^+$CD8 T cells (Fig. 1i and Supplementary Tables 1–5), indicating a transition of CXCR6$^+$CX$_3$CR1$^+$ into CXCR6$^+$CD8 T cells during persistent infection. Together, this indicates that CD69$^+$CXCR6$^+$CD8 T cells during persistent infection were not bona fide tissue-resident memory T cells.

To define the distinct transcriptional programmes associated with T cell dysfunction, we compared CXCR6$^+$CD8 T cells after resolved infection to CXCR6$^+$CD8 T cells during persistent infection. Expression of effector molecules was detected in CXCR6$^+$CD8 T cells after resolved infection, such as *Gzmb*, *Cxcr3*, *Ccr2* and *Ccr5*, in contrast to enriched expression of co-inhibitory receptors by CXCR6$^+$CD8 T cells during persistent hepatic infection, such as *Pdcd1* and *Lag3* (Fig. 1k). Of note, increased *Tox* gene expression did not reach statistical significance (Fig. 1k). We next used an unbiased transcription factor network analysis to identify transcription factors involved in the shutdown of T cell effector function in liver CXCR6$^+$CD8 T cells. This revealed cAMP-responsive element modulator (CREM) as the only transcription factor with predicted enhanced activity in CXCR6$^+$CD8 T cells during persistent hepatotropic infection (Fig. 1l). Gene set enrichment analysis (GSEA) corroborated enhanced expression of CREM-dependent genes in CXCR6$^+$CD8 T cells during persistent hepatotropic infection (Extended Data Fig. 2b). In line with high CREM transcriptional activity in antigen-specific CXCR6$^+$CD8 T cells during persistent infection, we detected increased expression of the CREM target gene *Tnfrsf9* (4-1BB) at protein level (Fig. 1m,n). We did not see evidence for TOX downregulating CD8 T cell effector function in the transcription factor network analysis, indicating that TOX may not be involved in the loss of CD8 T cell function during infection with hepatotropic viruses. Conversely, virus-specific (gp33) CD8 T cells isolated from the liver during systemic infection with LCMV, which infects all cell populations but lymphocytes[10,23] and induces T cell exhaustion, did not show enhanced expression of CREM-dependent genes (Extended Data Fig. 2c–e). This

suggested that distinct mechanisms mediate the loss of T cell effector functions during hepatotropic viral infection compared to the repetitive TCR stimulation leading to T cell exhaustion during persistent systemic LCMV clone 13 infection[10]. Together, antigen-specific CD8 T cells during persistent hepatotropic infection were characterized by loss of GzmB expression and cytotoxicity in addition to increased CREM expression and transcriptional activity.

## CREM signature in persistent HBV infection in mice

We next explored whether CD8 T cells, during persistent HBV gene expression in hepatocytes, similarly showed a CREM signature. Preclinical models for the study of HBV-specific immunity are hampered by a strict species restriction, which can be overcome by HBV genome transfer into hepatocytes using shuttle viruses such as adeno-associated virus (AAV) or adenovirus leading to the expression of HBV genes under the control of HBV-specific promoters[24–26]. AAV–HBV transduction of hepatocytes leads to HBV-specific immune tolerance with very scarce HBV-specific CD8 T cells[15,24,27], which are not sufficient for detailed analysis (Extended Data Fig. 3a–c). We, therefore, established a preclinical in vivo model in which hepatocytes after transduction with $1 \times 10^7$ international units (IU) of Ad–HBV were cleared by virus-specific immunity, resulting in more than 20-fold reduction in HBV copies to almost undetectable amounts in the liver from days 8 to 45 after transduction. By contrast, persistent HBV gene expression in hepatocytes developed after transduction with $1 \times 10^8$ IU of Ad–HBV, shown by continuously high-serum HBeAg amounts, a fourfold reduction in HBV copies and persistence of HB$_{core}^+$ hepatocytes in liver tissue (Fig. 2a and Extended Data Fig. 3d–f). To overcome variable surface expression of the TCR during chronic infection[28] and unequivocally identify HBV-specific CD8 T cells, we adoptively transferred naive CD45.1$^+$HB$_{core}$-specific CD8 T cells from Cor93-transgenic mice (HB$_{core}$CD8 T cells) the day before Ad–HBV transduction. After clearance of Ad–HBV-transduced hepatocytes (45 d.p.i.), liver CD45.1$^+$HB$_{core}$CD8 T cells were either CXCR6$^+$CD69$^+$GzmB$^+$ or CX$_3$CR1$^+$CD69$^-$GzmB$^{low}$, whereas in the spleen only CX$_3$CR1$^+$CD69$^-$GzmB$^{low}$ CD8 T cells were detected (Fig. 2b–f and Extended Data Fig. 3g,h). During persistence of Ad–HBV–transduced hepatocytes (45 d.p.i.), CD45.1$^+$HB$_{core}$CD8 T cells retained in the liver were CXCR6$^+$CD69$^+$GzmB$^-$ with variable co-expression of CX$_3$CR1, whereas almost no CD45.1$^+$HB$_{core}$CD8 T cells were detected in the spleen (Fig. 2b–f and Extended Data Fig. 3g,h). Liver GzmB$^-$CXCR6$^+$HB$_{core}$CD8 T cells were PD-1$^{hi}$TIGIT$^{hi}$TOX$^{hi}$ (Extended Data Fig. 3i,j) and did not produce any cytokines after ex vivo stimulation (Fig. 2g,h). These data are consistent with the development of liver-resident memory CXCR6$^+$CD8 T cells after clearance of hepatocytes expressing HBV genes and hepatic accumulation of GzmB$^-$CXCR6$^+$CD8 T cells with loss of effector function during persistent HBV gene expression in hepatocytes.

We next evaluated the transcriptional regulation of GzmB$^-$CXCR6$^+$ HB$_{core}$CD8 T cells during persistent HBV gene expression in hepatocytes compared to functional CXCR6$^+$HB$_{core}$CD8 T cells after clearance of transduced hepatocytes using Smart-Seq2 as the preferred method to analyse low-frequency T cell populations (Fig. 2i). This showed a distinct transcriptional profile of dysfunctional CXCR6$^+$HB$_{core}$CD8 T cells during persistent HBV gene expression in hepatocytes compared to CXCR6$^+$ and CX$_3$CR1$^+$HB$_{core}$CD8 T cells after resolved Ad–HBV infection and non-HBV-specific CD45.1$^-$CD8 T cells from the liver and spleen from the same mice (Fig. 2j,k). Notably, during persistent HBV gene expression in hepatocytes virus-specific CXCR6$^+$HB$_{core}$CD8 T cells had increased expression of *Crem* and genes encoding co-inhibitory receptors (Fig. 2k,l). Applying the GENIE3 algorithm to infer transcriptional regulatory networks[29], we confirmed enhanced transcriptional activity of CREM, as well as enhanced activities of TEAD1 and HEYl, both TGFβ-regulated transcription factors[30] but not TOX (Fig. 2m and Extended Data Fig. 3k). Enhanced CREM activity was similarly detected (Fig. 2n) in a recently published dataset from dysfunctional

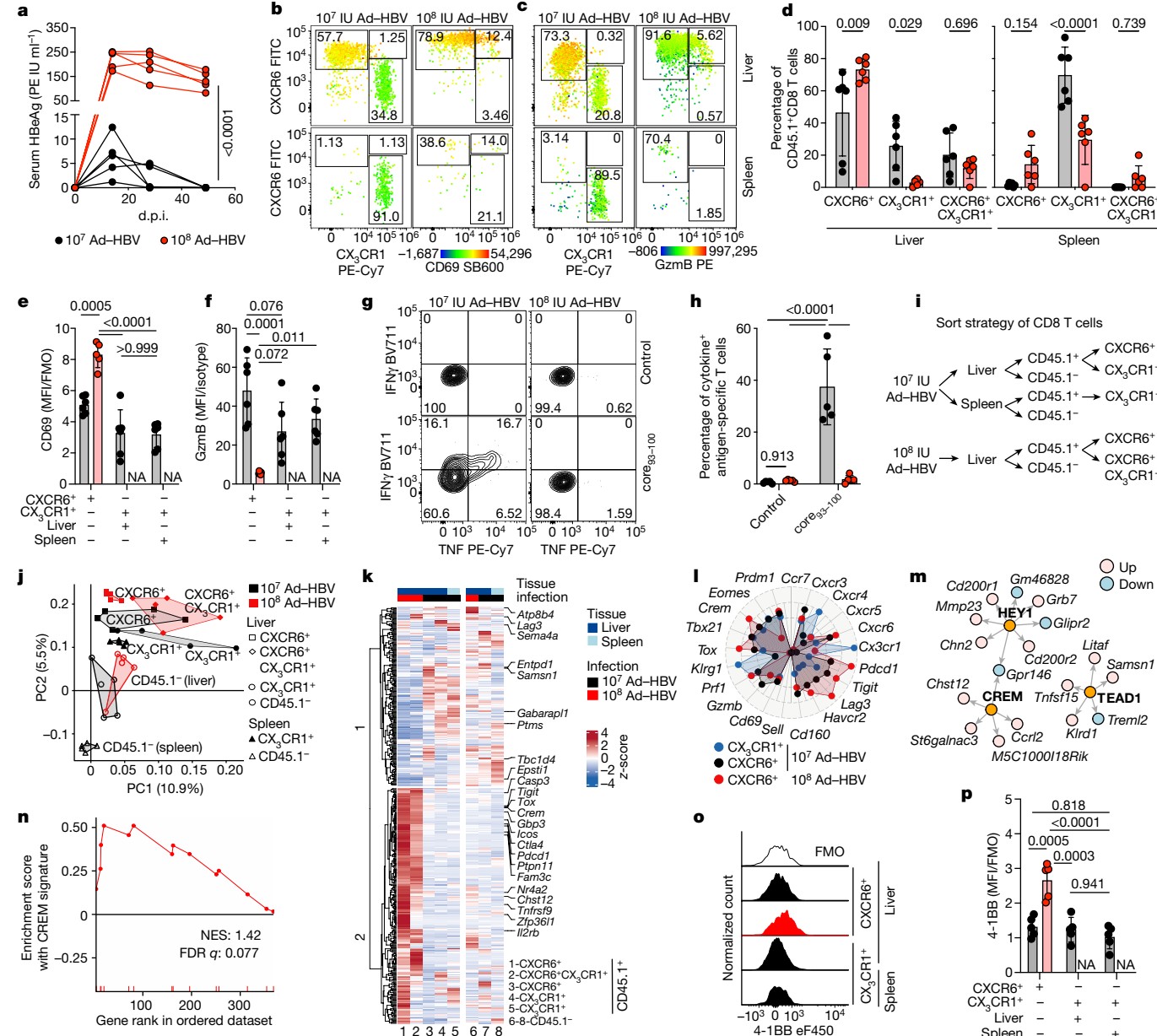

**Fig. 2 | A CREM signature in dysfunctional HBV-specific CXCR6⁺ CD8 T cells during persistent HBV gene expression in mice. a**, Serum HBeAg concentration after Ad–HBV transduction. PE IU, Paul Erlich Institute units. *P* values determined by two-way ANOVA with Sidak's multiple comparison for $P_{adj}$ ($n = 5$). **b,c**, Expression of CXCR6, CX₃CR1 and either CD69 (**b**) or GzmB (**c**) by liver HB$_{core}$-specific CD45.1⁺ CD8 T cells at 45 d.p.i. Quantification of CXCR6, CX3CR1 (**d**), CD69 (**e**) and GzmB (**f**) expression data from **b** and **c**. *P* values determined by one-way (**e,f**) or two-way (**d**) ANOVA with Tukey's multiple comparison for $P_{adj}$ ($n = 6$) (**d**). **g,h**, Expression of IFNγ and TNF by liver CXCR6⁺ HB$_{core}$ CD8 T cells after ex vivo stimulation with HBcore$_{93-100}$ peptide (**g**) and quantification (**h**). *P* values determined by two-way ANOVA with uncorrected Fisher's least significant difference (LSD) test for individual *P* values ($n = 5$).

**i**, Scheme of CD8 T cell FACSorting for RNA-seq analysis. **j**, Principal component analysis of Smart-Seq2 data from sorted HB$_{core}$ CD8 T cells isolated at 50 d.p.i. ($n \geq 4$). **k**, Hierarchical clustering of DEGs ($n \geq 4$, 50 d.p.i., $\log_{CPM} \geq 0$, FDR < 0.05, $\log_{FC} \geq 1$). **l**, Radar plot of selected marker genes. **m**, Transcriptional regulatory networks inferred by GENIE3 illustrating enhanced expression and transcriptional activity of CREM, HEYl and TEAD1 ($n \geq 4$). **n**, GSEA for the cAMP/CREM signature in liver HB$_{core}$-specific CD8 T cells recognizing antigen on hepatocytes[6]. **o,p**, Expression of 4-1BB by HB$_{core}$ CD8 T cells (**o**) and quantification (**p**). *P* values determined by one-way ANOVA with Tukey's multiple comparison ($n = 5$). In **a–h,o,p**, one out of two or more independent experiments is shown; $P \geq 0.05$, *$P < 0.05$, **$P < 0.01$, ***$P < 0.001$, ****$P < 0.0001$. Data are mean and s.d.

HB$_{core}$-specific CD8 T cells in the livers of transgenic mice expressing HBV antigens in hepatocytes[6]. Consistent with increased CREM activity, liver CXCR6⁺ HB$_{core}$ CD8 T cells expressed the CREM target gene 4-1BB during persistent HBV gene expression in transduced hepatocytes but not after resolved Ad–HBV infection (Fig. 2o,p). Thus, enhanced CREM transcriptional activity was a distinguishing feature of liver HB$_{core}$-specific GzmB⁻ CXCR6⁺ CD8 T cells during persistent HBV infection.

## CREM signature in patients with chronic HBV

To translate our findings beyond preclinical models of persistent HBV gene expression in hepatocytes, we analysed circulating CD8 T cells from five HLA-A2⁺ patients with chronic hepatitis B who did not receive antiviral treatment and were characterized by loss of HBeAg, low-serum HBsAg, low amounts of circulating HBV DNA and no signs of continuing liver damage (Supplementary Table I). Scarce HBV-specific CD8 T cells

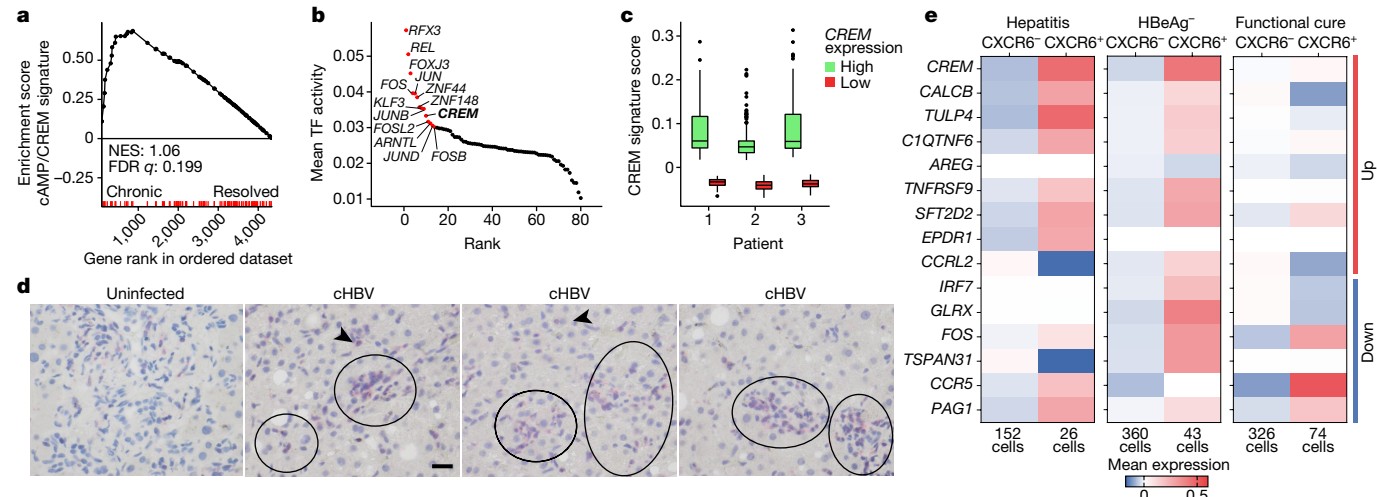

**Fig. 3 | High *CREM* expression and CREM transcriptional activity in circulating and hepatic HBV-specific CD8 T cells in patients with chronic hepatitis B. a**, GSEA for the cAMP/CREM signature using Smart-Seq2 results from circulating HBV-specific CD8 T cells isolated from patients with chronic hepatitis B (permutation test with Benjamini–Hochberg FDR). **b**, Transcription factor (TF) activity inferred using pySCENIC in scRNA-seq results from circulating HBV-specific CD8 T cells from patients with chronic hepatitis B; top 20 active transcription factors shown as red dots. **c**, CREM transcriptional activity in CREM+ HBV-specific CD8 T cells from patients shown in **b** (median, 25th and 75th percentiles, highest and lowest values limited at 1.5× interquartile range shown, two-sided Wilcoxon test with $P < 2 \times 10^{-16}$ comparing high versus low *CREM* expression for patients 1–3) ($n = 3$). **d**, Immunohistochemistry for CD3 (red) and RNAscope detecting CXCR6 (purple) in T cells in livers of patients with chronic hepatitis B (cHBV) ($n = 11$) compared to uninfected liver tissue ($n = 5$). Scale bar, 20 μm. **e**, CREM signature in 977 hepatic HBV-specific CD8 T cells comparing CXCR6− to CXCR6+ CD8 T cells obtained by fine-needle aspiration from 21 patients with continuing hepatitis (HBeAg+ or HbeAg−), HbeAg− chronic HBV infection and individuals with functional cure from chronic hepatitis B (total of $n = 11$ patients), numbers of HBV-specific CD8 T cells detected are shown below the graph.

were sorted using peptide-loaded HLA-A2 multimers and subjected to Smart-Seq2 sequencing. These HB$_{core}$-specific CD8 T cells showed enrichment of transcriptional targets of cAMP/CREM when compared to HB$_{core}$-specific CD8 T cells from two HLA-A2+ individuals with resolved HBV infection (Fig. 3a). As a control, we analysed circulating bulk non-HBV-specific CD8 T cells. When comparing these poly-specific CD8 T cells from patients with chronic hepatitis B to those with resolved HBV infection, we did not detect a cAMP/CREM signature (Extended Data Fig. 4a), together indicating that HB$_{core}$-specific CD8 T cells were characterized by a CREM signature only during chronic HBV infection.

Next, we performed single-cell RNA sequencing (scRNA-seq) of circulating HB$_{core}$-specific CD8 T cells from three HLA-A2+ patients with chronic HBV infection. Although the numbers of HBV-specific CD8 T cells obtained from these patients were too low to enable the detection of distinct cell clusters, transcription factor activity analysis in HB$_{core}$-specific CD8 T cells showed CREM to be among the top ten most active transcription factors in these patients (Fig. 3b and Extended Data Fig. 4b). When we stratified HB$_{core}$-specific CD8 T cells according to CREM expression, high CREM expression was associated with high transcriptional CREM activity (Fig. 3c). We confirmed enhanced CREM transcriptional activity in a second cohort of four HLA-A2+ patients with chronic HBV infection (Extended Data Fig. 4c). Among all HB$_{core}$-specific CD8 T cells subjected to scRNA-seq analysis (1,123 cells), approximately 25% (290 cells) had a high UCell score for CREM transcriptional activity (Extended Data Fig. 4d,e). By contrast, human immunodeficiency virus-specific CD8 T cells from patients with human immunodeficiency virus during persistent infection[31] did not show enrichment for a cAMP/CREM signature in their transcriptional profiles (Extended Data Fig. 4f). Thus, enhanced *CREM* expression and CREM activity are found in circulating HBV-specific CD8 T cells of patients with chronic HBV infection.

To address the question of whether HBV-specific CD8 T cells in the liver during chronic hepatitis B show a CREM signature, we investigated liver biopsies of patients with chronic hepatitis B by immunohistochemistry. We found CXCR6+CD3+ T cells in the livers of patients with chronic ($n = 11$) (Fig. 3d), with a frequency of 2.7–15.4% of all T cells,

whereas no CXCR6+CD3+ T cells were detected in livers from patients without HBV infection ($n = 5$). However, immunohistochemistry did not allow us to detect whether HBV-specific CD8 T cells were among the CXCR6+ T cells. Therefore, we analysed intrahepatic virus-specific CD8 T cells isolated by fine-needle liver aspirates from patients with chronic hepatitis B in different phases of infection by scRNA-seq analysis. Frequencies of *CXCR6*+ HBV-specific CD8 T cells were in the range 11.9–22.7% of all hepatic HBV-specific CD8 T cells detected (Fig. 3e), which corresponds with the proportions of CXCR6+ T cells detected by immunohistochemistry. We detected an increased expression of *CREM* and CREM target genes in *CXCR6*+ compared to *CXCR6*− HBV-specific CD8 T cells in patients with active chronic hepatitis and less pronounced in patients with HBeAg− chronic HBV infection (Fig. 3e). By contrast, HBV-specific T cells from patients with a functional cure of chronic HBV infection did not have this increased expression of *CREM* and CREM target genes (Fig. 3e). Together, these data demonstrate that expression of HBV antigens in infected hepatocytes during chronic hepatitis B is associated with the presence of intrahepatic HBV-specific CXCR6+ CD8 T cells with increased expression of *CREM* and CREM target genes.

These results raised the possibility that CREM itself might mediate decreased CD8 T cell effector function and led us to investigate its direct influence on T cell function. The *CREM* gene is composed of several exons[32] and various CREM isoforms contribute to T cell activation[33]. ICER is a unique CREM isoform which lacks a transcriptional activation domain, thereby acting as a repressor of CREB-induced target gene transcription[34–36]. We generated *Icer$^{fl/fl}$* mice (Methods; Extended Data Fig. 5a) and crossed *Icer$^{fl/fl}$* mice to *Cd4$^{cre}$* for a T cell-selective loss of ICER expression (*Cd4$^{cre}$ × Icer$^{fl/fl}$* mice). ICER-deficient CD8 T cells did not show increased activation and proliferation after TCR stimulation in vitro (Extended Data Fig. 5b–d). Furthermore, no immune-mediated clearance of infected hepatocytes was observed in Ad−HBV or Ad−TTR−GOL-infected *Cd4$^{cre}$ × Icer$^{fl/fl}$* mice (Extended Data Fig. 5e–k). Notwithstanding the reports on *CREM* expression by dysfunctional CD8 T cells or CD4 T cells[37,38] and regulatory CD4 T cells[36], our data provide evidence that increased CREM/ICER activity is not itself causing the loss of effector function in HBV-specific CD8 T cells during chronic liver infection.

## Immune rheostat blocks TCR signaling

To investigate the influence of the liver microenvironment on T cell function, we isolated antigen-specific CD45.1[+]CXCR6[+]CD8 T cells from the livers of mice with persistent or resolved infection and transferred them into recently infected mice which cleared infection or developed persistent infection, respectively (Extended Data Fig. 6a). When isolated at day 30 after transfer, CXCR6[+] T cells isolated from livers after resolved infection and transferred into mice developing persistent infection lost GzmB expression (Fig. 4a,b). Conversely, CD45.1[+]CXCR6[+] T cells from persistently infected mice, and adoptively transferred into mice resolving acute infection, gained GzmB expression (Fig. 4a,b). This points towards liver tissue factors that reversibly modulate the function of CD8 T cells recognizing their cognate antigen on virus-infected hepatocytes.

Re-analysing the transcriptional signature of CXCR6[+]CD8 T cells during persistent infection, we noted downregulation of genes associated with TCR signalling (Fig. 4c), indicative of impaired activation-induced signal transduction. The inability of hepatic T cells to respond to activation (Figs. 1 and 2) and their increased *CREM* expression led us to investigate cAMP signalling, which is known to induce *CREM* expression through phosphorylation and activation of protein kinase A (PKA) and to block signalling processes[39]. Regulatory CD4 T cells are known to inhibit CD8 T cells in a cAMP-dependent fashion[40,41] but their depletion does not affect the outcome of Ad–HBV-infection[42], prompting us to search whether other cells in the liver engaged in close contact with virus-specific CD8 T cells to induce cAMP signalling.

During persistent hepatic infection, we found virus-specific CXCR6[+]CD8 T cells to engage in very close physical contact with liver sinusoidal endothelial cells (LSECs) and establish a large contact surface with LSECs (Fig. 4d–f and Extended Data Fig. 6b), consistent with the reported intravascular sinusoidal localization of HBV-specific CD8 T cells that recognize their cognate antigen on infected hepatocytes by protruding their extensions through LSEC fenestrae[7]. Of note, the distance of CXCR6[+]CD8 T cells to liver dendritic cells was 100-fold higher and did not differ between resolved and persistent infection (Extended Data Fig. 6b–d). LSECs are known as tolerogenic antigen-(cross)presenting cells which induce dysfunction in naive CD8 T cells[43] but LSECs failed to cross-present HB$_{core}$ antigen to HB$_{core}$CD8 T cells[7] (Extended Data Fig. 6e,f), which points to distinct mechanisms by which LSECs influence those effector CD8 T cells engaging in close physical contact during persistent infection.

In line with increased cAMP signalling, we found increased PKA phosphorylation at serine 114 (pPKA) in antigen-specific CXCR6[+]CD8 T cells during persistent infection compared to resolved infection and in polyclonal unspecific CD8 T cells (Fig. 4g,h and Extended Data Fig. 6g,h). The pPKA concentrations increased in CD8 T cells after coculture with LSECs but not hepatocytes or dendritic cells (Fig. 4i,j), indicative of LSECs enhancing PKA activation in T cells in situ. Furthermore, coculture with LSECs led to the downregulation of GzmB expression in CD8 T cells (Fig. 4k,l). To evaluate the effect of increased cAMP–PKA signalling on T cell function, we exposed functional liver CXCR6[+]CD8 T cells isolated from mice with resolved infection to forskolin (Fsk) which increases cAMP generation by stimulating adenylyl cyclase[44]. Fsk treatment of CXCR6[+]CD8 T cells increased pPKA concentrations and 4-1BB expression caused loss of GzmB and cytokine expression after stimulation and abrogated cytotoxic effector function against peptide-pulsed hepatocytes (Fig. 4m–p), thus phenocopying the loss of effector function in liver CXCR6[+]GzmB[−]CD8 T cells against infected hepatocytes during persistent infection.

This led us to investigate which mechanisms upstream of adenylyl cyclase were involved in regulating T cell effector function. Adenosine receptor signalling leads to an increase in cAMP concentrations and inhibition of T cell function[45]. However, LSECs did not express the ectonucleotidase CD39 (Extended Data Fig. 7a,b), which is required

for the breakdown of extracellular ATP into ADP to generate adenosine[46]. Moreover, inhibition of adenosine receptor signalling, which activates adenylyl cyclase[47], did not rescue GzmB expression of T cells in coculture with LSECs (Extended Data Fig. 7c,d), making a major contribution of purinergic signalling to LSEC-mediated loss of effector function in T cells unlikely. Likewise, inhibition of PTPN22, a type I interferon-induced inhibitory tyrosine phosphatase detected during persistent LCMV infection[38], did not rescue GzmB expression of T cells in coculture with LSECs (Extended Data Fig. 7e). LSECs constitutively generated high concentrations of prostaglandin E$_2$ (PGE$_2$) (Extended Data Fig. 7f), a known inducer of increased cAMP signalling[48]. PGE$_2$ downregulated T cell effector function (Extended Data Fig. 7g) and pharmacological blockade of the PGE$_2$-producing enzyme cyclooxygenase-2 increased GzmB expression in CD8 T cells cocultured with LSECs (Extended Data Fig. 7h–i), albeit only at high concentration and during constant exposure. However, preventing cAMP generation by selective inhibition of adenylyl cyclase rescued T cells from losing GzmB expression when in coculture with LSECs (Extended Data Fig. 7j). Furthermore, the inhibition of adenylyl cyclase in virus-specific CD8 T cells adoptively transferred into mice with persistent hepatotropic infection prevented PKA phosphorylation and rescued their GzmB expression and cytotoxic effector function in situ (Fig. 4q–s and Extended Data Fig. 7k), which together indicates that the induction of adenylyl cyclase activity in T cells was critical for their loss of function in vivo.

Signalling downstream of cAMP is transmitted through PKA or, alternatively, the exchange protein directly activated by cAMP (EPAC)[49]. However, only inhibition of PKA but not EPAC rescued GzmB expression in CD8 T cells from the regulatory function of LSECs (Fig. 4t). Consistently, selective activation of PKA but not EPAC led to the loss of cytokine expression by CD8 T cells (Extended Data Fig. 7l). Together, these results demonstrate that control of effector CD8 T cell function through LSECs was mediated through PKA signalling. Of note, during persistent liver infection, we did not observe increased PKA phosphorylation in other hepatic immune effector cell populations, such as NK cells, NKT cells or CXCR6[+]CD4 T cells (Extended Data Fig. 7m), indicating that adenylyl cyclase activity was selectively induced in virus-specific CD8 T cells.

Increased adenylyl cyclase–cAMP–PKA signalling leads to activation of the protein tyrosine kinase Csk which blunts TCR signalling by reducing the activation of src kinases such as Lck[50] and may thereby affect TCR-associated signalling processes. In virus-specific CXCR6[+]CD8 T cells, which were isolated from livers of mice with persistent hepatic infection, we detected after ex vivo TCR stimulation reduced phosphorylation of Lck and Akt (Fig. 4u,v and Extended Data Fig. 7n,o), which are key signal-transducing molecules downstream of TCR signalling[51]. Together, these results demonstrate that during persistent hepatotropic infection, antigen-specific CD8 T cells recognizing their cognate antigen on infected hepatocytes closely interact with LSECs, which increases adenylyl cyclase activity in these T cells and results in enhanced cAMP–PKA signalling, which prevents their activation (Fig. 4w). Thus, close contact with LSECs, which function as liver immune rheostat, curbs the effector function in virus-specific CD8 T cells recognizing their cognate antigen on infected hepatocytes.

## Discussion

Chronic infection with HBV arises from the failure of virus-specific immunity to control viral replication and eliminate virus-infected hepatocytes[11,18]. Here we demonstrate that LSECs act as a liver immune rheostat which curbs the effector function of those virus-specific CD8 T cells recognizing their antigen on infected hepatocytes by rendering them dysfunctional through increased cAMP–PKA signalling. Enhanced cAMP–PKA signalling activates the kinase Csk and activated Csk, in turn, inhibits Lck, which shuts down TCR signalling[49–51]. This process may contribute to the scarcity and loss-of-function of HBV-specific T cells in the liver because it deprives them of the activation signals necessary

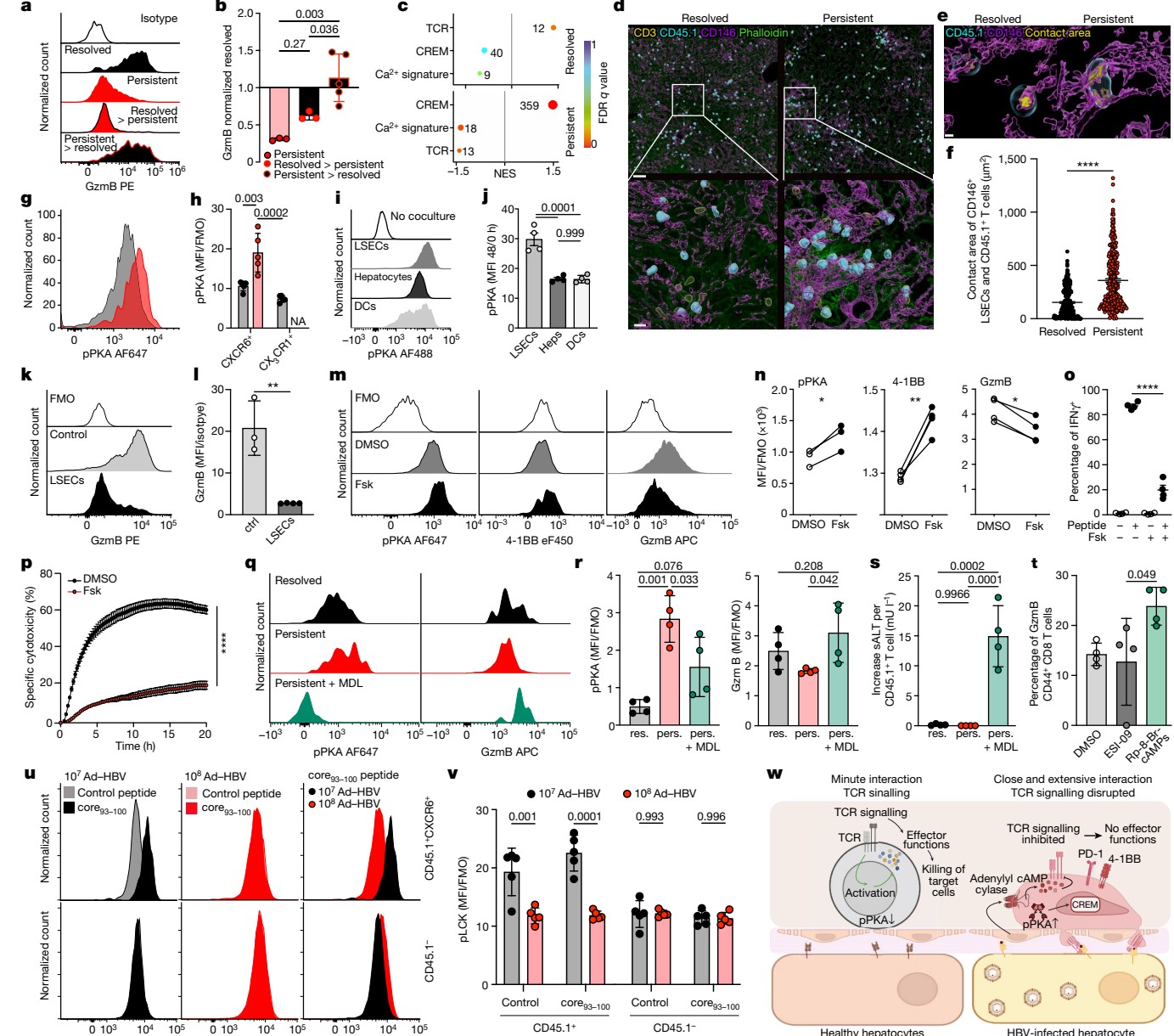

**Fig. 4 | A liver immune rheostat acts on antigen-specific CXCR6⁺ T cells through an inhibitory adenylyl cyclase–cAMP–PKA axis to disrupt TCR signalling. a,b**, GzmB expression by liver CXCR6⁺CD45.1⁺CD8 T cells from resolved or persistent infection (30 d.p.i.), transferred into mice with acute-resolving or persistent infection (2 d.p.i.), analysis 18 days after transfer (**a**) (*n* = 3 for groups persistent and resolved into persistent transfer, *n* = 4 for group persistent into resolved transfer) and quantification (**b**) by one-way ANOVA with Tukey's multiple comparison for *P*₍adj₎. **c**, GSEA of liver CXCR6⁺CD45.1⁺CD8 T cells from resolved or persistent infection (permutation test with Benjamini–Hochberg FDR, *n* = 3). **d,e**, Three-dimension-rendered volumetric confocal images of interacting CD45.1⁺ T cells and CD146⁺ LSECs at low (**d**) and high (**e**) resolution. Scale bars, 50 μm (**d**, top), 10 μm (**d**, bottom), and 2 μm (**e**) (*n* = 3). **f**, Quantification of T cell–LSEC contact area (*n* = 3, unpaired two-sided *t*-test: *P* < 0.0001). **g,h**, Phosphorylated (S114) PKA (pPKA) concentrations in liver CD45.1⁺CXCR6⁺CD8 T cells at 45 d.p.i. (resolved versus persistent infection) (**g**) and quantification (**h**) (*n* = 5, two-way ANOVA with Tukey's multiple comparison). **i,j**, Change in pPKA concentrations in CD8 T cells cocultured with liver cells (**i**) and quantification (**j**) (*n* = 4, one-way ANOVA with Tukey's multiple component for *P*₍adj₎). DCs, dendritic cells; Heps, hepatocytes. **k,l**, GzmB expression by T cells cocultured with LSECs (**k**) and quantification (**l**) (*n* = 3, unpaired two-sided *t*-test *P* = 0.0023). **m–p**, pPKA, 4-1BB and GzmB expression

(**m,n**), IFNγ expression (**o**) and antigen-specific cytotoxicity (**p**) by Fsk-treated CD45.1⁺CXCR6⁺CD8 T cells from resolved infection (*n* = 4; paired two-sided *t*-test pPKA *P* = 0.0323, 4-1BB *P* = 0.0045, GzmB *P* = 0.0069 (**n**); two-way ANOVA with Sidak's multiple comparison without *P* = 0.9970, with peptide *P* < 0.0001 (**o**); unpaired two-sided *t*-test on AUCs, *P* < 0.0001) (**p**)). **q,r**, pPKA and GzmB concentrations in virus-specific CD8 T cells treated with the adenylyl cyclase inhibitor MDL-12,330A before transfer into mice with persistent infection, analysis 3 days later (**q**) and quantification (**r**) (*n* = 4, pPKA: ordinary one-way ANOVA with Tukey's multiple comparison for *P*₍adj₎, GzmB: two-sided unpaired *t*-test). **s**, Serum alanine transaminase increase/hepatic CD45.1⁺/⁺ CD8 T cell (*n* = 4, ordinary one-way ANOVA with Tukey's multiple comparison for *P*₍adj₎). **t**, GzmB concentrations in PKA-inhibited (Rp-8-bromo-cAMPs) or EPAC-inhibited (ESI-09) CD8 T cells cocultured with LSECs (*n* = 4, one-way ANOVA with Tukey's multiple comparison). **u,v**, Phosphorylated pY394 Lck (pLck) concentrations in hepatic HB₍core₎-specific CD8 T cells after resolved and persistent infection and ex vivo peptide stimulation (**u**) and quantification (**v**) (*n* = 5, two-way ANOVA and Tukey's multiple comparison). **w**, Graphical abstract illustrating the function of the liver immune rheostat. Data are mean and s.d. (**b,h,j,l,o,r–t,v**) or mean and s.e.m. (**f,p**). In **a–e,g–v**, one of two or more independent experiments is shown; *P* ≥ 0.05, **P* < 0.05, ***P* < 0.01, ****P* < 0.001, *****P* < 0.0001. DCs, dendritic cells.

for proliferation and expansion, cytokine production and execution of specific cytotoxicity against HBV-infected hepatocytes. The unique micro-architecture of the liver, in which virus-specific CD8 T cells remain in sinusoidal vessels and reach through endothelial fenestrae to contact virus-infected hepatocytes[7], enforces the close physical contact of T cells with LSECs that increase T cell adenylyl cyclase activity and consequently inhibitory cAMP–PKA signalling. This process, which selectively, locally and dynamically induces the inhibitory cAMP axis in those virus-specific T cells, which engage in close contact with LSECs while they recognize their antigen on hepatocytes, may serve as a physiological mechanism to protect the liver from immune-mediated pathology but also favouring viral persistence at the same time. These roles are underscored by human HBV data which show higher CREM activity in HBV-specific CXCR6+ CD8 T cells in patients with active hepatitis and less CREM activity with increasing viral control and absence of liver damage.

By contrast, T cell exhaustion, which curtails T cell effector function during persistent infection with model viruses such as LCMV, is transcriptionally mediated through the exhaustion-inducing transcription factor TOX after repeated encounter of T cells with their cognate antigen[10,52,53]. T cell exhaustion develops early during infection and, through epigenetic imprinting[53,54], causes a permanent attenuation of effector function in T cells. Albeit at a lower concentration, exhausted T cells remain functionally competent[55]. We did not find evidence for the liver immune rheostat affecting LCMV-specific exhausted CD8 T cells during persistent LCMV infection and, conversely, did not find evidence for T cell exhaustion in HBV-specific CD8 T cells during persistent HBV infection. This probably results from LCMV infecting all organs and cells[56], which leads to ubiquitous antigen recognition on many cell populations by LCMV-specific CD8 T cells and does not support close physical interaction with LSECs. It is of note that CREM activity increases expression of the costimulatory molecule 4-1BB that might serve as a target for T cell activation. Compared to T cell exhaustion, the immune rheostat function of LSECs inhibits T cell effector function altogether and acts as a temporary brake of T cell effector function through a post-translational mechanism to block TCR signalling.

Understanding the molecular mechanisms that determine the loss of HBV-specific CD8 T cell effector function in chronic hepatitis B will be important to developing more efficient immune therapies. The strict tropism of HBV for hepatocytes avoiding infection of dendritic cells, the stealth function of HBV avoiding inflammation and the weak intrahepatic priming of virus-specific T cells[6,13,18] all contribute to insufficient priming of HBV-specific CD8 T cell immunity but the liver immune rheostat contributes to the inhibition of the effector function of HBV-specific CD8 T cells while they recognize HBV-infected hepatocytes. Modulating the function of the liver immune rheostat by targeting the inhibitory adenylyl cyclase–cAMP–PKA signalling axis might improve the efficacy of therapeutic approaches aiming at reconstituting HBV-specific CD8 T cell responses in chronic HBV infection.

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

Miriam Bosch[1,27], Nina Kallin[1,27], Sainitin Donakonda[1], Jitao David Zhang[2], Hannah Wintersteller[1], Silke Hegenbarth[1], Kathrin Heim[3], Carlos Ramirez[4], Anna Fürst[1], Elias Isaac Lattouf[5], Martin Feuerherd[5], Sutirtha Chattopadhyay[1], Nadine Kumpesa[2], Vera Griesser[2], Jean-Christophe Hoflack[2], Juliane Siebourg-Polster[2], Carolin Mogler[6], Leo Swadling[7], Laura J. Pallett[7], Philippa Meiser[1], Katrin Manske[1], Gustavo P. de Almeida[8], Anna D. Kosinska[9,10,11], Ioana Sandu[12], Annika Schneider[1], Vincent Steinbacher[1], Yan Teng[9], Julia Schnabel[13], Fabian Theis[14], Adam J. Gehring[15,16], Andre Boonstra[17], Harry L. A. Janssen[17,18], Michiel Vandenbosch[19], Eva Cuypers[19], Rupert Öllinger[20], Thomas Engleitner[20], Roland Rad[20], Katja Steiger[21], Annette Oxenius[12], Wan-Lin Lo[22], Victoria Klepsch[23], Gottfried Baier[23], Bernhard Holzmann[24], Mala K. Maini[6], Ron Heeren[19], Peter J. Murray[25], Robert Thimme[3], Carl Herrmann[4,5], Ulrike Protzer[8,9,10], Jan P. Böttcher[1], Dietmar Zehn[8], Dirk Wohlleber[5], Georg M. Lauer[5], Maike Hofmann[3], Souphalone Luangsay[2] & Percy A. Knolle[1,11,26] ✉

[1]Institute of Molecular Immunology, School of Medicine and Health, Technical University of Munich (TUM), Munich, Germany. [2]Roche Pharmaceutical Research and Early Development (pRED), Roche Innovation Center Basel, Basel, Switzerland. [3]Third Department of Medicine, University Hospital Freiburg, Freiburg, Germany. [4]Health Data Science Unit, Biomedical Genomics Group, Bioquant, Faculty of Medicine Heidelberg, Heidelberg, Germany. [5]Division of Gastroenterology, Massachusetts General Hospital and Harvard Medical School, Boston, MA, USA. [6]Institute of Pathology, School of Medicine and Health, TUM, Munich, Germany. [7]Division of Infection and Immunity, Institute of Immunity and Transplantation, University College London, London, UK. [8]Institute of Immunology and Animal Physiology, School of Life Science, TUM, Munich, Germany. [9]Institute of Virology, School of Medicine and Health, TUM, Munich, Germany. [10]Helmholtz Zentrum München, Munich, Germany. [11]German Center for Infection Research, Munich site, Munich, Germany. [12]Institute of Microbiology, ETH Zürich, Zürich, Switzerland. [13]Institute of Machine Learning and Biomedical Imaging, Helmholtz Zentrum Munich, Munich, Germany. [14]Institute of Computational Biology, TUM, Munich, Germany. [15]Toronto Centre for Liver Disease and Toronto General Hospital Research Institute, Toronto, Ontario, Canada. [16]Department of Immunology, University of Toronto, Toronto, Ontario, Canada. [17]Department of Gastroenterology and Hepatology, Erasmus University Medical Center, Rotterdam, The Netherlands. [18]Toronto General Hospital, University of Toronto, Toronto, Ontario, Canada. [19]Institute of Multimodal Imaging, University of Maastricht, Maastricht, The Netherlands. [20]Institute of Molecular Oncology and Functional Genomics, School of Medicine and Health, TUM, Munich, Germany. [21]Comparative Experimental Pathology, School of Medicine and Health, TUM, Munich, Germany. [22]Department of Pathology, University of Utah, Salt Lake City, UT, USA. [23]Institute of Cell Genetics, Medical University of Innsbruck, Innsbruck, Austria. [24]Department of Surgery, School of Medicine and Health, TUM, Munich, Germany. [25]Max Planck Institute of Biochemistry, Martinsried, Munich, Germany. [26]Institute of Molecular Immunology, School of Life Science, TUM, Munich, Germany. [27]These authors contributed equally: Miriam Bosch, Nina Kallin. ✉e-mail: percy.knolle@tum.de

## Methods

### Animals and viral infection models

Six-week-old C57BL6/J male mice were purchased from Janvier or Charles River. H-2K$^{bSIINFEKL}$-restricted OT-1 TCR-transgenic CD45.1$^+$ mice and H-2K$^{bMGLKFRQL}$-restricted Cor93 TCR-transgenic CD45.1$^+$ mice were purchased from Charles River and were bred under specific pathogen-free conditions at Animal Core Facility of the School of Medicine, TUM. Mice were housed with a 12 h light/12 h dark cycle. The temperature was set to 22 ± 2 °C, humidity to 55 ± 10% and checked daily. Guidelines of the Federation of Laboratory Animal Science Association were implemented for breeding and experiments. Experiments were approved by the District Government of Upper Bavaria (permission nos. ROB-55.2-2532.Vet_02-14-185, ROB-55.2-2532.Vet_02-16-55 and ROB-55.2-2532.Vet_02-18-100).

**T cell transfer into mice.** A total of $1 \times 10^2$ naive (CD44$^-$CD62L$^+$) H-2K$^{bSIINFEKL}$-restricted CD45.1$^+$ CD8 T cells or $1 \times 10^4$ naive (CD44$^-$CD62L$^+$) H-2K$^{bMGLKFRQL}$-restricted CD45.1$^+$ CD8 T were isolated from TCR-transgenic OT-1 or Cor93 mice, respectively, by untouched immunomagnetic separation from spleens with more than 95% purity. Naive T cells were directly injected intravenously in PBS 1 day before infection with recombinant adenoviruses.

**Generation of recombinant adenoviral vectors and transduction of hepatocytes in mice.** Hepatotropic recombinant adenoviruses were generated as described previously[19,26]. Two identical recombinant adenoviruses were generated expressing the cassette GOL, that is, genes for GFP, ovalbumin and luciferase, either under a minimal CMV promoter (resulting in acute resolved infection of hepatocytes) or under an hepatocyte-specific transthyretin promoter (TTR) (resulting in persistent hepatocyte infection) as described[19]. Recombinant adenoviruses for transduction of hepatocytes with replication-competent HBV were generated using a 1.3 overlength construct of HBV genome (genotype D) as previously reported[57]. Recombinant adenoviruses were amplified in HEK 293 cells and infectious titres were determined by in vitro infection assays. High-titre adenoviral stocks were aliquoted and kept at −80 °C before use. For in vivo transduction of hepatocytes, recombinant adenoviruses were dissolved in saline immediately after thawing and injected intravenously through the tail vein.

**In vivo bioluminescence imaging.** In vivo bioluminescence from the expression of luciferase after transduction with recombinant adenoviruses coding for the GOL expression cassette was quantified with the in vivo imaging system IVIS Lumina LT-Series III (PerkinElmer). Mice were anaesthetized using 2.5% isoflurane and received 100 mg kg$^{-1}$ of body weight D-luciferin-K-salt (PJK) as substrate for luciferase. Regions of interest were defined in the upper right quadrant of mice and photons detected in this region were quantified. System calibration of the IVIS Lumina LT III performed before every experiment assured comparability of results.

**Quantification of liver damage.** Serum alanine transaminase was measured from peripheral blood of mice using the Reflotron Plus system (Roche Diagnostics).

**Quantification of HBV replication in Ad–HBV and AAV–HBV-transduced hepatocytes.** HBeAg titres were determined in peripheral blood using an Architect platform and the HBeAg reagent kit (6C32-27) with HBeAg quantitative calibrators (7P24-01, Abbott Laboratories).

**Histology and immunohistochemistry.** Histology and immunohistochemistry of liver tissue sections were performed as described previously[19]. In brief, tissues were fixed in 4% formalin and paraffin embedded. For haematoxylin/eosin staining or immunohistochemistry, 2 μm sections were made. Haematoxylin/eosin staining was performed according to standard protocols. Tissue sections were stained with anti-HBc and anti-GFP (1.5 μg ml$^{-1}$ polyclonal anti-HBcAg (Origene); 0.4 ng μl$^{-1}$ of polyclonal anti-GFP (Fitzgerald)) with a Leica Biosystems Bond MAX (Leica) and binding was visualized with DAB (Dako) as a brown precipitate. Slides were scanned with an Aperio System and analysed with Aperio Image Scope v.12.4.0 software (Leica) and QuPath v.0.2.3 (ref. 58).

### Isolation and culture of primary mouse cells

**Splenocyte isolation.** Spleens were passed through a 100 μm cell strainer, red blood cells were lysed with ammonium-chloride-potassium lysing buffer for 2 min and splenocytes were used for further experiments.

**Isolation of liver-associated lymphocytes.** Before excision, livers were perfused with PBS through the portal vein. Liver tissue was passed through 100 μm mesh cell strainers and digested with 125 μg ml$^{-1}$ of collagenase type II (Worthington) in GBSS (PAN Biotech) for 10 min at 37 °C. For enrichment of liver-associated lymphocytes, a density gradient centrifugation with 40%/80% Percoll (GE Healthcare) was performed at 1,440g for 20 min.

**Isolation of primary mouse hepatocytes.** Livers were perfused with 0.12 U ml$^{-1}$ of collagenase (SERVA) at 6 ml min$^{-1}$ for 8 min through the portal vein. Livers were then removed, mechanically disrupted and passed through a 300 μm cell strainer. Liver cell suspensions were filtered through a 100 μm mesh and pelletized at 50g for 2 min. Hepatocytes were purified by density gradient centrifugation with 50%/80% Percoll (GE Healthcare) at 600g for 20 min. For cytotoxicity assays, 10,000 hepatocytes per well were seeded on 96-well E-plates (ACEA Biosciences) coated with 0.02% collagenR (SERVA). Cell attachment was achieved in supplemented William's E medium (PAN Biotech, 200 mM glutamine (Thermo Fisher Scientific), 1 M Hepes pH 7.4, 10$^4$ U ml$^{-1}$ of penicillin/streptomycin, 50 mg ml$^{-1}$ of gentamycin (Merck), 0.005 n ml$^{-1}$ of insulin (INSUMAN rapid, Sanofi), 1.6% DMSO (Merck) and 10% FBS (PAN Biotech). Attached cells were cultivated in supplemented William's E medium (as above) containing 1% FBS.

**Isolation of primary mouse liver sinusoidal endothelial cells.** Non-parenchymal liver cells were isolated from mouse livers after portal vein perfusion with collagenase in Gey's balanced salt solution, followed by in vitro digestion with collagenase in a rotatory water bath at 37 °C and density gradient centrifugation. LSECs were then obtained by immunomagnetic separation using anti-CD146 coated microbeads (Miltenyi), reaching a purity of 95% or more, as previously described[59–61]. To investigate the transfer of molecules from LSECs to T cells, LSECs were labelled with 10 μM CFSE (Invitrogen). LSECs were activated with 50 μg ml$^{-1}$ of IFNγ (Miltenyi) to increase adhesion before coculture experiments. To analyse cAMP signalling, LSECs were treated with 1 μM Celecoxib (Cayman Chemical) before coculture.

**Ex vivo treatments of CD8 T cells.** CD8 T cells were isolated from the liver or spleen and cultured in RPMI 1640 medium (GIBCO) supplemented with 10% FCS, 1% L-glutamine (200 mM), 1% penicillin/streptomycin (5,000 U ml$^{-1}$), 50 μM 2-mercaptoethanol. For ex vivo stimulation and intracellular cytokine staining, cells were stimulated with 10 nM recombinant SIINFEKL peptide (peptides&elephants), HB$_{core}$ peptide MGLKFRQL (peptides&elephants) or 1× eBioscience cell stimulation cocktail (Thermo Fisher Scientific) with 3 μg ml$^{-1}$ of Brefeldin A (Invitrogen). To analyse cAMP signalling, T cells were incubated for 1 h with the adenylyl cyclase agonist Fsk (25 μM, Sigma-Aldrich), the PKA agonist Sp-8br-cAMPS (250 μM, Cayman Chemical), the EPAC agonist 8-pCPT-2′-O-Me-cAMP (30 μM, Tocris) or the adenosine A2A receptor agonist CGS21680 (100 nM, Tocris) solved in DMSO (Sigma-Aldrich).

CD8 T cells were cocultured at a 1:1 ratio with primary mouse LSECs or dendritic cells and were then separated from the antigen-presenting cells and directly analysed or restimulated with cognate peptide for 18 h. To analyse cAMP signalling in cocultures, T cells were pre-treated with the EPAC inhibitor ESI-09 (10 µM, Tocris), PKA antagonist Rp-8-bromo-cAMPS (1 mM, Cayman Chemical), adenylyl cyclase antagonist MDL-12330A (100 µM, Tocris), A2AR antagonist SCH58261 (100 nM, Tocris) and PTPN22 inhibitor PTPN22-IN-1 (1.4 µM, MedChemExpress).

## Isolation and culture of patient-derived cells and assessment of clinical parameters
**Clinical diagnostics.** Blood samples of participants with viral hepatitis were recruited at the Department of Medicine II of the University Hospital Freiburg, Germany, and at the Department of Gastroenterology, Hepatology and Endocrinology. Peripheral blood and liver fine-needle aspirations were collected from participants living with chronic hepatitis B at the Erasmus MC University Medical Center (Rotterdam, The Netherlands), the Toronto General Hospital (Toronto, Canada) and the Massachusetts General Hospital (Boston, United States). All participants provided written informed consent. This study was approved by institutional review boards at all three sites and was conducted in accordance with the declarations of Helsinki and Istanbul. Individuals were classified into different clinical phases of chronic or resolved HBV infection according to the European Association for the Study of the Liver guideline of 2017, which considers the presence of HBeAg, HBV DNA concentrations, transaminase concentrations (alanine transaminase and aspartate transaminase) and the presence or absence of liver inflammation[62]. HBeAg, serum HBV DNA and aspartate transaminase/alanine transaminase values were determined as part of the clinical diagnostics at the University Hospital Freiburg, Germany. Confirmation of HLA-A*02:01 was performed by HLA-typing by next-generation sequencing on a MiSeq system using commercially available primers (GenDx). Written informed consent was obtained from all participants before blood donation. The study was conducted according to federal guidelines, local ethics committee regulations of Albert-Ludwigs-Universität, Freiburg, Germany (no. 474/14) and the Declaration of Helsinki (1975).

**Peripheral blood mononuclear cell isolation from patients.** Venous blood samples were collected in EDTA-coated tubes. Peripheral blood mononuclear cells were isolated by density gradient centrifugation using lymphocyte separation medium (PAN Biotech). Isolated peripheral blood mononuclear cells were resuspended in RPMI 1640 medium supplemented with 10% FCS, 1% penicillin/streptomycin and 1.5% 1 M HEPES buffer (Thermo Fisher) and stored at −80 °C until used. Frozen peripheral blood mononuclear cells were thawed in complete medium (RPMI 1640 supplemented with 10% FCS, 1% penicillin/streptomycin and 1.5% 1 M HEPES buffer (ThermoFisher) containing 50 U ml$^{-1}$ of benzonase (Sigma).

**Magnetic bead-based enrichment of** $HBV_{core_{18}}$ **CD8 T cells from patients.** A total of $1 \times 10^7$ to $2 \times 10^7$ peripheral blood mononuclear cells were incubated for 30 min with PE-coupled peptide-loaded HLA class I multimers. Enrichment was then performed with anti-PE beads using magnetic-activated cell sorting technology (Miltenyi) according to the manufacturer's instructions. Enriched $HBV_{core_{18}}$-specific CD8 T cells were subsequently used for transcriptome analysis.

## Antibodies and multimers used for cell characterization by flow cytometry
Cell staining for flow cytometry was performed at 4 °C for 30 min. The following antibodies (clones, dilution, catalogue number) were used for staining of mouse cells: anti-CD8 (53-6.7, 1:250, 100752), anti-CD45.1 (A20, 1:200,110722, 110704 and 110748), anti-CXCR6 (SA051D1, 1:200, 151117, 151104, 151108, 151109 and 151115), anti-CX3CR1 (SA011F11, 1:200,149016, 149004 and 149006), anti-CD44 (IM7, 1:200, 103036), anti-CD69 (H1.2F3, 1:100, 104503), anti-TIM-3 (B8.2C12, 1:200, 134008), anti-TIGIT (1G9, 1:200, 142111), anti-IFNγ (XMG1.2, 1:200, 505808), anti-CD19 (1D3, 1:200, 152404), anti-CD335 (29A1.4, 1:200, 137606), anti-Lck pY394 (A18002D, 1:100, 933104), CD39 (Duha59, 1:200, 143805), anti-CD45.2 (104, 1:200, 109805), anti-CD3 (17A2, 1:200, 100217), anti-NK1.1 (PK136, 1:100, 108747), anti-CD4 (GK1.5, 1:200, 100449), anti-CD49a (HMa1, 1:200, 142606), all Biolegend, and anti-CD69 (H1.2F3, 1:100, 63-069-82), anti-PD-1 (J43, 1:200, 46-9985-82), anti-LAG-3 (eBioC9B7W, 1:200, 406-2239-42 and 12-2231-82), anti-TIM-3 (B8.2C12, 1:200, 12-2231-82) anti-TOX (TXRX10, 1:100, 12-6502-82), anti-granzyme B (GB11, 1:200, GRB04 and GRB05), anti-TNF (MP6-XT22, 1:200, 25-7321-82), anti-4-1BB (17B5, 1:100, 48-1371-82), anti-CD25 (PC61.5, 1:200, 48-0251-82), anti-Akt pS473 (SDRNR, 1:100, 25-9715-42), CD73 (TY/11.8, 1:200, 48-0731-82) (all Thermo Fisher Scientific) and anti-pPKA (47/PKA; BD Biosciences, 1:5,560205). MHC class I H-2K$^{bSIINFEKL}$-restricted or H-2K$^{bMGLKFRQL}$-restricted streptamers[63] were provided by D. Busch (Institute of Microbiology, TUM). For labelling of antigen-specific CD8 T cells, 0.4 µg of peptide-loaded streptamer per sample was incubated with 0.4 µl of Strep-Tactin-PE/APC (IBA Lifesciences) in PBS for 30 min on ice before incubation with cell suspensions. To exclude dead cells, fixable viability dye eFluor780 (Invitrogen) was included in the staining panels. For intracellular staining of cytokines, intracellular fixation buffer (Invitrogen) was used according to the manufacturer's instructions. Staining of GzmB and TOX was performed in combination with Foxp3/transcription factor staining buffer set (Thermo Fisher Scientific) according to the manufacturer's instructions. For staining of pPKA, cells were fixed in IC fixation buffer (Invitrogen) for 30 min and permeabilized with ice-cold methanol for 30 min before staining.

For staining of human cells, the following antibodies (clones, dilution, catalogue number, lot number) were used: anti-CD14 (61D3, 1:100, A15453, 2406638,), anti-CD19 (HIB19, 1:100, 17-0199-42, 2472560) (all eBioscience), anti-CD45RA (HI100, 1:200, 304178, 2327528), anti-CCR7 (G043H7, 1:20, 353244, B347205) (all Biolegend), anti-CD8 (RPA-T8, 1:200, 563795, 9346411) and anti-GZMB (GB11, 1:100, 563388, 3317967) (all BD Bioscience). Fixable viability dye eFluor 780 (65-086-14, eBioscience) was used for live/dead discrimination. HLA class I epitope-specific tetramers were generated through conjugation of biotinylated peptide/HLA class I monomers with PE-conjugated streptavidin (ProZyme) at a peptide/HLA I:streptavidin molar ratio of 5:1. Of note, targeted epitopes of $HB_{core}$-specific CD8 T cells were previously analysed for viral sequence mutations. T cell responses of patients harbouring viral sequence mutations in the targeted epitope were excluded. HLA-A*02:01/HBV$_{core_{18}}$, FLPSDFFPSV peptide was synthesized with standard Fmoc chemistry and a purity of more than 70% (Genaxxon).

## Flow cytometry and cell sorting
Multicolour flow cytometry data were acquired on a Sony SP6800 spectral analyser (Sony Biotechnology) or a CytoFLEX S (Beckman Coulter). Cells were sorted with a Sony SH800 (Sony Biotechnology) or a MoFlo Astrios EQ (Beckman Coulter). Flow cytometry data were analysed with FlowJo software v.10.7.1 and v.10.8.0 (BD Biosciences), GraphPad Prism v.10.0.3 (Graphpad Software), R v.4.0.2 and R cytofkit GUI v.0.99.

## Real-time impedance-based cytotoxicity assay
Ex vivo cytotoxicity assays were performed with timelapse xCELLigence-based cell impedance measurement. Primary murine hepatocytes were used as target cells and seeded on a collagenR-coated 96-well E-plate. Sorted CD8 T cells were added to peptide-pulsed or mock-treated primary mouse hepatocytes 24 h after isolation and cell impedance quantified as cell index was recorded with an

xCELLigence RTCA MP instrument (ACEA Biosciences) as a measure of antigen-specific CD8 T cell cytotoxicity.

## Confocal immunofluorescence imaging of liver tissue

Livers were perfused with 2.5 ml of Antigenfix solution (Diapath) through the portal vein, excised and fixed for 4 h in 1 ml of Antigenfix. Fixed liver lobes were embedded in Tissue-Tek O.C.T. (Sakura Finetek) and frozen at −80 °C, from which 50 μm cryosections were cut with a cryotome (Leica). Liver sections were permeabilized and blocked with 0.1 M Tris (AppliChem) containing 1% BSA, 0.3% Triton X-100 (Gebru Biotechnik), 1% normal mouse serum (Sigma) for 2 h or more. Sections were stained in blocking buffer with anti-CD3 (clone 17A2, 100240, 1:200, Biolegend), anti-CD45.1 (clone A20, 110732, 1:200, Biolegend), anti-CD146 (clone ME-9F1, 130-102-846, 1:100, Miltenyi) and Phalloidin DyLight 488 (21833, 1:100, Thermo Fisher Scientific) or anti-CD3 (clone 17A2, 100240, 1:200, Biolegend), anti-CD45.1 (clone A20, 110732, 1:200, Biolegend), anti-I-A/I-E (MHC class II) (clone M5/114.15.2, 107622, 1:200, Biolegend) and anti-CD103 (goat polyclonal, AF1990, 1:200, R&D Systems) followed by anti-goat IgG (donkey polyclonal, 705-625-147, 1:500, Jackson ImmunoResearch). Tissue sections were mounted with Mowiol and imaged using an inverted TCS SP8 confocal microscope (Leica). Images were analysed with Imaris 9.6 software (Bitplane).

## Human liver immunohistochemistry

Human liver samples (formalin-fixed, paraffin embedded, $n = 21$; ethical approval: 518/19 S) were double-stained by RNAscope (CXCR6) and CD3 (MRQ39, 1:1,500). Briefly, after deparaffinization and standard pretreatment, slides were incubated with RNA probes for CXCR6 (468468, ACD, Bio-Techne), detected with a RNAscope 2.5 Leica Assay-brown (Leica Biosystems) followed by incubation with a primary antibody against CD3 (103R-95, CellMarque) and detection with a Bond Polymer Refine Red Detection Kit (Leica Biosystems) on a Bond Rxm system (Leica Biosystems). All slides were counterstained with haematoxylin, cover slipped and digitalized using an AT2 scanner (Leica Biosystems). The study was conducted according to federal guidelines, local ethics committee regulations of the Technical University of Munich, Germany (no. 518/19 S-SR)

## RNA sequencing, bioinformatic and pathway analysis

**Sample preparation for RNA-seq of OVA$_{257-264}$-specific CD45.1$^+$ CD8 T cells.** Liver-associated lymphocytes and splenocytes from mice with resolved Ad−CMV−GOL infection were sorted into CD45.1$^+$CXCR6$^+$CX$_3$CR1$^-$ CD8 and CD45.1$^+$CXCR6$^-$CX$_3$CR1$^+$CD8 T cells. CD8 T cells derived from mice with persistent Ad−TTR−GOL infection were sorted into CXCR6$^+$CX$_3$CR1$^-$CD45.1$^+$ CD8 and CXCR6$^+$CX$_3$CR1$^+$CD45.1$^+$ CD8 populations. A total of 5,000 cells per sample were collected in 1× TCL lysis buffer (Qiagen) supplemented with 1% (v/v) 2-mercaptoethanol and immediately frozen on dry ice.

Library construction for bulk 3′-sequencing of poly(A)-RNA was performed as described previously[64]. In brief, each sample was produced with a Maxima RT polymerase (Thermo Fisher) with barcoded complementary DNA. Unique molecular identifiers (UMIs) and template switch oligo (TSO) were used to elongate adaptor 5′ ends of the cDNAs. All samples were united and full-length cDNA was amplified with primers. The cDNA was complemented with the Nextera XT kit (Illumina) and 3′-end-fragments and supplemented with P5 and P7 Illumina overhangs. Library was sequenced using NextSeq 500 (Illumina). The UMI tables were spawned for samples and genes using Drop-seq pipeline (https://github.com/broadinstitute/Drop-seq). We annotated the reads using GRCm38 reference genome ENSEMBL annotation release 75. We used DESeq2 R package v.2.1.28.1 (ref. 65) to extract the DEGs ($\log_2$ fold-change 1 and $P_{adj} \leq 0.05$). DEGs were visualized as volcano plot using ggplot2 R package v.3.3.2. Principal component analysis was executed using prcomp R function (in stats R package

v.3.6.1) and pictured using ggplot 2 and ggrepel R v.0.9.4 packages. See Figs. 1 and 4 and Extended Data Fig. 2.

**Sample preparation for RNA-seq of P14 LCMV-specific CD8 T cells.** P14 cells were adoptively transferred into C57BL/6 mice and infected one day later with either LCMV clone 13 or LCMV Armstrong. Resident (CD69$^+$CD101$^+$CXCR6$^+$CX3CR1$^-$) and effector/effector-memory (CX$_3$CR1$^+$) P14 cells from the liver were sorted at 27 d.p.i. Total RNA was isolated using the RNAdvance Cell v.2 kit (Beckmann-Coulter). Quality and quantity of isolated RNA was analysed with the Bioanalyzer RNA Pico Chip (Agilent). The cDNA synthesis was performed with the Smart-Seq v.4 Ultra Low Input RNA kit (Takara) following the manufacturer's protocol with 12 cycles of PCR amplification. Input amount was 1 ng of each RNA sample. The cDNA was measured with Bioanalyzer DNA HS Chip (Agilent) and 300 pg of amplified cDNA were used for library preparation with the Nextera XP DNA Library Preparation Kit (Illumina). Libraries were analysed with a Bioanalyzer DNA HS Chip (Agilent) and quantified by quantitative PCR following guidelines from Illumina and using Kapa SYBR master mix (Kapa Biosystems). After the normalization of all libraries to 2 nM, 13 samples each were pooled and sequenced on two single-end runs (1× 100 base pairs, dual-index) on a HiSeq2500 (Illumina) using HiSeq Rapid v.2 chemistry (Illumina). See Extended Data Fig. 2.

**Sample preparation for RNA-seq of core$_{93-100}$-specific CD45.1$^+$ CD8 T cells.** Liver-associated lymphocytes and splenocytes from mice with Ad−HBV infection were pregated on (CD19/Ly6G/TER119/CD335)$^-$ CD8 T cells and sorted into liver CXCR6$^+$CD45.1$^+$, liver CD45.1$^+$CX$_3$CR1$^+$CD8 T cells, spleen CD45.1$^+$CX$_3$CR1$^+$ CD8 T cells and liver CD45.1$^-$ CD8 T cells from resolved infections and liver CD45.1$^+$CXCR6$^+$ and liver CD45.1$^+$CXCR6$^+$CX$_3$CR1$^+$CD8 T cells and liver CD45.1$^-$ CD8 T cells from persistent infection. A total 100 CD8 T cells were directly sorted into 96-well plates prepared with 1× reaction buffer consisting of lysis buffer and RNase inhibitor for low input RNA-seq (Takara). Plates were spun down and immediately stored on dry ice or at −80 °C until further processing. Sample plates containing lysed T cells were subjected to cDNA library preparation using the Smart-Seq v.4 Ultra Low Input RNA Kit (Takara) followed by sequencing library preparation using the Nextera XT DNA Library Preparation Kit (Illumina) as per manufacturer's instructions with minor modifications. Briefly, full-length cDNA was generated by reverse transcription, template-switching reaction and PCR pre-amplification of polyadenylated mRNA as previously described[66]. The cDNA libraries were quantified using the Qubit dsDNA High Sensitivity Kit and quality was assessed on a bioanalyser using DNA high-sensitivity chips (Agilent). Double-stranded cDNA was subjected to fragmentation and PCR-based addition of Illumina barcoded sequencing adaptors at both fragment ends. Sequencing library quantity and quality was assessed as described above. The 50× cycles paired-end sequencing was performed on a NovaSeq 6000 instrument (Illumina) at a targeted read depth of 25 M per sample. See Fig. 2 and Extended Data Fig. 3.

**Sample preparation for scRNA-seq of human HB$_{core}$-specific CD8 T cells.** HBV$_{core18}$-specific CD8 T cells were enriched by magnetic bead-based sorting and surface staining was performed. In total, 1,152 live HBV$_{core_{18}}$-specific CD8 T cells were sorted in 384-well plates (Bio-Rad) containing lysis buffer and mineral oil using FACS Melody Cell Sorter in single-cell sorting mode. Naive CD45RA$^+$CCR7$^+$ T cells were excluded. After the sorting, the plates were centrifuged for 1 min at 2,200$g$ at 4 °C, snap-frozen in liquid nitrogen and stored at −80 °C until processed. The scRNA-seq was performed using the mCEL-Seq2 protocol, an automated and miniaturized version of CEL-Seq2 on a mosquito nanolitre-scale liquid-handling robot (TTP LabTech)[67,68]. Twenty-two libraries with 96 cells each were sequenced per lane on an Illumina HiSeq 3000 sequencing system (pair-end multiplexing run)

at a depth of about 130,000–200,000 reads per cell. Sequencing was performed at the sequencing facility of the Max Planck Institute of Immunobiology and Epigenetics (Freiburg, Germany). See Fig. 3 and Extended Data Fig. 4.

**scRNA-seq of human HBV-specific CD8 T cells isolated from the liver by fine-needle aspiration.** We analysed HBV-specific CD8 T cells from 23 cryopreserved fine-needle liver aspirates (three patients with HBV hepatitis, eight patients with HBe⁻HBV infection and ten patients with HBV functional cure). Cells were thawed and stained with lineage marker antibodies as well as HBV multimers for two distinct HBV-specificities. The live HBV-specific CD8 T cells were sorted in 96-well Armadillo plates (Thermo Fisher Scientific) containing RNA lysis buffer using a BD SORP FACS Aria in index single-cell sorting mode. After sorting, plates were centrifuged and snap-frozen on dry ice. The scRNA-seq was performed at the Broad Institute walk-up sequencing facility (Cambridge, MA, United States) using the Smart-Seq2 protocol and Illumina Nextseq500. After quality control, 977 HBV-specific cells from the 21 liver samples could be analysed using R v.4.1.2 with the Seurat package v.4.3.0. Raw counts were normalized and scaled using the Seurat v.4.3.0 Normalize-Data and ScaleData functions, respectively, by dividing feature counts in each cell by the total counts of the cell, applying natural-log transformation to the result using log1p and scaling and centring expression levels for every gene. Subsequently, the HBV-specific cells in each sample were categorized on the basis of whether they did or did not exhibit CXCR6 expression. Gene expression levels were averaged per outcome group for each gene of the CREM signature according to CXCR6 status, followed by visualization in a heatmap. Liver fine-needle aspirations were collected from participants living with chronic hepatitis B at the Erasmus MC University Medical Center (Rotterdam, The Netherlands), the Toronto General Hospital (Toronto, Canada) and the Massachusetts General Hospital (Boston, United States). All participants provided written informed consent. This study was approved by institutional review boards at all three sites and was conducted in accordance with the declarations of Helsinki and Istanbul. See Fig. 3e.

**Gene set enrichment and pathway analyses.** We performed GSEA on gut, skin and lung tissue-resident memory T cell dataset[69] as follows: first, we downloaded raw microarray data pertaining from the GEO database (accession ID: GSE47045, tissue-resident memory T cells: gut, lung and skin versus tissue effector-memory cells (spleen)) and extracted DEGs from each comparison using Limma R package v.3.58.1 (ref. 70). We used GSEA v.4.0.3 to perform enrichment analysis using DEGs which were ordered according to log₂-fold-changes delivered by DESeq2 v.2.1.28.1. We also performed core signature analysis using GSEA scores as follows. Initially, we extracted genes which contribute to core enrichment from the tissue-residency signature. The gene set associated with Hobit and Blimp was obtained from ref. 71 (GEO accession ID: GSE70813) and the raw dataset was processed using GREIN DB v.1 (ref. 72). DEGs were determined using the DESeq2 v.2.1.28.1 R package[65]. The gene set related to TCR signalling was obtained from the MsigDB BIOCARTA dataset (https://www.gsea-msigdb.org/gsea/msigdb/). We retrieved the calcium signalling pathway genes from the Molecular Genome Informatics database (http://www.informatics.jax.org/go/term/GO:0019722). Gene sets from *Hobit*-deficient, *Blimp1*-deficient cells were matched to DEGs from hepatic CXCR6⁺ CD8 T cells versus spleen CX₃CR1⁺ CD8 T cells. The gene sets dependent on CREM were obtained from ref. 73. To create the gene set for cAMP signalling, gene symbols for all genes encoding adenylyl cylases, phosphodiesterases, PKA regulatory and catalytic subunits, kinase anchoring proteins, EZRIN, EPCA1, EPAC2 and small GTPases were downloaded from the human gene database GeneCards. The PreRanked tool from GSEA v.4.0.3 (ref. 74), was used to evaluate the normalized enrichment score and FDR ($q < 0.25$) was used to measure the statistical significance of normalized enrichment score. See Figs. 1–4 and Extended Data Figs. 2 and 4.

**Identification of transcription factors and network analysis .** We performed transcription factor network analysis using DEGs CXCR6⁺ CD8 T cells from livers after Ad–CMV–GOL versus CXCR6⁺ CD8 T cells from livers during Ad–TTR–GOL infection. Transcription factors regulated in the transcriptomes were extracted using the transcription factor checkpoint database[75]. Through this analysis we mined seven and two transcription factors from the transcriptome datasets. We evaluated transcription factor–transcription factor network: (1) promoter sequences (−1 kilobases (kb)) of significantly regulated DEGs were downloaded from Eukaryotic promoter database and UCSC (GRCm38/mm10) https://genome.ucsc.edu/cgi-bin/hgTrackUi?db=mm10&c=chrX&g=encode3RenEnhancerEpdNewPromoter and ref. 76; (2) we extracted the transcription factor binding sites from the JASPAR core and HOCOMOCO databases[77,78]; (3) finally, scanned promoter sequences (−1 kb promoters) of DEGs and transcription factors for binding sites using the custom Python v.3.12 script (https://zenodo.org/records/11040043). Transcription factor networks were generated and visualized in Cytoscape v.3.7.1 (ref. 79). To evaluate the hierarchy of transcription factor networks, in ($I$) and out degrees ($O$) were computed for each transcription factor and their targets using the igraph R package v.2.0.2 (https://igraph.org/) and hierarchy height ($H$). $H = (O − I)/(O + I)$ was calculated as explained previously[80]. Hierarchy height score defined three and two levels of transcription factor–transcription factor network. See Fig. 1.

**Analysis of RNA-seq data from P14 LCMV-specific CD8 T cells.** Demultiplexing was done with the bcl2fastq software v.2.20.0.422. Reads were processed using snakemake pipelines[81] as described at https://gitlab.lrz.de/ImmunoPhysio/bulkSeqPipe. Reads were filtered using Trimmomatic v.0.36 (ref. 82). STAR v.2.5.3a (ref. 83) was used for mapping to annotation release no. 91 and genome build no. 38 from *Mus musculus* (Ensembl GRCm38). Multimapped reads were discarded. Read counting was performed using htseq v.0.9.1 (ref. 84) and DESeq2 v.1.24.0 (ref. 65) was used for differential expression analysis. Genes showing total counts of less than 10 were discarded. Differences were considered significant when absolute log₂ fold-change greater than 1 and $P_{adj} < 0.05$. See Extended Data Fig. 2.

**Analysis of scRNA-seq data from human HB_core-specific CD8 T cells.** For data preprocessing, Fastq files were mapped to the human genome (v.GRCh38), annotated, demultiplexed and counted using the scPipe R package workflow v.1.12.0, R v.3.5.0. Cells with less than 150 UMI counts were filtered out. Cells were clustered using the Louvain method, UMAP projection and DEA were carried out using Seurat v.3.2.0. We scored the cells using the AddModuleScore function from Seurat v.3.2.0 with nbin = 5. For the human CD8 T cells blood signatures we used the signatures from ref. 85. We removed signatures with less than ten genes and additionally clusters 11–13, which corresponds to marginal clusters in the Galletti study[85]. Transcription factor activity levels were calculated using the pySCENIC pipeline (v.0.10.10). We selected 10 kb around the gene TSS for motif search. For analysing the CREM signature in circulating human HB_core-specific CD8 T cells, we performed unsupervised clustering of scRNA-seq data by calculating the principal components using the RunPCA function in the Seurat R package v.3.2.0. Next, we integrated four patient datasets using Harmony v.1.2.0. We identified clustering resolution (0.6) using the clustree R package v.0.4.0 (ref. 86). Finally, we analysed the CREM signature using the UCELL R package v.1.2.4 (ref. 87). See Fig. 3 and Extended Data Fig. 4.

**Generation of conditional *Icer*-deficient mice**
The genomic region encompassing the ICER-specific exon as well as the alternative promoters driving expression of ICER and smICER, respectively, was flanked by *loxP* sites using homologous recombination in mouse ES cells (Extended Data Fig. 8a). The neomycin-resistance cassette was flanked by FRT sites and removed by intercross with

*Flp*-deleter mice, thereby generating the *Icer<sup>fl</sup>* allele (B). An *Icer*<sup>null</sup> allele is generated by Cre-mediated recombination (Extended Data Fig. 8b). Mice bearing the *Icer<sup>fl</sup>* allele were backcrossed to the C57BL/6 background for more than ten generations. For specific deletion of ICER in T cells, *Icer<sup>fl/fl</sup>* mice were intercrossed with *Cd4<sup>cre</sup>* mice. Mice were maintained in a specific pathogen-free facility.

## Reporting summary

Further information on research design is available in the Nature Portfolio Reporting Summary linked to this article.

## Data availability

RNA-seq data for mouse HB$_{core}$-specific CD8 T cells are deposited in the Gene Expression Omnibus (GEO) at accessions GSE214151 and GSE233661. RNA-seq data for mouse ovalbumin-specific CD8 T cells are deposited at GSE168096. RNA-seq data for mouse LCMV-specific CD8 T cells are deposited at GSE212925. RNA-seq data for human HBV-specific CD8 T cells are available at Figshare (https://figshare.com/s/245d38cb7c4901b70b3f (ref. 88) and https://figshare.com/s/0198184966164a2aabf4 (ref. 89)). All high-content data shown in this manuscript are deposited at publicly available databases (Extended Data). Source data are provided with this paper.

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

**Acknowledgements** We thank the LiverConsortium, supported by Janssen Pharma, for providing data on intrahepatic CD8 T cells isolated from patients with chronic hepatitis B. We thank the tissue bank from the Institute of Pathology, School of Medicine and Health at TUM for contributing human liver tissue samples. This study was supported by grants from the Deutsche Forschungsgemeinschaft (DFG) (CRC-TRR179 to P.A.K., U.P., M.H., R.T., C.H. and D.W.; CRC1160 to M.H.; CRC1371, CRC1054, ZE 832/6-1 and ZE 832/8-1 to D.Z.; DFG grants 424926990 and 442405234 to J.P.B.; and DFG grants INST 95/1763-1, 95/1738-1 and 95-1651-1 to P.A.K.), by grants from the German Center for Infection Research Munich site (P.A.K., U.P. and D.W.) and by the EU (TherVacB to U.P. and P.A.K.) and ERC AG to GB 786462. M.H. was supported by the DFG Heisenberg programme, A.O. was supported by the Swiss National Science Foundation (SNSF), grant 310030_146140 and P.J.M. was supported by FOR2599 and CRC-TRR127.

**Author contributions** M.B., N. Kallin, H.W., S.H., K.H., A.F., N. Kumpesa, V.G., J.-C.H., L.S., L.J.P., P.M., K.M., A.K., A.S., V.S., Y.T., M.V., E.C., R.Ö., T.E., K.S., A.D.K. and J.S.-P. performed the experiments. A.J.G., A.B., H.L.A.J., B.H., G.M.L., M.H. and R.T. provided essential reagents or data for analysis. S.D., J.D.Z., C.R., E.I.L., M.F., S.C., C.M., G.P.d.A., I.S., J.S., F.T., R.R., A.O., W.-L.L., V.K., G.B., M.K.M., R.H., P.J.M., R.T., C.H., U.P., J.P.B., D.Z., D.W., G.M.L., S.L. and P.A.K. analysed data. M.B., S.D., J.D.Z., P.J.M., M.K.M., R.T., G.M.L., S.L., U.P. and P.A.K. wrote the manuscript.

**Funding** Open access funding provided by Technische Universität München.

**Competing interests** The authors declare no competing interests.

**Additional information**
**Correspondence and requests for materials** should be addressed to Percy A. Knolle.

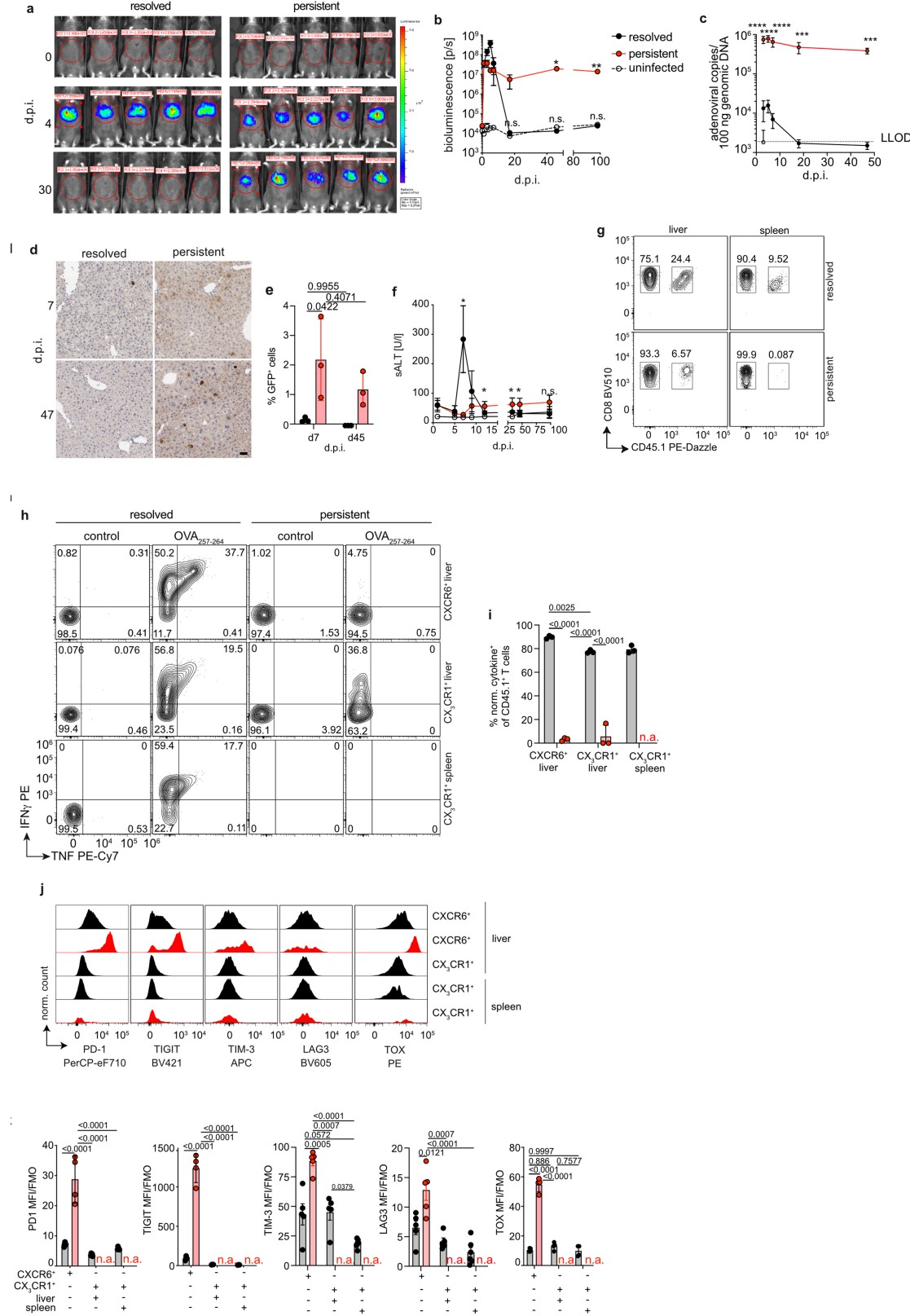

**Extended Data Fig. 1** | See next page for caption.

**Extended Data Fig. 1 | Kinetics of infection and frequencies of antigen-specific CD8 T cells during hepatotropic adenoviral infection. a,b**, In vivo bioluminescence imaging kinetic after hepatotropic infection and quantification (two-way ANOVA with Tukey's multiple comparisons; d0: resolved vs. chronic Padj=0.9295, resolved vs. uninfected Padj=0.0959, chronic vs. uninfected Padj=0.1498; d1: resolved vs. chronic Padj=0.9902, resolved vs. uninfected Padj=0.0209, chronic vs. uninfected Padj=0.1021; d3: resolved vs. chronic Padj=0.2179, resolved vs. uninfected Padj=0.1025, chronic vs. uninfected Padj=0.1269; d5: resolved vs. chronic Padj=0.1413, resolved vs. uninfected Padj=0.1270, chronic vs. uninfected Padj=0.0168; d7: resolved vs. chronic: Padj=0.8271, resolved vs. uninfected Padj=0.5733, chronic vs. uninfected Padj=0.1293; d17: resolved vs. chronic Padj=0.3965, resolved vs. uninfected Padj=0.6564, chronic vs. uninfected Padj=0.3961; d46: resolved vs. chronic Padj=0.0218, resolved vs. uninfected Padj=0.0766, chronic vs. uninfected Padj=0.0219; d98: resolved vs. chronic Padj=0.0087, resolved vs. uninfected Padj=0.8296, chronic vs. uninfected Padj=0.0087; n = 5). **c**, Quantification of adenoviral copies in liver tissue (two-way ANOVA with Sidak's multiple comparison, resolved vs. chronic Padj<0.0001 for all timepoints, n = 4). **d,e**, Liver immunohistochemistry detecting GFP-expressing virus-infected hepatocytes in brown (scale bar 50 μm) and quantification (two-way ANOVA with Tukey's multiple comparisons for Padj, n = 3). **f**, Time kinetics of sALT (two-way ANOVA with Tukey's multiple comparison, d0: uninfected vs.

resolved Padj=0.0560, uninfected vs. chronic Padj=0.1210, resolved vs. chronic Padj=0.9971; d5: uninfected vs. resolved Padj=0.5088, uninfected vs. chronic Padj=0.0265, resolved vs. chronic Padj=0.6827: d7: uninfected vs. resolved Padj=0.0981, uninfected vs. chronic Padj=0.3799, resolved vs. chronic Padj=0.0163; d9: uninfected vs. resolved Padj=0.3044, uninfected vs. chronic Padj=0.1871, resolved vs. chronic Padj=0.1963; d12: uninfected vs. resolved Padj=0.0788, uninfected vs. chronic Padj=0.0442, resolved vs. chronic Padj=0.1289; d33: uninfected vs. resolved Padj=0.2294, uninfected vs. chronic Padj=0.1477, resolved vs. chronic Padj=0.0481; d46: uninfected vs. resolved Padj=0.9976, uninfected vs. chronic Padj=0.0102, resolved vs. chronic Padj=0.0809; d98: uninfected vs. resolved Padj=0.9274, uninfected vs. chronic Padj=0.2301, resolved vs. chronic Padj=0.1714; n = 5). **g**, Gating strategy for antigen-specific CD45.1[+] T cells at d45 p.i. after adoptive transfer of 100 naive CD8 T cells on d-1 (n = 4). **h,i**, IFNγ and TNF expression by liver CD45.1[+]CD8 T cells at d45 p.i. after ex vivo re-stimulation with $OVA_{257-264}$-peptide and quantification (two-way ANOVA with Sidak's and Tukey's multiple comparison, n = 4). **j,k**, PD-1, TIGIT, TIM-3, LAG3 and TOX expression by liver and spleen CD45.1[+]CD8 T cells at d45 p.i. and quantification (one-way ANOVA with Tukey's multiple comparison, n ≥ 4) One out of ≥ two independent experiments shown; LLOD = lower limit of detection; not significant (n.s.) p≥0.05, *p < 0.05, **p < 0.01, ***p < 0.001, ****p < 0.0001, errors shown as s.d.; FMO = fluorescence minus one, MFI = geometric mean fluorescence intensity, p.i. = post infection.

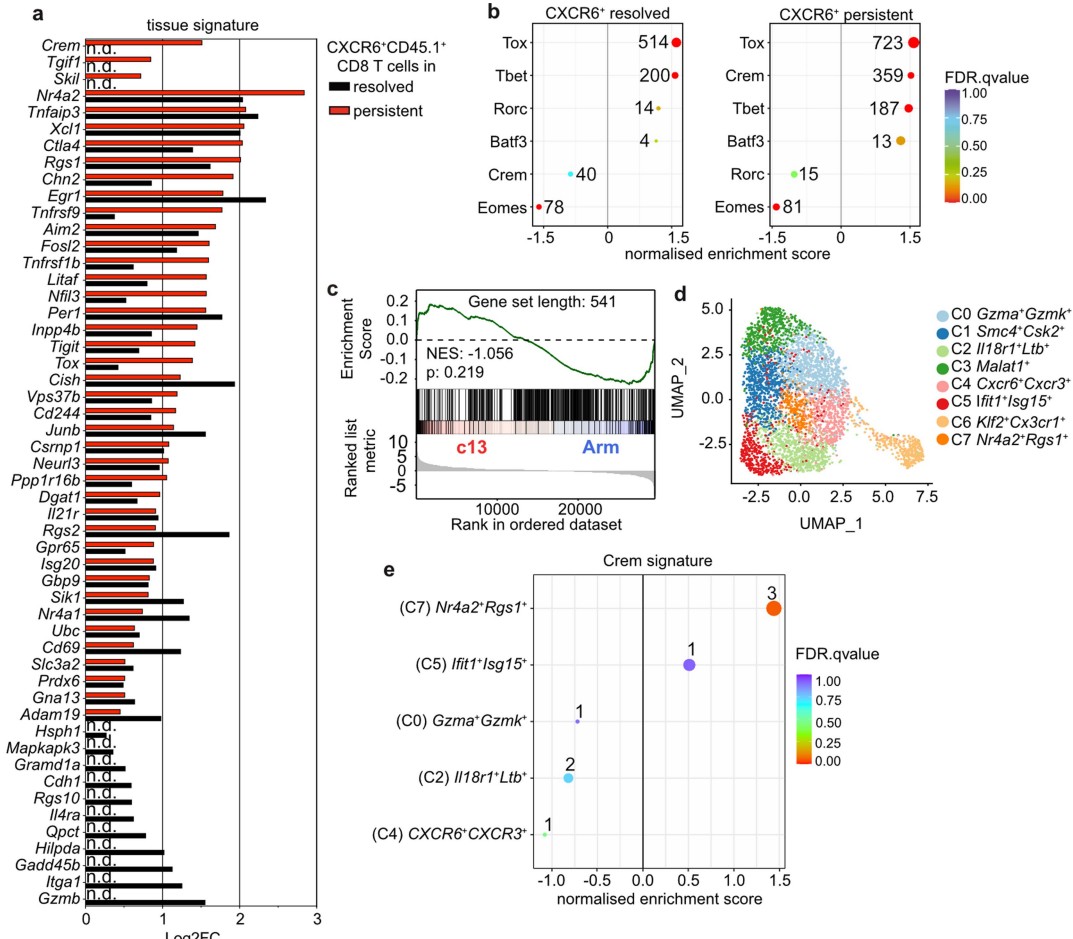

**Extended Data Fig. 2 | Transcriptional regulation of antigen-specific CD8 T cells after resolved and during persistent hepatotropic viral infection and LCMV infection. a**, Expression of tissue signature genes extracted from GSEA by CD45.1⁺CXCR6⁺CD8 T cells (n = 3). **b**, GSEA of liver CD45.1⁺CXCR6⁺CD8 T cells compared to spleen CD45.1⁺CX₃CR1⁺CD8 T cells (n = 3). **c**, GSEA for CREM-dependent genes in liver LCMV gp33-specific CXCR6⁺CD8 T cells after LCMV Armstrong compared to LCMV clone 13 infection (permutation test with Benjamini-Hochberg FDR, n = 3). **d**,**e**, UMAP clusters of publicly available scRNA-seq of liver CD8 T cells during persistent LCMV infection[90] and GSEA for CREM transcription factor target genes, no enrichment was found for clusters 1, 3, 6 (permutation test with Benjamini-Hochberg FDR). false discovery rate (FDR).

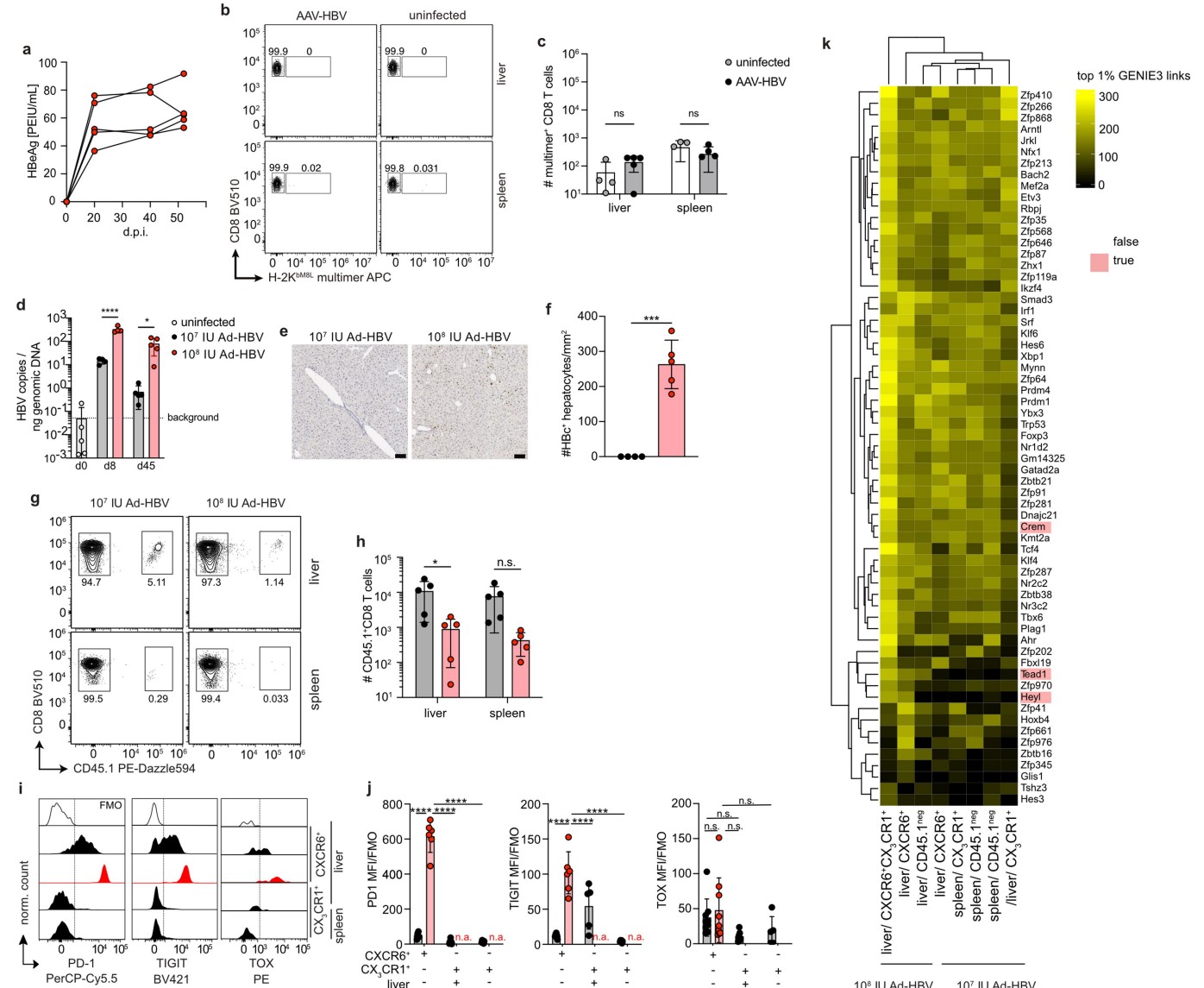

**Extended Data Fig. 3 | Liver CD45.1⁺CXCR6⁺CD8 T cells in a preclinical model of persistent versus acute-resolving HBV infection. a**, Serum HBeAg levels in mice after AAV–HBV infection (n = 4). **b,c**, HB$_{core}$-specific multimer⁺ CD8 T cells in liver and spleen on d≥84 p.i. (AAV–HBV) or uninfected controls and quantification (two-way ANOVA with Sidak's multiple comparison, n = 4). **d**, HBV copies in liver tissue on d8 (left) and d45 (right) p.i. (Ad-HBV) (two-way ANOVA with uncorrected Fisher's LSD, d8 p < 0.0001, d45 p = 0.0434, n = 5). **e,f**, anti-HBcore immunohistochemistry (brown) detecting HBV-replicating hepatocytes at d45 p.i. (Ad-HBV); scale bar 100 μm and quantification (unpaired two-sided t-test p = 0.0001, 10⁷ IU Ad-HBV n = 4, 10⁸ IU Ad-HBV n = 5). **g,h**, gating strategy to detect adoptively transferred CD45.1⁺ TCR-transgenic HB$_{core}$-specific CD8 T cells in liver and spleen d45 p.i. (Ad-HBV) and quantification (two-way ANOVA with Sidak's multiple comparison, liver Padj=0.0311, spleen Padj=0.4872, n = 5). **i,j**, Expression of PD-1, TIGIT and TOX by HB$_{core}$-CD8 T cells in liver and spleen d45 p.i. (Ad-HBV) and quantification (two-way ANOVA with Tukey's multiple comparison, PD1: liver CXCR6⁺ 10⁷ vs. liver 10⁸ IU Ad-HBV Padj<0.0001, liver CXCR6⁺ 10⁸ IU Ad-HBV vs. liver CX₃CR1⁺ 10⁷ IU Ad-HBV Padj<0.0001, liver CXCR6⁺ 10⁸ IU Ad-HBV vs. spleen CX₃CR1⁺ 10⁷ IU Ad-HBV Padj<0.0001, liver CXCR6⁺ vs. liver CX₃CR1⁺ 10⁷ IU Ad-HBV Padj=0.7782, liver CXCR6⁺ vs. spleen

CX₃CR1⁺ 10⁷ IU Ad-HBV Padj=0.7807, liver CX₃CR1⁺ vs. spleen CX₃CR1⁺ 10⁷ IU Ad-HBV Padj>0.9999, TIGIT: liver CXCR6⁺ 10⁷ vs. liver 10⁸ IU Ad-HBV Padj<0.0001, liver CXCR6⁺ 10⁸ IU Ad-HBV vs. liver CX₃CR1⁺ 10⁷ IU Ad-HBV Padj=0.0177, liver CXCR6⁺ 10⁸ IU Ad-HBV vs. spleen CX₃CR1⁺ 10⁷ IU Ad-HBV Padj<0.0001, liver CXCR6⁺ vs. liver CX₃CR1⁺ 10⁷ IU Ad-HBV Padj=0.0427, liver CXCR6⁺ vs. spleen CX₃CR1⁺ 10⁷ IU Ad-HBV Padj=0.9860, liver CX₃CR1⁺ vs. spleen CX₃CR1⁺ 10⁷ IU Ad-HBV Padj=0.0118, TOX: liver CXCR6⁺ 10⁷ vs. liver 10⁸ IU Ad-HBV Padj=0.5073, liver CXCR6⁺ 10⁸ IU Ad-HBV vs. liver CX₃CR1⁺ 10⁷ IU Ad-HBV Padj=0.2543, liver CXCR6⁺ 10⁸ IU Ad-HBV vs. spleen CX₃CR1⁺ 10⁷ IU Ad-HBV Padj=0.3209, liver CXCR6⁺ vs. liver CX₃CR1⁺ 10⁷ IU Ad-HBV Padj=0.9849, liver CXCR6⁺ vs. spleen CX₃CR1⁺ 10⁷ IU Ad-HBV Padj=0.9920, liver CX₃CR1⁺ vs. spleen CX₃CR1⁺ 10⁷ IU Ad-HBV Padj>0.9999, n = 5). **k**, Inferred upstream transcriptional regulators with GENIE3 (top 1% shown) for SMART-Seq2-transcriptomes of CD45.1$^{neg}$, CD45.1⁺CX₃CR1⁺, CD45.1⁺CXCR6⁺ and CD45.1⁺CXCR6⁺CX₃CR1⁺ CD8 T cells after Ad-HBV infection (n≥4). **d-j**: one out of ≥ two independent experiments; n.a. = not analysed; p≥0.05, *p < 0.05, **p < 0.01, ***p < 0.001, ****p < 0.0001, not significant (n.s.) errors shown as s.d., FMO = fluorescence minus one, MFI = geometric mean fluorescence intensity, p.i. = post infection.

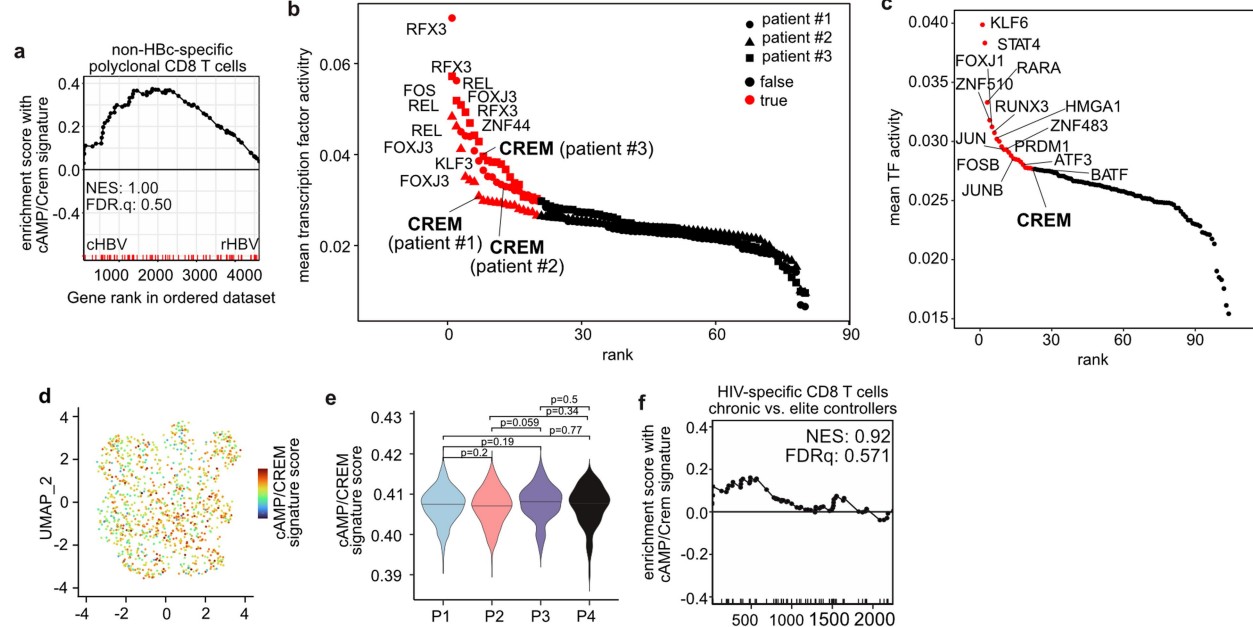

**Extended Data Fig. 4 | Transcriptional profiles of circulating HBV-specific CD8 T cells in patients with chronic Hepatitis B. a**, GSEA for cAMP signalling/ CREM dependent genes with non-HBV-specific bulk CD8 T cells in chronic hepatitis B patients compared to patients with resolved infection (n = 5). **b,c**, Transcription factor activity analysis in circulating HBcore-specific CD8 T cells in two cohorts (n = 3 and n = 4) of chronic hepatitis B patients, top 20

transcription factors with enhanced activity are shown for each patient. **d,e**, UMAP and Ucell cAMP/CREM signature score analysis for circulating HBcore-specific CD8 T cells from four patients with chronic hepatitis B (Wilcoxon test). **f**, GSEA for cAMP signalling/CREM dependent genes with CD8 T cells in patients with persistent compared to controlled HIV infection[31] (n = 4).

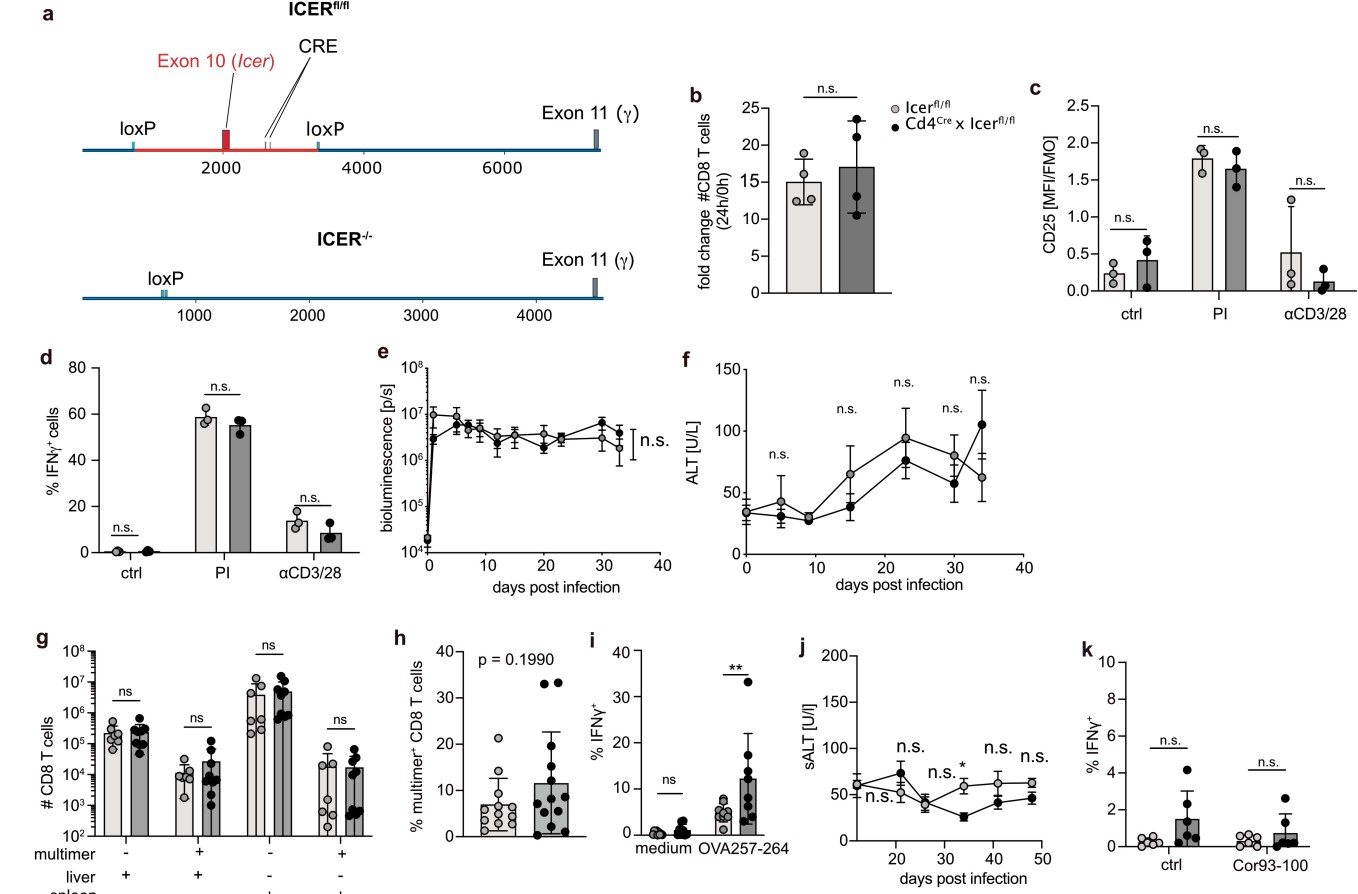

**Extended Data Fig. 5 | CREM/ICER has no checkpoint role to limit effector function in CD8 T cells during persistent hepatotropic infection. a**, Strategy for generation of T cell-specific *Icer* deficient mice and targeting of the *Icer* locus among the *Crem* exons for integration of loxP sites, for details see material and method section. **b**, Expansion of CD8 T cells from *Icer*$^{fl/fl}$ and *Cd4*$^{Cre}$ x *Icer*$^{fl/fl}$ mice after 24 h stimulation in vitro (unpaired two-sided t-test p = 0.5814, n = 4). **c,d**, CD25 expression and INFγ production of CD8 T cells from the spleen of *Icer*$^{fl/fl}$ and *Cd4*$^{Cre}$ x *Icer*$^{fl/fl}$ mice after activation for 3 d in vitro followed by 4 h restimulation with PMA/Ionomycin (PI) or anti-CD3/CD28-coated beads (αCD3/28) or left in medium as control (ctrl) (two-way ANOVA with Sidak's multiple comparison, n = 3). **e,f**, Monitoring of Ad-TTR-GOL-infected *Icer*$^{fl/fl}$ and *Cd4*$^{Cre}$ x *Icer*$^{fl/fl}$ mice via bioluminescence in vivo imaging and sALT measurements (two-way ANOVA with Šídák's multiple comparison, n = 5). **g**, Liver and spleen CD8 T cells from *Icer*$^{fl/fl}$ and *Cd4*$^{Cre}$ x *Icer*$^{fl/fl}$ mice on d30 p.i. (Ad-TTR-GOL)

(two-way ANOVA with Šídák's multiple comparison, Icer$^{fl/fl}$: n = 7, Cd4$^{Cre}$xIcer$^{fl/fl}$: n = 9). **h**, Frequencies of liver antigen-specific multimer$^+$ CD8 T cells in *Icer*$^{fl/fl}$ and *Cd4*$^{Cre}$ x *Icer*$^{fl/fl}$ mice on d30 p.i. (Ad-TTR-GOL) (unpaired two-sided t-test, Icer$^{fl/fl}$: n = 7, Cd4$^{Cre}$xIcer$^{fl/fl}$: n = 9). **i**, INFγ-expressing CXCR6$^+$ CD8 T cells from *Icer*$^{fl/fl}$ and *Cd4*$^{Cre}$ x *Icer*$^{fl/fl}$ mice on d30 p.i. (Ad-TTR-GOL) after re-stimulation with OVA$_{257-264}$ peptide (one-way ANOVA with Sidak's multiple comparison, medium Padj=0.9467, OVA$_{257-265}$ Padj=0.0084, Icer$^{fl/fl}$: n = 7, Cd4$^{Cre}$xIcer$^{fl/fl}$: n = 9). **j**, sALT levels after 10$^8$ IU Ad-HBV infection of *ICER*$^{fl/fl}$ or *Cd4*$^{Cre}$ x *Icer*$^{fl/fl}$ mice (two-way ANOVA with Šídák's multiple comparison, n = 6). **k**, IFNγ production by liver HBcore-specific CD8 T cells from *Cd4*$^{Cre}$ x *Icer*$^{fl/fl}$ or littermate control mice after Cor$_{93-100}$ peptide restimulation ex vivo (two-way ANOVA with Turkey's multiple comparison, n = 6). n.a. = not analysed; b-f one out of ≥ two independent experiments shown; p≥0.05, *p < 0.05, **p < 0.01, ***p < 0.001, ****p < 0.0001, not significant (n.s.). errors are shown as SD with mean.

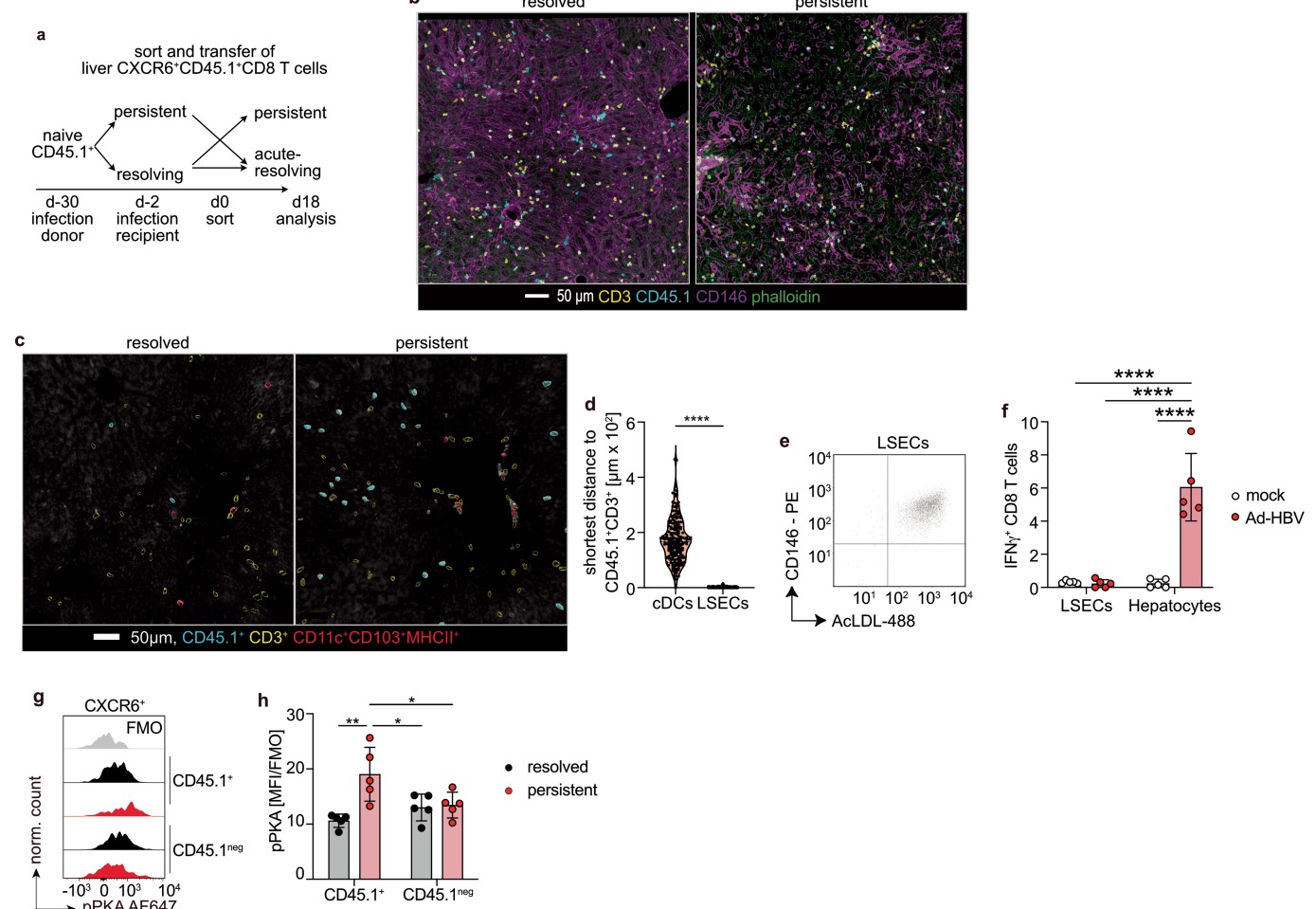

**Extended Data Fig. 6 | Impact of the liver tissue rheostat via AC/cAMP/PKA signalling on virus-specific CD8 T cell function. a**, Experimental scheme illustrating the isolation of liver CXCR6⁺CD45.1⁺ CD8 T cells at d30 p.i. from mice with resolved or persistent infection that were transferred into mice with Ad−CMV-GOL resolving infection or into mice infected with Ad-TTR-GOL developing a persistent infection (d2 p.i.). **b**, Confocal volumetric imaging of liver tissue from mice with resolved or persistent infection (d45 p.i.) analysing localization of CD45.1⁺CD3⁺ T cells and CD146⁺ liver sinusoidal endothelial cells (LSECs), phalloidin for staining of cytoskeleton, bar 50 µm (n = 3). **c**, 3D-rendered surfaces of volumetric confocal microscopy imaging of CD103⁺CD11c⁺MHCII⁺ dendritic cells and CD45.1⁺ antigen-specific T cells in livers of mice with resolved or persistent infection (d45 p.i.; n = 5) **d**, Distance between CD45.1⁺CD3⁺ T cells and CD146⁺ LSECs or cDCs (unpaired two-sided t-test p < 0.0001, n = 3). **e**, Purity of LSECs ( ≥ 98%) isolated from murine livers determined by AcLDL uptake and CD146 expression. **f**, INFγ expression by HBc-specific CD8 T cells cocultured with Ad-HBV or mock-infected hepatocytes or LSECs pre-treated with supernatant of Ad-HBV or mock-infected hepatocytes before coculture to investigate cross-presentation of HBcore antigen (two-way ANOVA with Tukey's multiple comparison, LSEC-ctrl vs. LSEC-Ad-HBV p = 0.9987, LCEC-ctrl vs. hepatocytes-ctrl p = 0.9992, LSEC-ctrl vs. hepatocytes-Ad-HBV p < 0.0001, LSEC-Ad-HBV vs. hepatocytes-ctrl p > 0.9999, LSEC-AdHBV vs. hepatocytes-AdHBV p < 0.0001, hepatocytes-ctrl vs hepatocytes-Ad-HBV p < 0.0001, n = 4, mean with SD). **g**, **h**, pS114 PKA (pPKA) levels in liver and spleen CD45.1⁺CD8 T cells on d45 p.i. (two-way ANOVA with Tukey's multiple comparison, CD45.1⁺ resolved vs. persistent p = 0.0022, CD45.1ⁿᵉᵍ resolved vs. persistent p = 0.9958, CD45.1⁺ resolved vs. CD45.1ⁿᵉᵍ resolved p = 0.6024, CD45.1⁺ persistent vs. CD45.1ⁿᵉᵍ persistent p = 0.0443, CD45.1⁺ persistent vs. CD45.1ⁿᵉᵍ resolved p = 0.0285, CD45.1⁺ resolved vs. CD45.1ⁿᵉᵍ persistent p = 0.4709, n = 5, mean with SD) c,h: one out of ≥ two independent experiments shown (n = 5); p≥0.05, *p < 0.05, **p < 0.01, ***p < 0.001, ****p < 0.0001, not significant (n.s.); FMO = fluorescence minus one, MFI = geometric mean fluorescence intensity, p.i. = post infection.

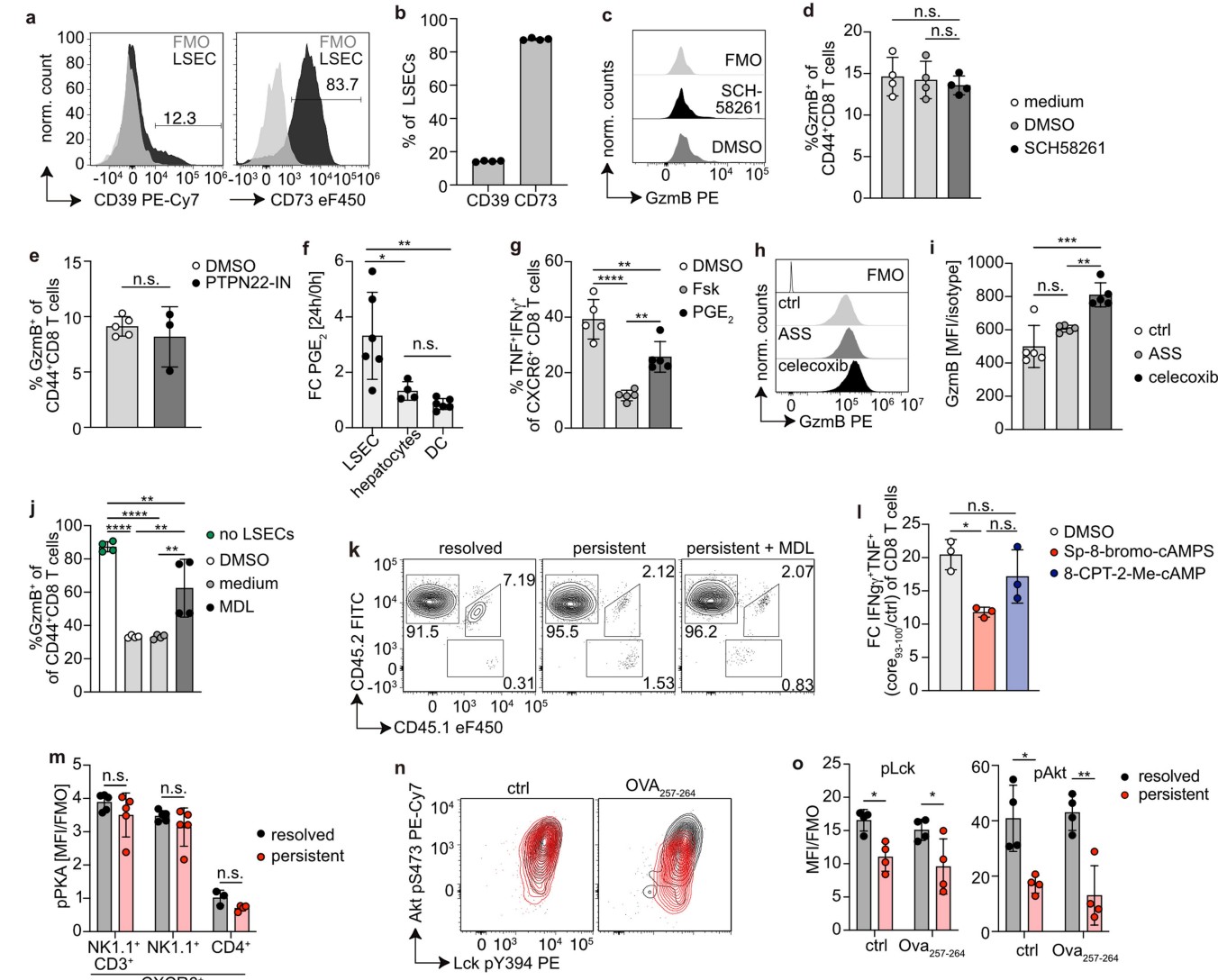

**Extended Data Fig. 7 | Increased adenylyl cyclase-cAMP-PKA signalling and disrupted T cell receptor signalling in CXCR6+ CD8 T cells during persistent hepatotropic infection. a,b,** CD39 and CD73 expression by LSECs and quantification (n = 4). **c,d,** GzmB expression by CD8 T cells co-cultured with activated LSECs for 24 h, T cells were treated with SCH58261 (A2AR antagonist), solvent control, or medium and quantification (one-way ANOVA with Dunnett's multiple comparison, n = 4, mean with SD). **e,** Quantification of GzmB expression by CD44+CD8 T cells in coculture with activated LSECs, T cells were treated with PTPN22-IN (PTP22 inhibitor) or solvent control (unpaired two-sided t-test, DMSO: n = 5, PTP22-IN: n = 3, mean with SD). **f,** prostanoid E2 (PGE$_2$) secretion by mouse LSECs, hepatocytes and dendritic cells (DCs) (one-way ANOVA, Tukey's multiple comparison, LSECs vs. hepatocytes Padj=0.0214, LSECs vs. DCs Padj=0.0023, hepatocytes vs. DCs Padj=0.7465, LSEC: n = 6, hepatocytes: n = 4, DC: n = 6, mean with SD). **g,** Quantification of cytokine expression after OVA$_{257-264}$ peptide stimulation by CD45.1+CXCR6+CD8 T cells isolated from resolved infection and treated with PGE$_2$, Fsk, or solvent control (one-way ANOVA, Tukey's multiple comparison, DMSO vs. Fsk Padj<0.0001, DMSO vs. PGE$_2$ Padj=0.004, Fsk vs. PGE$_2$ Padj=0.0036, n = 5, mean with SEM). **h,i,** GzmB expression and quantification by CD44+CD8 T cells in coculture with LSECs and the selective Cox2 inhibitor celecoxib or acetylsalicylic acid (ASS, two-way ANOVA with Tukey's multiple comparison, ASS vs celecoxib Padj=0.0068, ASS vs ctrl Padj=0.1579, celecoxib vs ctrl Padj=0.0002, n = 5, mean with SD). **J,** GzmB expression and quantification by CD8 T cells co-cultured with activated LSECs for 24 h after 1 h pre-treatment of T cells with MDL-12330A (MDL), solvent

control, or medium and T cells without LSEC contact (one-way ANOVA with Tukey's multiple comparison no LSEC vs. MDL p = 0.0085, no LSEC vs. DMSO p < 0.0001, no LSEC vs. medium p < 0.0001, MDL vs. DMSO p = 0.0026, MDL vs. medium p = 0.0025, DMSO vs. medium p > 0.9999, n = 4, mean with SD). **k,** Gating strategy for the reisolation of CD45.1+/+ CD8 T cells activated in vitro for 3 d followed by 1 h pre-treatment with MDL-12,330 A (MDL) or mock before transfer into mice with resolved or persistent infection for 3d (unpaired t-test, n = 4). **l,).l,** IFNγ+TNF+ CD8 T cells (peptide-stimulated normalized to medium control) after 4 h pre-treatment of activated CD8 T cells with Sp-8br-cAMPS (PKA agonist), 8-pCPT-2′-O-Me-cAMP (EPAC agonist), or solvent control followed by 15 h peptide restimulation (one-way ANOVA with Tukey's multiple comparison Sp-8br-camps vs, 8-pCPT-2′-O-Me-cAMP Padj=0.1091, Sp-8br-camps vs, DMSO Padj=0.0179, 8-pCPT-2′-O-Me-cAMP vs. DMSO Padj=0.3597, n = 3, mean with SD). **M,** pPKA levels by liver CXCR6+ NK, NKT or CD4 T cells at d45 after infection with $10^7$ or $10^8$ IU Ad-HBV (two-way ANOVA with Tukey's multiple comparison, n = 5, mean with SD). **n,o,** pS473 Akt (pAkt) and pY394 Lck (pLck) levels in virus-specific liver CXCR6+CD8 T cells after ex vivo OVA$_{257-264}$ peptide restimulation or medium control at d30 p.i. with Ad-CMV-GOL or Ad-TTR-GOL (one-way ANOVA with Sidak's multiple comparison, pLck ctrl Padj=0.0320, Ova$_{257-264}$ Padj=0.0315, pAkt ctrl Padj=0.0401, Ova$_{257-264}$ Padj=0.0087, n = 5, mean with SD). A-o: one out of ≥ two independent experiments shown; p≥0.05, *p < 0.05, **p < 0.01, ***p < 0.001, ****p < 0.0001, not significant (n.s.); FMO = fluorescence minus one, MFI = geometric mean fluorescence intensity, Padj = adjusted p-value.

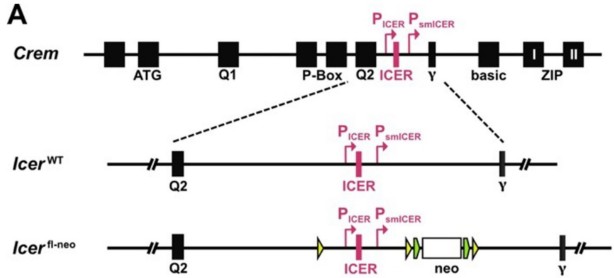

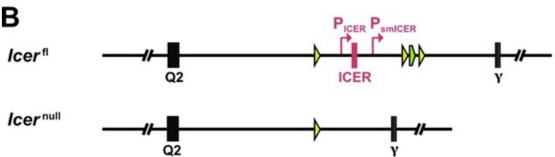

**Extended Data Fig. 8 | Strategy for the generation of a floxed ICER knockout mouse to achieve a T cell-specific ICER knockout. a**, Illustration of the genomic region encompassing the ICER-specific exon as well as the alternative promoters driving expression of ICER and smICER, respectively and of the flanked by loxP sites used for homologous recombination in mouse ES cells. **b**, illustration of the strategy for generating the *Icer*^fl allele.

# Reporting Summary

## Statistics

For all statistical analyses, confirm that the following items are present in the figure legend, table legend, main text, or Methods section.

| n/a | Confirmed | |
|---|---|---|
| ☐ | ☒ | The exact sample size (*n*) for each experimental group/condition, given as a discrete number and unit of measurement |
| ☐ | ☒ | A statement on whether measurements were taken from distinct samples or whether the same sample was measured repeatedly |
| ☐ | ☒ | The statistical test(s) used AND whether they are one- or two-sided *Only common tests should be described solely by name; describe more complex techniques in the Methods section.* |
| ☐ | ☒ | A description of all covariates tested |
| ☐ | ☒ | A description of any assumptions or corrections, such as tests of normality and adjustment for multiple comparisons |
| ☐ | ☒ | A full description of the statistical parameters including central tendency (e.g. means) or other basic estimates (e.g. regression coefficient) AND variation (e.g. standard deviation) or associated estimates of uncertainty (e.g. confidence intervals) |
| ☐ | ☒ | For null hypothesis testing, the test statistic (e.g. *F*, *t*, *r*) with confidence intervals, effect sizes, degrees of freedom and *P* value noted *Give P values as exact values whenever suitable.* |
| ☒ | ☐ | For Bayesian analysis, information on the choice of priors and Markov chain Monte Carlo settings |
| ☐ | ☒ | For hierarchical and complex designs, identification of the appropriate level for tests and full reporting of outcomes |
| ☒ | ☐ | Estimates of effect sizes (e.g. Cohen's *d*, Pearson's *r*), indicating how they were calculated |

*Our web collection on statistics for biologists contains articles on many of the points above.*

## Software and code

Policy information about availability of computer code

| Data collection | The following machines were used for data collection for the respective application: |
|---|---|
| | Flow cytometry: SP6800 (Sony Biotechnology); CytoFLEX S (Beckman Coulter) |
| | Fluorescence activated cell sorting: SH800 (Sony Biotechnology); MoFlo Astrios EQ (Beckman Coulter); FACSMelody Cell Sorter (BD Biosciences); BD SORP FACS Aria (BD Biosciences) |
| | Real-time impedance based cytotoxicity measurement: xCelligence RTCA MP device (ACEA Biosciences) |
| | Confocal microscopy: TCS SP8 (Leica) |
| | qRT-PCR: LightCycler 480 (Roche) |
| | in vivo imaging: IVIS Lumina LT-Series III (Perkin Elmer) |
| | sALT measurement: Reflotron®plus system (Roche) |
| | HBeAg measurement in peripheral blood: ArchitectTM platform and the HBeAg reagent kit (Ref.: 6C32-27) with HBeAg quantitative calibrators (Ref.: 7P24-01, all: Abbott Laboratories) |
| | Immunohistochemistry: Bond MAX, Bond Rxm, Aperio (Leica Biosystems) |

Next Generation Sequencing: NextSeq 500 (Illumina); HiSeq2500 (Illumina); HiSeq3000 (Illumina); NovaSeq6000 (Illumina); MiSeq system (GenDX)

Data analysis

The following software, tools, packages and algorithms were used for data analysis:

Aperio Image Scope v12.4.0 (Leica); QuPath (v0.2.3); FlowJo v10.7.1 & v10.8.0 (BD); Imaris v9.6 (Bitplane); Drop-seq v1.12 pipeline (https://github.com/broadinstitute/Drop-seq); DESeq2 R package  v2.1.28.1; GREIN DB v1; ggplot2 R package v3.3.2; prcomp function R v3.6.1; ggplot 2 & ggrepel R package v0.9.4; R v4.1.2 with Seurat package v4.3.0; Limma R package v3.58.1; GSEA v4.0.3; Python v3.12 custom script (https://zenodo.org/records/11040043); Cytoscape v3.7.1; igraph R package v2.0.2 (https://igraph.org/); bcl2fastq software v2.20.0.422; snakemake pipelines (https://gitlab.lrz.de/ImmunoPhysio/bulkSeqPipe);
Trimmomatic v0.36;  STAR v2.5.3a; htseq v0.9.1; DESeq2 v1.24.03a;  scPipe R package workflow v1.12.0, Seurat R package v3.2.0; pySCENIC pipeline v0.10.10; Harmony v1.2.0; clustree R package v0.4.0; UCELL R package v1.2.4

For further details please see methods section.

For manuscripts utilizing custom algorithms or software that are central to the research but not yet described in published literature, software must be made available to editors and reviewers. We strongly encourage code deposition in a community repository (e.g. GitHub). See the Nature Portfolio guidelines for submitting code & software for further information.

# Data

Policy information about availability of data

All manuscripts must include a data availability statement. This statement should provide the following information, where applicable:
- Accession codes, unique identifiers, or web links for publicly available datasets
- A description of any restrictions on data availability
- For clinical datasets or third party data, please ensure that the statement adheres to our policy

Data from mouse RNAseq are deposited at GEO accession number:
GSE168096 (https://www.ncbi.nlm.nih.gov/geo/query/acc.cgi?acc=GSE168096 token: yhwvqmmaxhudjqh)
GSE212925 (https://www.ncbi.nlm.nih.gov/geo/query/acc.cgi?acc=GSE212925 token: qxylgomkrhivfwx)
GSE214151 (https://www.ncbi.nlm.nih.gov/geo/query/acc.cgi?acc=GSE214151 token:wzyteyoyhzmntqx)

PRJEB36998 (https://www.ebi.ac.uk/ena/browser/view/PRJEB36998, published in Sandu et al., 2020, Cell reports)

Single-cell RNA-sequencing data of CD8 T cells from liver explants: https://figshare.com/s/9db8a1f1de89c1e4f18c
Single cell RNA-sequencing data of HBc-specific CD8 T cells from patients with chronic Hepatitis B virus infection: https://figshare.com/s/245d38cb7c4901b70b3f

Publicly available data sets:

GSE47045 (ref17); GSE70813 (ref19); MsigDB BIOCARTA dataset (https://www.gsea-msigdb.org/gsea/msigdb/); Molecular Genome Informatics database (http://www.informatics.jax.org/go/term/GO:0019722); http://amp.pharm.mssm.edu/Harmonizome (ref21); human gene database GeneCards (https://www.genecards.org/; TF checkpoint database (ref23); Eukaryotic promoter database  & UCSC (GRCm38/mm10) database (ref24, https://genome.ucsc.edu/cgi-bin/hgTrackUi?db=mm10&c=chrX&g=encode3RenEnhancerEpdNewPromoter); JASPAR core database (ref25); HOCOMOCO database (ref26); GRCm38 reference genome ENSEMBL (annotation release #75, #91); human genome (version GRCh38); human CD8 T cell signatures from ref33

For further details please see methods section.

# Human research participants

Policy information about studies involving human research participants and Sex and Gender in Research.

Reporting on sex and gender

Human patient data (persistent HBV infection): PMBCs were collected from female patients (sex). Data on gender was not collected.

Population characteristics

Human patient data (persistent HBV infection): PBMCs were collected from patients with a age range from 29 to 53 years who were diagnosed for chronic Hepatitis B virus infection, naive to therapy, HBeAg negative, anti-HBs antibody negative, anti-HBe antibody negative, ALT [U/ml] in a range from 19-46, viral load [IU/mL] ranging from 31-1795.
Human research participants with hepatitis B virus infections whose liver CD8 T cells were analysed after fine-needle aspriation were stratified according to their hepatitis and HBeAg status as outlined in Fig.3e.

Recruitment

Patients with chronic hepatitis B for PBMC collection were recruited after giving informed consent to participate in this study. Patients with chronic hepatitis B or patients who had cleared the infection were recruited from outpatient clinics at the different sites for analysis of circulating or hepatic HBV-specific CD8 T cells and were included in the study based on the disease state (ongoing viral hepatitis with detection of viral markers and high ALT levels, anti-HBe+ infection with detection of viral markers and low ALT levels and clearance of infection with loss of viral markers and absence of increased ALT levels).

| Ethics oversight | Immunohistochemistry of human liver tissue was conducted according to federal guidelines, local ethics committee regulations of the Technical University of Munich, Germany (No 518/19 S-SR). Isolation of PBMCs from patients was conducted according to federal guidelines, local ethics committee regulations of Albert-Ludwigs-Universität, Freiburg, Germany (no. 474/14). Liver fine-needle-aspirations (FNAs) were collected from participants living with CHB at the Erasmus MC University Medical Center (Rotterdam, The Netherlands), the Toronto General Hospital (Toronto, Canada), and the Massachusetts General Hospital (Boston, USA). All participants provided written informed consent. This study was approved by institutional review boards at all 3 sites and was conducted in accordance with both the declaration of Helsinki and Istanbul. |
|---|---|

Note that full information on the approval of the study protocol must also be provided in the manuscript.

# Field-specific reporting

Please select the one below that is the best fit for your research. If you are not sure, read the appropriate sections before making your selection.

☒ Life sciences ☐ Behavioural & social sciences ☐ Ecological, evolutionary & environmental sciences

For a reference copy of the document with all sections, see nature.com/documents/nr-reporting-summary-flat.pdf

# Life sciences study design

All studies must disclose on these points even when the disclosure is negative.

| Sample size | Pilot experiments were used to estimate the sample size such that an appropriate statistical test could yield significant results. The exact n numbers used in this study are indicated in the each figure legend. |
|---|---|
| Data exclusions | Every mouse infected with an adenovirus or uninfected control mice was included in the data analysis. The acquired data of an individual (human or mouse) was only excluded if technical problems during sample processing or data acquisition occurred. |
| Replication | All experiments were repeated several times, details are given in each figure legend. |
| Randomization | Murine experiments: all mice were randomly assigned into experimental groups and treated accordingly.<br>The allocation of patients with chronic hepatitis B occurred based on the presence of viral markers and immune response markers (anti-HBe) and ALT levels. |
| Blinding | Bioinformatic analyses, histological analyses and analyses of confocal images were performed with code-labeled samples. Human samples were pseudonomised for all further processing. Researchers were not blinded for treatment or genotypes of mice to avoid mix-up of samples and handling by several scientists and research associates, experimental design and appropriate controls ensured accuracy and reproducibility of measurements and analyses. |

# Reporting for specific materials, systems and methods

We require information from authors about some types of materials, experimental systems and methods used in many studies. Here, indicate whether each material, system or method listed is relevant to your study. If you are not sure if a list item applies to your research, read the appropriate section before selecting a response.

## Materials & experimental systems

| n/a | Involved in the study |
|---|---|
| ☐ | ☒ Antibodies |
| ☐ | ☒ Eukaryotic cell lines |
| ☒ | ☐ Palaeontology and archaeology |
| ☐ | ☒ Animals and other organisms |
| ☒ | ☐ Clinical data |
| ☒ | ☐ Dual use research of concern |

## Methods

| n/a | Involved in the study |
|---|---|
| ☒ | ☐ ChIP-seq |
| ☐ | ☒ Flow cytometry |
| ☒ | ☐ MRI-based neuroimaging |

# Antibodies

| Antibodies used | The following antibodies (clone, dilution, supplier, catalogue number) were used for staining of mouse cells:<br>anti-CD8 (53-6.7, 1:250, Biolegend, #100752), anti-CD45.1 (A20, 1:200, Biolegend, #110722, #110704, #110748), anti-CXCR6 (SA051D1, 1:200, Biolegend, #151117, #151104, #151108, #151109, #151115), anti-CX3CR1 (SA011F11, 1:200, Biolegend, #149016, #149004, #149006), anti-CD44 (IM7, 1:200, Biolegend, #103036), anti-CD69 (H1.2F3, 1:100, Biolegend, #104530 or Thermo Fisher Scientific #63-0691-82), anti-TIM-3 (B8.2C12, 1:200, Biolegend, #134008 and ThermoFisher Scientific #12-2231-82), anti-TIGIT (1G9, 1:200, Biolegend, #142111), anti-IFN-g (XMG1.2, 1:200, Biolegend, #505808), anti-CD19 (1D3, 1:200, Biolegend, #152404), anti-CD335 (29A1.4, 1:200, Biolegend, #137606), anti-Lck pY394 (A18002D, 1:100, Biolegend, #933104), CD39 (Duha59, 1:200, Biolegend, |
|---|---|

#143805), anti-CD45.2 (104, 1:200, Biolegend, #109805), anti-CD3 (17A2, 1:200, Biolegend, #100217), anti-NK1.1 (PK136, 1:100, Biolegend, #108747), anti-CD4 (GK1.5, 1:200, Biolegend, 100449), anti-CD49a (HMa1, 1:200, Biolegend, #142606),anti-PD-1 (J43, 1:200, Thermo Fisher Scientific, #46-9985-82), anti-LAG-3 (eBioC9B7W, 1:200, Thermo Fisher Scientific, #12-2231-82, #406-2239-42), anti-Tox (TXRX10, 1:100, Thermo Fisher Scientific, #12-6502-82), anti-Granzyme B (GB11, 1:200, Thermo Fisher Scientific, #GRB04 and #GRB05), anti-TNF (MP6-XT22, 1:200, Thermo Fisher Scientific, #25-7321-82), anti-4-1BB (17B5, 1:100, Thermo Fisher Scientific, #48-1371-82), anti-CD25 (PC61.5, 1:200, Thermo Fisher Scientific, #48-0251-82), anti-Akt pS473 (SDRNR, 1:100, Thermo Fisher Scientific, #25-9715-42), anti-rabbit IgG Fab2 (1:500, Cell signalling, #79408), anti-pPKA (47/PKA, 1:5, BD Biosciences, #560205), anti-CD103 (goat polyclonal, 1:200, R&D Systems, #AF1990), anti-MHCII (M5/144.15.2, 1:200, Biolegend, #107636), anti-CD146 (ME9F1, 1:100, Miltenyi, #130-102-846) anti-CD335 (29A1.4, 1:200, Biolegend, #137606), anti-CD73 (TY/11.8, 1:200, Thermo Fisher Scientific, #48-0731-82).

For staining of human cells, the following antibodies (clone, dilution, catalog number, Lot number) were used:
anti-CD14 (61D3, 1:100, #A15453, Lot 2406638), anti-CD19 (HIB19, 1:100, #17-0199-42, Lot 2472560) (all Thermo Fisher Scientific), anti-CD45RA (HI100, 1:200, #304178, Lot #2327528), anti-CCR7 (1:20, G043H7, #353244, Lot B347205) (all Biolegend), anti-CD8 (RPA-T8, 1:200, #563795, Lot 9346411), and anti-GZMB (GB11, 1:100, #563388, Lot 3317967) (all BD Bioscience)

The following antibodies were used for immunohistochemistry:
1.5 µg/mL polyclonal anti-HBcAg (Origene, #AP08118PU-S); 0.4 ng/µl polyclonal anti-GFP (Fitzgerald, #70R-10652)

The following antibodies (clone, dilution, supplier, catalog number) were used for staining of tissue sections analysed by confocal immunofluorescence imaging:
anti-CD3 (clone 17A2, 1:200, Biolegend, #100240,), anti-CD45.1 (clone A20, 1:200, Biolegend, #110732), anti-CD146 (clone ME-9F1, 1:100, Miltenyi, #130-102-846), anti-I-A/I-E (MHC class II) (clone M5/114.15.2, 1:200, Biolegend, #107622) and anti-CD103 (goat polyclonal, 1:200, R&D Systems, #AF1990) followed by anti-goat IgG (donkey polyclonal, 1:500, Jackson ImmunoResearch, #705-625-147).

| Validation | All antibodies listed in the previous section were validated by the manufacturer and/or by previous studies.

Information on the validation of antibodies for flow cytometry can be found as stated below:

Biolegend antibodies: https://www.biolegend.com/en-us/quality/quality-control
Biolegend employs a comprehensive approach to antibody validation, analyzing 1-3 target cell types with single- and multi-colour analysis to encompass positive and negative cell types. Upon confirming specificity, each new lot is required to match the intensity of the in-date reference lot, with the brightness (MFI) evaluated across both positive and negative populations to ensure consistency. Furthermore, quality control testing, including a series of titration dilutions, is conducted for every lot.

Thermo Fisher Scientific antibodies: https://www.thermofisher.com/de/de/home/life-science/antibodies/invitrogen-antibodyvalidation.html
Thermo Fisher Scientific tests each antibody using different methods, including flow cytometry, Immunoprecipitation-Mass Spectrometry Antibody Validation, Knockout and Knockdown Antibody Validation, Independent Antibody Validation, Peptide Array Antibody Validation, Cell Treatment, Neutralization Antibody Validation, Relative Expression Antibody Validation, and SNAP-ChIP Antibody Validation. The precise validation method for each antibody is outlined in its respective antibody datasheet.

BD Biosciences antibodies: https://www.bdbiosciences.com/en-eu/products/reagents/flow-cytometry-reagents/research-reagents/quality-and-reproducibility
BD Biosciences tests each antibody on primary cells, cell lines or transfectant models using different methods, including flow cytometry, immunofluorescence, immunohistochemistry, or western blot. The precise validation method for each antibody is outlined in its respective antibody datasheet.

Cell signalling antibodies: https://www.cellsignal.com/about-us/our-approach-process/antibody-validation-flow-cytometry
Flow-validated products undergo rigorous testing in biologically relevant models, ensuring specificity and an optimal signal-to-noise ratio (S/N) for both conjugated and unconjugated antibodies. Cross-platform validation further confirms antibody specificity. In addition, all antibodies have been tested for optimal dilution, specificity, stability and lot-to-lot reproducibility.

Miltenyi antibodies: https://www.miltenyibiotec.com/DE-en/products/macs-antibodies/antibody-validation.html
All antibodies are rigorously tested and validated before release. The precise validation method for each antibody is outlined in its respective antibody datasheet.

R&D Systems antibodies:https://www.rndsystems.com/products/rd-systems-approach-antibody-quality
Each antibody is manufactured under controlled conditions, undergoing rigorous quality control testing to ensure lot-to-lot consistency. Validation includes externsive specificity testing and testing of cross-reactivity using a variety of applications. he precise validation method for each antibody is outlined in its respective antibody datasheet.

The anti-GFP and the anti-HBcAg antibodies were validated by the Institute of Pathology, School of Medicine, TUM. |

# Eukaryotic cell lines

Policy information about cell lines and Sex and Gender in Research

| Cell line source(s) | HEK293 cells (CRL-1573™) were obtained by ATCC, USA |

| Authentication | cell line was not authenticated |
|---|---|
| Mycoplasma contamination | Cell line was regularly tested for mycoplasma contamination and results were always negative. |
| Commonly misidentified lines (See ICLAC register) | *Name any commonly misidentified cell lines used in the study and provide a rationale for their use.* |

# Animals and other research organisms

Policy information about studies involving animals; ARRIVE guidelines recommended for reporting animal research, and Sex and Gender in Research

| Laboratory animals | 6-8 week old C57Bl/6J male mice were purchased from Janvier or Charles River. H-2Kb(SIINFEKL)-restricted TCR-transgenic CD45.1+ mice, H-2Kb(MGLKFRQL)-restricted TCR-transgenic CD45.1+ mice (Ref: Isogawa 2013, PLOS Pathogens, doi:10.1371 journal.ppat.1003490, purchased from Charles River), and CD4-CrexICERfl/fl mice (B6.Cg-Tg(Cd4-cre)1Cwi x ICER-fl/fl) were bred under specific pathogen free conditions at TranslaTUM, Klinikum rechts der Isar. Mice were housed with a 12 h light - 12 h dark cycle. Temperature was set to 22+/-2 °C, humidity to 55+/-10% and checked daily. Wild type littermates were used as controls as indicated. |
|---|---|
| Wild animals | No wild animals were used in this study. |
| Reporting on sex | In vivo experiments in mice were performed in male mice to have comparable virus to bodyweight ratios and quantitatively comparable antiviral immune responses. |
| Field-collected samples | No field-collected samples were used in this study. |
| Ethics oversight | Guidelines of the Federation of Laboratory Animal Science Association were implemented for breeding and experiments. Experiments were approved by the District Government of Upper Bavaria, Germany (permission numbers ROB-55.2-2532.Vet_02-14-185; ROB-55.2-2532.Vet_02-16-55, ROB-55.2-2532.Vet_02-18-100). |

Note that full information on the approval of the study protocol must also be provided in the manuscript.

# Flow Cytometry

## Plots

Confirm that:

☒ The axis labels state the marker and fluorochrome used (e.g. CD4-FITC).

☒ The axis scales are clearly visible. Include numbers along axes only for bottom left plot of group (a 'group' is an analysis of identical markers).

☒ All plots are contour plots with outliers or pseudocolor plots.

☒ A numerical value for number of cells or percentage (with statistics) is provided.

## Methodology

| Sample preparation | Isolation and culture of primary mouse cells: |
|---|---|
| | Splenocyte isolation |
| | Spleens were passed through a 100 μm cell strainer and red blood cells were lysed with Ammonium-Chloride-Potassium lysing buffer for 2 min. |
| | |
| | Isolation of liver-associated lymphocytes |
| | Before excision, livers were perfused with PBS via the portal vein. Livers were passed through 100 μm cell strainers and digested with 125 μg/mL collagenase type II (Worthington) in Gey's balanced salt solution (GBSS, PAN Biotech) for 10 min at 37° C. For enrichment of liver-associated lymphocytes, a density gradient centrifugation with 40%/80% Percoll (GE Healthcare) was performed at 1440 x g for 20 min. |
| | |
| | Isolation of primary mouse hepatocytes |
| | Livers were perfused with 0.12 U/mL collagenase (SERVA) at 6 mL/min for 8 min via the portal vein. Livers were then removed, mechanically disrupted and passed through a 300 μm cell strainer. Liver cell suspensions were filtered through a 100 μm mesh and pelletised at 50 x g for 2 min. Hepatocytes were purified by density gradient centrifugation with 50%/80% Percoll (GE Healthcare) at 600 x g for 20 min. For cytotoxicity assays, 10,000 hepatocytes per well were seeded on 96 well E-plates (ACEA Biosciences) coated with 0.02% collagenR (SERVA). Cell attachment was achieved in supplemented William's E medium (PAN Biotech, 200 mM Glutamine (Thermo Fisher Scientific), 1 M Hepes pH 7.4, 104 U/mL Penicillin/Streptomycin, 50 mg/mL gentamycin (Merck), 0.005 ng/mL insulin (INSUMAN rapid, Sanofi), 1.6% DMSO (Merck) and 10% FBS (PAN Biotech). Attached cells were cultivated in supplemented William's E medium (as above) containing 1% FBS. |
| | |
| | Isolation of primary mouse liver sinusoidal endothelial cells |
| | Nonparenchymal liver cells were isolated from mouse livers after portal vein perfusion with collagenase collagenase type II (Worthington) in GBSS (PAN Biotech), followed by in vitro digestion with collagenase (type II, Worthington) in a rotatory water bath at 37 °C and density gradient centrifugation. Liver sinusoidal endothelial cells (LSECs) were then obtained by |

immunomagnetic separation using anti-CD146 coated microbeads (Miltenyi biotec) reaching a purity of ≥95%. LSECs were cultured in collagen coated flat-bottom 96 well microplates until they reached confluence for 48 h after isolation, and after careful medium exchange LSECs were then used for experiments.

Ex vivo stimulation/treatment
T cells were cultivated in RPMI-1640 medium (GIBCO) supplemented with 10% FCS, 1% L-Glutamine (200 mM), 1% Penicillin/Streptomycin (5000 U/mL), 50 μM 2-mercaptoethanol. For ex vivo stimulation and intracellular cytokine staining, cells were stimulated with 10 nM recombinant OVA peptide (SIINFEKL, peptides&elephants GmbH), HBV core peptide (MGLKFRQL, peptides&elephants GmbH) or 1x eBioscienceTM Cell stimulation cocktail (Thermo Fisher Scientific) together with 3 μg/mL Brefeldin A (Invitrogen). To analyse cAMP signalling, T cells were incubated for 1 h with the adenylyl cyclase agonist Forskolin (25 μM, Sigma-Aldrich), the PKA agonist Sp-8br-cAMPS (250 μM, Cayman Chemical), the EPAC agonist 8-pCPT-2'-O-Me-cAMP (30μM, Tocris), or the adenosine A2A receptor agonist CGS21680 (100 nM, Tocris) solved in DMSO (Sigma-Aldrich).

PBMC isolation from patients:
Venous blood samples were collected in EDTA-coated tubes. PBMCs were isolated by density gradient centrifugation using lymphocyte separation medium (PAN Biotech). Isolated PBMCs were resuspended in RPMI 1640 medium supplemented with 10% FCS, 1% penicillin/streptomycin and 1.5% 1 M HEPES buffer (ThermoFisher) and stored at -80 °C until used. Frozen PBMCs were thawed in complete medium (RPMI 1640 supplemented with 10% FCS, 1% penicillin/streptomycin and 1.5% 1 M HEPES buffer (ThermoFisher)) containing 50 U ml−1 benzonase (Sigma).

Surface stainings were performed at 4° C for 30 min. MHC class I H-2KbSIINFEKL-restricted or H-2KbMGLKFRQL -restricted streptamers for staining of murine T cells (Nauerth et al., 2016, DOI: 10.1002/cyto.a.22933) were kindly provided by D. Busch (Institute of Microbiology, TUM). For labelling, 0.4 μg streptamer per sample were incubated with 0.4 μL Strep-Tactin-PE/APC (IBA lifesciences) in PBS for 30 min on ice prior to incubation with cell suspensions.
HLA class I epitope-specific tetramers for staining of human T cells were generated through conjugation of biotinylated peptide/HLA class I monomers with PE-conjugated streptavidin (ProZyme, USA) at a peptide/HLA I:streptavidin molar ratio of 5:1.
To exclude dead cells, Fixable Viability Dye eFluor780 (Invitrogen) was included in the staining panels. For intracellular staining of cytokines, IC fixation buffer (Invitrogen) was used according to the manufacturer´s instructions. Staining of Granzyme B and Tox was performed in combination with Foxp3 / Transcription Factor staining Buffer set (Thermo Fisher Scientific) according to the manufacturer´s instructions. For staining of Crem and pPKA, cells were fixed in IC fixation buffer (Invitrogen) for 30 min and permeabilized with ice-cold Methanol for 30 min before staining.

Sample preparation for bulk RNA sequencing of OVA257-264-specific CD45.1+ CD8 T cells: Liver-associated lymphocytes and splenocytes from mice with resolved Ad-CMV-GOL infection were sorted into CD45.1+CXCR6+CX3CR1negCD8 and CD45.1+CXCR6negCX3CR1+CD8 T cells. CD8 T cells derived from mice with persistent Ad-TTR-GOL infection were sorted into CXCR6+CX3CR1negCD45.1+CD8 and CXCR6+CX3CR1+CD45.1+CD8 populations. 5000 cells per sample were collected in 1 x TCL lysis buffer (Qiagen) supplemented with 1% (v/v) 2-mercaptoethanol and immediately frozen on dry ice.

Sample preparation for bulk RNA sequencing of P14 LCMV-specific CD8 T cells: P14 cells were adoptively transferred into C57BL/6 mice and infected one day later with either LCMV Clone 13 or LCMV Armstrong. Resident (CD69+CD101+CXCR6+CX3CR1neg) and effector/effector-memory (CX3CR1+) P14 cells from the liver were sorted at d27 p.i. . Total RNA was isolated using the RNAdvance Cell v2 kit (Beckmann-Coulter).

Sample preparation for bulk RNA sequencing of Cor93-100-specific CD45.1+CD8 T cells: Liver-associated lymphocytes and splenocytes from mice with Ad-HBV infection were pre-gated on (CD19/Ly6G/TER119/CD335)neg CD8 T cells and sorted into liver CXCR6+CD45.1+, liver CD45.1+CX3CR1+ CD8 T cells, spleen CD45.1+CX3CR1+ CD8 T cells and liver CD45.1neg CD8 T cells from resolved infections and liver CD45.1+CXCR6+ and liver CD45.1+CXCR6+CX3CR1+ CD8 T cells and liver CD45.1neg CD8 T cells from persistent infection. 100 CD8 T cells were directly sorted into 96 well plates prepared with 1X Reaction Buffer consisting of lysis buffer and RNase Inhibitor for low input RNA sequencing (Takara). Plates were spun down and immediately stored on dry ice or at -80°C until further processing.

Sample preparation for scRNAseq of human HBcore-specific CD8 T cells: HBVcore18-specific CD8 T cells were enriched by magnetic bead-based sorting, and surface staining was performed. In total, 1152 live HBVcore18-specific CD8 T cells were sorted in 384-well plates (Bio-Rad) containing lysis buffer and mineral oil using FACS Melody Cell Sorter in single-cell sorting mode. Naive CD45RA+CCR7+ T cells were excluded.

scRNAseq of human HBV-specific CD8 T cells isolated from the liver by fine needle aspiration: Cells were thawed and stained with lineage marker antibodies as well as HBV multimers for two distinct HBV-specificities. The live HBV-specific CD8 T cells were sorted in 96-well Armadillo plates (Thermo Fisher Scientific) containing RNA lysis buffer using a BD SORP FACS Aria in index single-cell sorting mode.

| Instrument | Sony SP6800 spectral analyzer (Sony Biotechnology) and CytoFLEX S (Beckman Coulter) |
| --- | --- |
| Software | Data collection was performed with the SP6800 software (Sony Biotechnology) and analysed with FlowJo v10.7.1 & v10.8.0 (BD), R v4.0.2 and R cytofkit GUI v0.99 |
| Cell population abundance | Samples were sorted using the purity mode. Purity check post sorting confirmed >95% purity. |
| Gating strategy | FSC-A/SSC-A (Lymphocytes) -> FSC-W/FSC-H (Singlets) -> live-dead/autofluorescence (viable autofluorescence-negative) -> T cell stainings |

☒ Tick this box to confirm that a figure exemplifying the gating strategy is provided in the Supplementary Information.

