## [Peer Review File · Nature]

Manuscript Title: A liver immune rheostat regulates CD8 T cell immunity in chronic HBV infection

Reviewer Comments & Author Rebuttals

Reviewer Reports on the Initial Version:

Referees' comments:

Referee #1 (Remarks to the Author):

Bosch and Kallin et al. studied the phenotype and function of antigen-specific CD8 T cells mainly in in vivo (mouse) models of hepatotropic viral infection, and substantiated some links in human chronic HBV in PBMCs. Using both OVA- and HBV-based hepatotropic infection models, and contrasting with systemic (LCMV) infection models, they demonstrated features of antigen-specific T cells unique to liver-restricted persistent infection, including their almost-exclusive hepatic localization, CXCR6+CX3CR1± phenotype, dysfunctionality for effector cytokine and cytolytic capacity, and Crem transcriptional network/signature enrichment. Furthermore, to explain these features, they identified increased LSEC-CD8 T cell interactions using confocal microscopy in persistent infection, with OVA-presenting LSEC/CD8 T cell co-culture demonstrating loss of IFN γ production in restimulation. Mechanistically the cAMP/PKA axis was identified downstream, with the functional phenotype of persistent infection recapitulated in T cells from resolved infection by in vitro pharmacological activation of cAMP/PKA. In humans, they find CREM among the top transcription factors in circulating HBcore-specific CD8 T cells.

The work is technically well executed and the characterization of the mechanisms of the hepatic CD8 T cell dysfunction is novel and detailed. On the other hand, for a paper that is designed to analyze the cause of CD8 T cell dysfunction present in chronic HBV infection, the data related to real HBV infection is very limited. Understandably, due to the infeasibility of the direct use of HBV in mice, the authors used an HBV-Adenoviral system to mimic HBV infection. They then analyzed HBcore-specific CD8 T cells sorted from only 3 patients, who were poorly characterized in terms of HBV chronic stage. While I understand that there are technical limitations to obtain HBV-specific T cells in CHB patients, particularly from the liver, I am still not convinced that this study can prove that the mechanism shown in different models of persistence antigen presentation in the liver accurately mimics what is occurring in HBV infection. The authors also claim that this manuscript identifies a new "liver tissue rheostat that induces loss of the T cell function", but this reviewer remains doubtful of the differences between this "novel" phenomena and the mechanism of hepatic CD8 T cell dysfunction described by the same authors 20 years ago in their paper "Efficient presentation of exogenous antigen by liver endothelial cells to CD8+ T cells results in antigen-specific T-cell tolerance" [Nature Medicine 6 (2000), 12. - S. 1348-1354 <https://dx.doi.org/10.1038/82161>]; particularly whether we are talking of the same or different phenomena, and as such, how this "new liver tissue rheostat" mechanism can be differentiated by the old one.

Main comments:

- a) Is it possible to differentiate the rheostat mechanism that involved LSEC-CD8 T cell interactions from the previous observation (Nat Med 2000) that shows that it is the presentation of the antigens by LSEC to CD8 T cells that directly induces T cell tolerance? Are we talking about identical or different mechanisms? How can the authors ascertain that the inhibition of CD8 T cell function showed in this paper is not due to LSEC antigen-presentation to CD8 T cells?
- b) The CREM signature observed in HBc core specific CD8 T cells is interesting and supports the mouse model data. However, to demonstrate that this is specific for T cells recognizing viral antigen in the liver CD8 T cells of different specificities sorted in the same patients should be used as control. Furthermore, since CHB infection can occur in the presence or absence of liver inflammation (thus differentiating HBV infection with or without hepatitis) it might be important to understand whether the HBc-specific CD8 T cells were sorted from individuals with different or similar profile of infection and hepatitis. Is the CREM signature present in HBV-specific CD8 T cells irrespective of the presence/absence of liver inflammation? Overall the title is suggesting that such rheostat mechanism is occurring in chronic hepatitis.
- c) The novelty of the paper is about the mechanism of cAMP/PKA activation antigen-specific T cells. I am wondering whether instead of using activators, the inhibition of the cAMP/PKA pathway both upstream (prostaglandin synthesis in LSECs with prostaglandin synthesis inhibitors i.e. NSAIDs, etc.), or downstream with phosphodiesterase activation in CD8 T cells (or other relevant activators/inhibitors) can rescue CD8 functionality/prevent dysfunction in persistent infection.

Referee #2 (Remarks to the Author):

Nature 2022 A PKA-associated liver-tissue rheostat curbs T cell receptor signalling and effector function of virus-specific CD8 T cells in chronic viral hepatitis
M.Bosch et al

The immune-tolerogenic environment of the liver has been recognised for decades and defining the mechanisms of this potential could have significant relevance for the treatment of all autoimmune and allergic diseases as well as a better understanding of liver-based malignancies and chronic infections. Susceptibility of T lymphocytes (especially activated T lymphocytes) to enhanced death in the liver has long thought to be an important mechanism of liver mediated tolerance <https://pubmed.ncbi.nlm.nih.gov/10807506/>. But that is a different concept to the idea of modulating T cell function in a controlled fashion as proposed here and by others <https://aasldpubs.onlinelibrary.wiley.com/doi/full/10.1002/hep.21786>. The concept of a liver-specific rheostat for T cell activity is therefore attractive and of interest to many hepatologist and immunologists. [Whether this is of interest to chemists, physicists and astronomers as well as biologists, is up to the editor of Nature to decide.] However, having introduced the intriguing term 'rheostat' in the title <https://eepower.com/resistor-guide/resistor-types/rheostat/#>, it is not clearly defined in the paper. The term rheostat has also been used to describe a finely-tuned biological process, such as gene expression (<https://doi.org/10.1016/j.molcel.2006.09.002>) and protein kinase activity, but its use here is not fully developed nor supported by the data (although the data presented in Figure 4 and the graphical abstract is a beginning.)

This paper uses an elegant model of adenovirus driven acute and chronic viral infection of the liver to provide evidence that during persistent infection, viral-specific CD8+T cells establish close contact with liver sinusoidal endothelial cells, which then enforce cAMP/PKA phosphorylation in the T cells thereby stimulating transcription of cAMP response element modulator (CREM) which is associated with loss of T cell function. CREM has previously been shown to correlate with exhaustion in virus-specific CD8+ T lymphocytes in a paper in Blood from 2009

<https://ashpublications.org/blood/article/113/19/4575/25922/> as well as CD4+ T cell populations (<https://doi.org/10.1073/pnas.1603738113>). The paper from Sherman's lab at Scripps described in 2016 expression of the protein tyrosine phosphatase, PTPN22 and declared that it contributed to chronic viral infection by increasing the production of IFN- β following infection, resulting in increased expression of CREM and associated exhaustion in CD4 T lymphocytes. In that study, CREM prevented production of IL-2, thereby contributing to T-cell exhaustion and therefore chronic viral infection. The abstract states 'These findings implicate the IFN β /CREM/IL-2 axis in regulating T-lymphocyte function during chronic viral infection'. More recently, CREM expression has been shown to be associated with CD8 exhaustion and poor prognosis in a human GI malignancy <https://www.ncbi.nlm.nih.gov/pmc/articles/PMC8685542/>. However, there is recent evidence of PTPN22 expression in tissue resident memory cells in lung cancer <https://rupress.org/jem/article/216/9/2128/120725/Single-cell-transcriptomic-analysis-of-tissue> described in a paper where the hypothesis that high expression of PD-1 and other related markers of apparent exhaustion by T memory cells from the lung reflects tissue-residency rather than exhaustion, was explored. Bosch et al do not describe how these earlier findings are challenged and developed here although the liver seems to provide an important novel context.

Bosch et al use their previously described murine model of antigen specific acute and chronic viral infection of the liver to compare how specific liver T cell populations function in both scenarios. This mouse model uses two adenoviruses with two different promoters to drive viral gene expression and the outcome of infection

- a. a cytomegalovirus promoter (Ad-CMV-GOL) that leads to acute resolved infection of the liver and
- b. a hepatocyte-specific transthyretin promoter (Ad-TTR-GOL) that drives persistent infection of the liver'.

Markers expressed by hepatic CXCR6+CD8 T cells after resolved infection included GzmB, CXCR3, CCR2 and CCR5, and contrasted with enhanced expression of co-inhibitory receptors such as Pcd1, Lag3 and Tnfrsf9 by CXCR6+CD8+ T cells during persistent hepatic infection. Increased CREM expression is seen, as described by other groups, in the exhausted viral-specific CD8 T cells but this time during chronic infection of the liver. The possibility that close interaction between CD8T cells and LSECs might induce CREM expression by the T lymphocytes is novel and particularly intriguing. However, the authors show that CREM expression is not upstream of T cell exhaustion, but is rather a consequence of PKA signalling. They demonstrate that treatment of memory CXCR6+CD8 T cells with the cAMP-inducing agent forskolin, increased PKA (the upstream activator of CREM) and caused loss of cytotoxic effector function (Fig. 4m), similar to the loss of effector function in liver CXCR6+CD8 T cells during persistent infection. But they provide little evidence of any mechanism that might explain activation of PKA in the LSEC-adherent hepatic liver CXCR6+CD8. They hypothesise that PKA activation might be induced by prostanoids produced by LSECs which is an intriguing, plausible and exciting concept but not explored in any depth. Interferon-secreting liver-resident DCs might also be

important. It is disappointing that the authors do not pursue this line of investigation.

CXCR6 is a marker of liver residency. Is there something about CXCR6 expression (and the expression of other markers that are indicative of tissue residency like DR) that alters liver lymphocyte activation and makes liver lymphocytes more susceptible to cAMP/PKA signalling, CREM activation and ultimate exhaustion or apparent exhaustion? Do tissue resident populations of lymphocytes have different mechanisms of exhaustion to circulating populations? Are CXCR6 liver NK cells equivalently susceptible?

More clearly presented data defining exhaustion of hepatic T cells and the inhibition of their function in the two models are required. 'and hepatic accumulation of activated CD8 T cells that...'. Are CD8+ hepatic lymphocytes preferentially effected? What about liver-resident CD4 T cells? Liver-resident NK cells? It is not clear how the human data contribute to this core concept of the study 'circulating HBcore-specific CD8 T cells did not express CXCR6...'. Metabolic activity, inevitably driven by the microenvironment, is a key determinant of lymphocyte activation status, function and phenotypic marker expression. The striking differences in T cell numbers and LSEC density in the persistent versus resolved liver infections (Fig 4b) - presumably there's also associated stromal cell activation and fibrosis - are indicative of major differences in microenvironment likely to impact T cell metabolism. While metabolomics may be beyond the scope of this research programme, additional probing of the NGS data may well be informative. The recent paper by Fisher et al on 'Modelling intercellular communication in tissues using spatial graphs of cells' <https://www.nature.com/articles/s41587-022-01467-z> may be informative'

What is it about the model that the effects are only seen with adenovirus and not with other viruses?

Other comments

This paper seems to have been assembled in haste, without adequate curation and editing. Nine figures of extended experiments and data (in addition to the figures provided for the paper) are indeed important for conveying the amount of work that has gone into this study, but still seem excessive for a Nature paper which presumably is aiming to communicate a single 'paradigm shifting' concept. Should the reader have to work so hard to extract the key points from the terabytes of data provided to identify the paradigm shift?

Typographical errors are additional indication of the haste

Line 60 'explaining their loss of effector.'[sic]

Line 99 ...'only emerged only' [sic]

Line 100: 'Thus, this model system reflects the scarcity of virus-specific CD8 T cells during chronic HBV infection and hepatic accumulation of activated CD8 T cells that fail to clear virus-infected hepatocytes.' This is a key observation but where is the figure? If previously published, it needs to be referenced.

Line 138: 'Few genes were mutually exclusively expressed and characterized liver CXCR6+CD8 T cells after resolved infection, such as Gzmb, Itga1 and Gadd45b, or liver CXCR6+CD8 T cells during persistent infection, such as Crem and the TGF β -dependent transcription factors Tgif1 and Ski' Awkward sentence.

Line 172: 'in killing of hepatocytes'.

The discussion is merely a brief summary of the main findings of the paper; no attempt is made to place the data into the context of the title or previous discoveries.

Some key references seem to be missing [see above].

Referee #3 (Remarks to the Author):

Title: "A PKA-associated liver-tissue rheostat curbs T cell receptor signalling and effector function of virus-specific CD8 T cells in chronic viral hepatitis " by Percy Knolle and colleagues.

In the present manuscript by Percy Knolle and colleagues, the authors intended to understand the molecular mechanisms underlying exhaustion of virus-specific CD8 T cells during persistent hepatotropic hepatitis B virus (HBV) infection.

Validity

Data presented in this paper are thoroughly analyzed and statistical testing is performed where indicated. Data are not over-interpreted, although additional experiments should be considered to further understand the cAMP-steered pathways involved and to substantiate the HBV- and liver specificity of their findings. In general, there is no concern about the validity of the findings presented in this manuscript.

Significance

The presented study is highly significant for the field of viral immune evasion mechanisms and especially HBV and chronic viral hepatitis. Supportive for this finding are also older publications, in which the cAMP and PKA pathways have already been shown to be implicated in exhausted T cells of human immunodeficiency virus (HIV)-infected patients. Also, T cells from HIV-infected patients show elevated levels of cAMP and hyperactivation of PKA. A comparable mechanism also contributes to T cell dysfunction in a subset of patients with common variable immunodeficiency, and to an anergy-like phenotype of T cells observed in a murine model termed MAIDS.

Data and methodology

Data presented in this study are of high quality, experimental repeats have been performed as required.

Analytical approach

More emphasis should be put on understanding the exact molecular mechanism of how cAMP is mediating these effects.

Suggested improvements

Major criticism

1. In Figure 1 and the corresponding Extended Data Figure 1 the authors provide a canonical tissue-residency gene signature of liver CXCR6+ CD8 T cells, indicative of the involvement of cAMP-mediated signal transduction. Can this gene signature also be found in CD8+ T cells of HBV-infected patients? Is this canonical tissue-residency gene signature differentially expressed in HBcoreCD8 T cells, that completely lost cytokine expression when compared to liver CD8 T cells in general?
2. Cyclic AMP-mediated signal transduction was also shown to be involved in dysfunction of T cells in other viral infections. For example, T cells from human immunodeficiency virus (HIV)-infected patients show elevated levels of cAMP and hyperactivation of PKA. Targeting of the cAMP-PKA type I pathway by selective antagonists reverses T cell dysfunction in HIV T cells *ex vivo*^{1,2}. A similar mechanism contributes to T cell dysfunction in a subset of patients with common variable immunodeficiency³, and to promote an anergy-like T cell phenotype in a murine immunodeficiency model termed MAIDS⁴. Hence, the authors should be encouraged to analyze to what extent enrichment of these genes can also be found in already published data sets of dysfunctional T cells from other viral infections like HIV in order to substantiate HBV infection- and tissue-specificity of this immune evasion program.
3. In Fig. 4e,f and Extended Data Fig. 9f the authors provide evidence of enhanced PKA activity in CXCR6+ CD8 T cells during persistent infection. To substantiate the involvement of PKA the authors should be encouraged to design experiments e.g. comparable to the adoptive transfer experiment described and pretreat the adoptively transferred cells with a PKA inhibitor like Rp-cAMPS5 or any other PKA inhibitor like H89 and compare the phenotype of those cells to endogenous non-treated CD8+ T cells.
4. In the same vein, have the authors analyzed involvement of other cAMP-dependent but PKA-independent mechanisms like EPACs^{6,7}, which have been shown to be involved in the TCR-dependent activation of T cells.
5. In order to underline the importance of cAMP in the observed T cell phenotype the authors conducted experiments involving Forskolin treatment to increase cAMP. However, the authors should also be encouraged to use adenylyl cyclase inhibitors like MDL-128,9 to demonstrate the importance of *de novo* cAMP production *in vitro* as well as *in vivo*.
6. As a possible underlying molecular mechanism, the authors suggested that LSEC-derived prostanoids might stimulate *de novo* production of cAMP in T cells and induce the observed phenotype. If this holds true, COX-2 inhibitors or prostanoid antagonists should revert the observed T cell phenotype and, in best case, prevent chronification after infection with HBV. Hence, the authors should be encouraged to use e.g. COX-2 inhibitors in the models described.
7. In the same sense, have the authors considered other cAMP-inducing mechanisms like adenosine-mediated inhibition of T cell function via receptors A2B and A2B10 or transfer via gap junctions¹¹.

Minor criticism:

8. In Fig. 1l and Extended Data Fig. 2i the authors suggest involvement of Tox. Are T cells from Tox-deficient mice refractory to the described mechanism?

Clarity and context

The text is written in a clear and accessible way. Some typos are still present.

References

Relevant literature has been cited.

Literature

1. Aandahl, E.M., Aukrust, P., Müller, F., Hansson, V., Taskén, K., and Frøland, S.S. (1999). Additive effects of IL-2 and protein kinase A type I antagonist on function of T cells from HIV-infected patients on HAART. *Aids* 13, 109–114. [10.1097/00002030-199912030-00001](https://doi.org/10.1097/00002030-199912030-00001).
2. Aandahl, E.M., Aukrust, P., Skålhegg, B.S., Müller, F., Frøland, S.S., Hansson, V., and Taskén, K. (1998). Protein kinase A type I antagonist restores immune responses of T cells from HIV-infected patients. *Faseb J* 12, 855–862. [10.1096/fasebj.12.10.855](https://doi.org/10.1096/fasebj.12.10.855).
3. Aukrust, P., Aandahl, E.M., Skålhegg, B.S., Nordøy, I., Hansson, V., Taskén, K., Frøland, S.S., and Müller, F. (1999). Increased activation of protein kinase A type I contributes to the T cell deficiency in common variable immunodeficiency. *J Immunol Baltim Md 1950* 162, 1178–1185.
4. Rahmouni, S., Aandahl, E.M., Trebak, M., Boniver, J., Taskén, K., and Moutschen, M. (2001). Increased cAMP levels and protein kinase (PKA) type I activation in CD4+ T cells and B cells contribute to the retrovirus-induced immunodeficiency of mice (MAIDS). A useful in vivo model for drug testing in PKA type I-induced immunodeficiency. *Faseb J* 15, 1466–1468. [10.1096/fj.00-0813fje](https://doi.org/10.1096/fj.00-0813fje).
5. Dostmann, W.R.G. (1995). (R P)-cAMPS inhibits the cAMP-dependent protein kinase by blocking the cAMP-induced conformational transition. *Febs Lett* 375, 231–234. [10.1016/0014-5793\(95\)01201-o](https://doi.org/10.1016/0014-5793(95)01201-o).
6. Rooij, J. de, Zwartkruis, F.J.T., Verheijen, M.H.G., Cool, R.H., Nijman, S.M.B., Wittinghofer, A., and Bos, J.L. (1998). Epac is a Rap1 guanine-nucleotide-exchange factor directly activated by cyclic AMP. *Nature* 396, 474–477. [10.1038/24884](https://doi.org/10.1038/24884).
7. Enserink, J.M., Christensen, A.E., Rooij, J. de, Triest, M. van, Schwede, F., Genieser, H.G., Døskeland, S.O., Blank, J.L., and Bos, J.L. (2002). A novel Epac-specific cAMP analogue demonstrates independent regulation of Rap1 and ERK. *Nat Cell Biol* 4, 901–906. [10.1038/ncb874](https://doi.org/10.1038/ncb874).
8. Klein, M., Vaeth, M., Scheel, T., Grabbe, S., Baumgrass, R., Berberich-Siebelt, F., Bopp, T., Schmitt, E., and Becker, C. (2012). Repression of Cyclic Adenosine Monophosphate Upregulation Disarms and Expands Human Regulatory T Cells. *J Immunol* 188, 1091–1097. [10.4049/jimmunol.1102045](https://doi.org/10.4049/jimmunol.1102045).
9. Johann, K., Bohn, T., Shahneh, F., Luther, N., Birke, A., Jaurich, H., Helm, M., Klein, M., Raker, V.K., Bopp, T., et al. (2021). Therapeutic melanoma inhibition by local micelle-mediated cyclic nucleotide repression. *Nat Commun* 12, 5981. [10.1038/s41467-021-26269-w](https://doi.org/10.1038/s41467-021-26269-w).
10. Mastelic-Gavillet, B., Rodrigo, B.N., Décombaz, L., Wang, H., Ercolano, G., Ahmed, R., Lozano, L.E., Ianaro, A., Derré, L., Valerio, M., et al. (2019). Adenosine mediates functional and metabolic

suppression of peripheral and tumor-infiltrating CD8+ T cells. *J Immunother Cancer* 7, 257.
10.1186/s40425-019-0719-5.

11. Bopp, T., Becker, C., Klein, M., Klein-Hessling, S., Palmethofer, A., Serfling, E., Heib, V., Becker, M., Kubach, J., Schmitt, S., et al. (2007). Cyclic adenosine monophosphate is a key component of regulatory T cell-mediated suppression. *The Journal of experimental medicine* 204, 1303–1310.
10.1084/jem.20062129.

Point-by-point reply to the reviewers' questions

Referee #1 (Remarks to the Author):

Bosch and Kallin et al. studied the phenotype and function of antigen-specific CD8 T cells mainly in in vivo (mouse) models of hepatotropic viral infection, and substantiated some links in human chronic HBV in PBMCs. Using both OVA- and HBV-based hepatotropic infection models, and contrasting with systemic (LCMV) infection models, they demonstrated features of antigen-specific T cells unique to liver-restricted persistent infection, including their almost-exclusive hepatic localization, CXCR6+CX3CR1± phenotype, dysfunctionality for effector cytokine and cytolytic capacity, and Crem transcriptional network/signature enrichment. Furthermore, to explain these features, they identified increased LSEC-CD8 T cell interactions using confocal microscopy in persistent infection, with OVA-presenting LSEC/CD8 T cell co-culture demonstrating loss of IFN γ production in restimulation. Mechanistically the cAMP/PKA axis was identified downstream, with the functional phenotype of persistent infection recapitulated in T cells from resolved infection by in vitro pharmacological activation of cAMP/PKA. In humans, they find CREM among the top transcription factors in circulating HBcore-specific CD8 T cells.

Reviewer: The work is technically well executed and the characterization of the mechanisms of the hepatic CD8 T cell dysfunction is novel and detailed. On the other hand, for a paper that is designed to analyze the cause of CD8 T cell dysfunction present in chronic HBV infection, the data related to real HBV infection is very limited. Understandably, due to the infeasibility of the direct use of HBV in mice, the authors used an HBV-Adenoviral system to mimic HBV infection. They then analyzed HBcore-specific CD8 T cells sorted from only 3 patients, who were poorly characterized in terms of HBV chronic stage. While I understand that there are technical limitations to obtain HBV-specific T cells in CHB patients, particularly from the liver, I am still not convinced that this study can prove that the mechanism shown in different models of persistence antigen presentation in the liver accurately mimics what is occurring in HBV infection.

Answer: We thank the reviewer for this positive evaluation of our findings. We have followed the suggestion and extended the analysis of HBV-specific T cells in chronic hepatitis B in patients.

We fully agree with the reviewer that chronic hepatitis B in patients is a disease that cannot be mirrored entirely in animal models. Therefore, we have made every possible effort to extend our analysis to human samples from chronic hepatitis B patients. First, we analysed liver sections from patients with chronic hepatitis B compared to healthy controls. In these liver tissue sections, we found an enrichment of CXCR6⁺ T cells in chronic hepatitis B patients (mean=5% of all CD3⁺ T cells) as compared to less than <1% in patients without HBV infection (**Fig. 1a**), validating our findings in the preclinical models of chronic hepatitis B.

Second, we recruited further five HLA-A2⁺ patients with chronic hepatitis B and isolated HLA-A2-restricted HBcore-specific CD8 T cells from peripheral blood for RNAseq analysis. Following the reviewer's suggestion we also isolated non-HBV-specific CD8 T cells as controls. We were also able to isolate HLA-A2-restricted HBcore-specific CD8 T cells from two HLA-A2⁺ patients who had resolved chronic hepatitis B infection and established immune control of infection.

As requested by the reviewer, we now included the relevant clinical data of the patients with chronic hepatitis B (**Fig. 1b-e**), with **Fig. 1b,c** showing the data from the patients that were already included in the original manuscript, and **Fig. 1d,e** showing the clinical data from the further five patients with chronic hepatitis B. All patients were untreated, had low to absent liver inflammation and had no evidence of advanced liver disease shown by normal bilirubin levels and low fibroscan results (**Table 1 and Table 2**). Thus, there is no evidence of ongoing liver inflammation in the patients with chronic hepatitis B, from whom the circulating HBcore-specific CD8 T cells were isolated, and all patients analysed here had a Child-Pugh A liver score (**Table 1 and Table 2**). Similar conditions with the absence of liver inflammation and no significant liver fibrosis were present in the animal models of liver infection used here. All these data are shown in the new Fig. 3 and Extended Data Fig. 4 in the revised manuscript.

Legend to Figure 1

a, CXCR6 gene expression measured by RNA scope (purple) in CD3⁺ T cells identified by immunohistochemistry (red) in livers of chronic hepatitis B patients (cHBV) and uninfected patients (n=5, scale bar 20 μm). **b,c**, Viral and host parameters in serum from patients with chronic hepatitis B (n=3, scRNAseq analysis); **d,e**, Viral and host parameters in the serum of patients with chronic hepatitis B (n=5, SMART-Seq2 analysis).

patient ID	age	diagnosis	therapy	viral load [IU/ml]	HBeAg [PEIU/mL]	anti-HBe [PEIU/mL]	HBsAg [PEIU/mL]	anti-HBs [PEIU/mL]	ALT [U/ml]	AST [U/ml]	g-GT [U/l]	bilirubin [mg/dl]	albumin [g/dl]	fibrosis scan [kPa]
1	29	cHBV	none	31	neg	nd	nd	neg	19	23	10	0.3	4.7	NA
2	53	cHBV	none	1795	neg	nd	nd	neg	46	31	27	0.3	4.7	5
3	44	cHBV	none	915	neg	nd	nd	neg	28	nd	15	0.3	4.7	4.9

Table 1: Viral and host parameters in the serum of patients with chronic HBV infection whose HBcore-specific CD8 T cells were analysed by scRNAseq analysis.

patient ID	age	diagnosis	diagnosis date	therapy	HBV DNA [IU/mL]	HBsAg [IU/mL]	HBeAg [PEIU/mL]	HBV genotype	cell count	g-GT [U/L]	bilirubin [mg/dL]	albumin [g/dL]	fibrosis scan [kPa]
P11	53	cHBV	2011	none	378	NA	NA	A	331	31	1.2	4.8	5.3
P12	47	cHBV	perinatal?	none	1077	800.97	<0.01/neg	D	4814	15	0.4	4.7	4.3
P13	53	cHBV	2001	none	29	5.94	<0.01/neg	NA	856	20	1.30	4.7	3.3
P14	47	cHBV	perinatal?	none	1077	800.97	<0.01/neg	D	213	15	0.4	4.7	4.3
P15	56	cHBV	2005	none	2045	1446.83	0.01/neg	D	966	22	0.3	4.7	3.10
P24	41	rHBV	NA	NA	<10	<0.01/neg	NA	NA	252	55	0.7	4.50	5.9
P25	57	rHBV	unknown	NA	<10	NA	NA	NA	577	NA	NA	NA	3.10

Table 2: Viral and host parameters in the serum of patients with resolved or chronic HBV infection whose HBcore-specific CD8 T cells were analysed by SMART-Seq2 analysis.

The new RNAseq data from HBcore-specific CD8 T cells confirmed the presence of a cAMP/CREM signature in HBcore-specific CD8 T cells in patients with chronic hepatitis B compared to patients with resolved HBV infection and demonstrated that this signature is not present in non-HBV-specific bulk CD8 T cells from the same patients (**Fig. 2a,b**). These results are now shown in the revised manuscript as new Figures 3 and in Extended Data Figure 4. Together, we now demonstrate the regulation of HBcore-specific CD8 T cells by cAMP/CREM signalling in a cohort of 8 patients with chronic hepatitis B. Although we cannot formally exclude that other factors present in patients with chronic HBV infection influence virus-specific CD8 T cells, the similarities observed between virus-specific CD8 T cells isolated from patients and from preclinical models of persistent HBV infection strongly suggest that similar mechanisms are operative in the control of virus-specific T cell function.

Legend to Figure 2

a,b, Gene set enrichment analysis (GSEA) of HBcore-specific CD8 T cells or non-HBV-specific bulk CD8 T cells from patients with chronic HBV infection (n=5) compared to resolved HBV infection (n=2), revealing enrichment of a cAMP/CREM signature. Please note that the cut-off for statistical significance of GSEA is at FDR.q 0.25 (see <https://software.broadinstitute.org/cancer/software/gsea/wiki/index.php/FAQ>)

Reviewer: *The authors also claim that this manuscript identifies a new “liver tissue rheostat that induces loss of the T cell function”, but this reviewer remains doubtful of the differences between this “novel” phenomena and the mechanism of hepatic CD8 T cell dysfunction described by the same authors 20 years ago in their paper “Efficient presentation of exogenous antigen by liver endothelial cells to CD8+ T cells results in antigen-specific T-cell tolerance” [Nature Medicine 6 (2000), 12. - S. 1348-1354 <https://dx.doi.org/10.1038/82161>]; particularly whether we are talking of the same or different phenomena, and as such, how this “new liver tissue rheostat” mechanism can be differentiated by the old one.*

Main comments:

a) Is it possible to differentiate the rheostat mechanism that involved LSEC-CD8 T cell interactions from the previous observation (Nat Med 2000) that shows that it is the presentation of the antigens by LSEC to CD8 T cells that directly induces T cell tolerance? Are we talking about identical or different mechanisms? How can the authors ascertain that the inhibition of CD8 T cell function showed in this paper is not due to LSEC antigen-presentation to CD8 T cells?

Answer: We thank the reviewer for this question, which we have addressed in several additional experiments.

First, we would like to point out that cross-presentation of HBV antigens on MHC class I, particularly HBcore antigen, by LSECs has been excluded in recent publications by the Iannacone and Guidotti Labs in *Cell* in 2015 and *Nature* 2019, ref^{1,2}. We confirmed these results by performing experiments with primary LSECs isolated from mouse livers to high purity (**Fig. 3a**). Hepatocytes infected with Ad-HBV *in vitro* stimulated HBcore-specific CD8 T cells, shown by increased expression of IFN γ from these T cells (**Fig. 3b**). We next incubated LSECs with the supernatant of these Ad-HBV-infected primary mouse hepatocytes that replicate HBV and release viral antigens into the supernatant as a possible source of viral antigens for cross-presentation by LSECs. We did not detect cross-presentation of HBcore antigen to HBcore-specific CD8 T cells by LSECs as demonstrated by the absence of T cell cytokine expression (**Fig. 3b**). Together, with the previously published data^{1,2}, our results exclude a role of HBcore cross-presentation by LSECs to HBcore-specific CD8 T cells during persistent replication of HBV in hepatocytes, and therefore also excludes a role for tolerogenic antigen-presentation by LSECs to influence T cell function.

Legend to Figure 3

a, Purity of isolated primary mouse LSECs demonstrated by their characteristic scavenger activity to take up AcLDL and expression of the LSEC-defining marker CD146. **b**, IFN γ expression as a measure of cognate stimulation in HBc-specific CD8 T cells cocultured with Ad-HBV or mock-infected hepatocytes or LSECs pre-incubated with supernatant of Ad-HBV or mock-infected hepatocytes (n=5, two-way ANOVA with Tukey's multiple comparison). **c-e** Cytokine expression and quantification by CD45.1⁺ CD8 T cells and serum ALT levels of mice on d0 before and d3 after transfer of *in vitro* activated effector CD8 T cells into mice with resolved Ad-CMV-GOL or persistent Ad-TTR-GOL infection (n=4, two-way ANOVA with Sidak's multiple comparison, two independent experiments). c-e: one out of two independent experiments shown.

A second important difference to our *Nature Medicine* publication from 2000 is that we now demonstrate that LSECs affect the functional capacity of previously activated effector CD8 T cells. In 2000, we demonstrated that priming of naïve CD8 T cells by cross-presenting LSECs led to T cell non-responsiveness. Thus, the inhibition of effector CD8 T cells' function through the liver tissue-rheostat that we report here must occur via a mechanism distinct from the previously reported antigen-dependent induction of T cell non-responsiveness.

To investigate the influence of LSECs on the function of effector CD8 T cells, we cocultured primary LSECs with effector CD8 T cells. To enhance the interaction of CD8 T cells with LSECs in the absence of cognate antigen recognition, we pre-stimulated LSECs with IFN γ to increase expression of adhesion molecules known to mediate strong adhesion of T cells. In effector CD8 T cells cocultured with LSECs and engaging in strong interaction, we found increased phosphorylation of PKA (**Fig. 4a,b**). Moreover, we detected a downregulation of GzmB in those CD8 T cells engaging in strong contact with LSECs (**Fig. 4c,d**). Mechanistically, we identified increased adenylyl cyclase activity causing downregulation of GzmB expression in CD8 T cells, because irreversible inhibition of adenylyl cyclase activity by MDL-12,330A in effector CD8 T cells rescued them from downregulation of GzmB during coculture with LSECs (**Fig. 4e**). Thus, the results from these experiments have uncovered the key role of adenylyl cyclase activation in regulating the function of effector CD8 T cells by LSECs. This is a novel finding that has not been reported before.

Legend to Figure 4

a,b, Change in pPKA levels in CD8 T cells cocultured with liver cells (n=4, two-way ANOVA with Tukey's multiple comparison). **c,d**, GzmB expression by T cells in coculture with LSECs and quantification (n=3, unpaired t-test). **e**, GzmB expression and quantification by CD8 T cells cocultured with activated LSECs for 24 h after 1 h pre-treatment of T cells with MDL-12330A (100 μ M, MDL), solvent control, or medium and T cells without LSEC contact (one-way ANOVA with Dunnett's multiple comparison, n=4). a-e: one out of \geq two independent experiments shown.

Together, these new results strongly suggest that enhanced cAMP signalling is involved downregulating the function of effector CD8 T cells by the liver tissue-rheostat independent from antigen presentation by LSECs.

Reviewer: b) The CREM signature observed in HBcore specific CD8 T cells is interesting and supports the mouse model data. However, to demonstrate that this is specific for T cells recognizing viral antigen in the liver CD8 T cells of different specificities sorted in the same patients should be use as control. Furthermore, since CHB infection can occur in the presence or absence of liver inflammation (thus differentiating HBV infection with or without hepatitis) it might be important to understand whether the HBc-specific CD8 T cells were sorted from individuals with different or similar profile of infection and hepatitis. Is the CREM signature present in HBV-specific CD8 T cells irrespective of the presence/absence of liver inflammation? Overall, the title is suggesting that such rheostat mechanism is occurring in chronic hepatitis.

Answer: We thank the reviewer for these comments that helped us to sharpen the concept of the liver tissue-rheostat through additional experiments. As already outlined in answer 1, we have

performed additional experiments isolating non-HBV-specific CD8 T cells for RNAseq analysis from patients with chronic hepatitis B and further analysed HBV-specific CD8 T cells as well as non-HBV-specific CD8 T cells from patients with resolved hepatitis B. These results demonstrated that a cAMP/Crem signature was only present in HBcore-specific CD8 T cells but not in bulk CD8 T cells from the same patient during persistent hepatic infection (*please, see Fig. 2*).

We understand the reviewer's remark on the relevance of liver inflammation. All chronic hepatitis B patients in this study had only slightly elevated or normal serum ALT levels and we do not have evidence for late-stage liver disease (Child-Pugh score B or C) (*please, see Fig. 1b-e and Tables 1 and 2*). Also, in the preclinical model system of persistent hepatotropic infection using Ad-HBV that we studied here, we detected slightly elevated sALT levels without clear histological evidence of liver inflammation (**Fig. 5a,b**). However, the elevated sALT levels in persistently infected mice did not reach statistical significance.

Legend to Figure 5

a, sALT in mice with resolved 10^7 or persistent 10^8 IU Ad-HBV infection ($n \geq 4$, two-way ANOVA with Tukey's multiple comparison). **b**, H/E staining of liver sections from mice with resolved or persistent infection on d45 p.i. ($n \geq 4$). a-b: one out of ≥ 2 independent experiments shown.

Thus, all data presented in this manuscript show a distinctive cAMP/CREM signature under similar conditions in patients with chronic hepatitis B and in preclinical models of chronic HBV infection. We fully acknowledge the notion of the reviewer that the situation is different when liver inflammation is present, which may either lead to changes in liver tissue-rheostat function by changes in the microenvironment or may be related to large numbers of effector T cells that enter the liver after being generated in lymphoid tissues and recruitment to inflamed liver tissue. In both situations, one might assume that the function of the liver tissue-rheostat may be affected, but we cannot provide any formal evidence for this. We have carefully considered this aspect in the discussion of the revised manuscript.

Reviewer: c) The novelty of the paper is about the mechanism of cAMP/PKA activation antigen-specific T cells. I am wondering whether instead of using activators, the inhibition of the cAMP/PKA pathway both upstream (prostaglandin synthesis in LSECs with prostaglandin synthesis inhibitors i.e. NSAIDs, etc.), or downstream with phosphodiesterase activation in CD8 T cells (or other relevant activators/inhibitors) can rescue CD8 functionality/prevent dysfunction in persistent infection.

Answer: We thank the reviewer for these suggestions, that prompted us to perform further experiments to increase the mechanistic understanding of the liver tissue-rheostat. As suggested by the reviewer, we performed experiments to investigate upstream and downstream mechanisms of cAMP signalling and their impact on the regulation of effector T cell function.

We addressed the question raised by the reviewer, which agents might induce increased cAMP signalling in T cells that are in close and extensive contact with LSECs. Purinergic signalling, particularly signalling through adenosine receptors, is a potent inducer of cAMP signalling³. Although LSECs constitutively expressed the ectonucleotidase CD73 (**Fig. 6a,b**), that is relevant for the breakdown of AMP to adenosine and subsequent adenosine receptor signalling, they did not express CD39 (**Fig. 6a,b**) an ectonucleotidase cleaving extracellular ATP to ADP and AMP⁴. Thus, LSECs cannot stimulate adenosine receptor signalling from extracellular ATP, suggesting that adenosine receptor signalling was not involved in LSEC-induced regulation of T cell function. Consequently, when we blocked adenosine receptor signalling with a specific pharmacological inhibitor, we did not detect an increase in GzmB expression by T cells in coculture with LSECs (**Fig. 6c,d**). Together, these results did not support a role of adenosine receptor signalling in LSEC-induced control of CD8 T cell effector function.

In contrast, we found that LSECs produced large amounts of PGE₂ (**Fig. 6e**), a known inducer of cAMP signalling⁵. Inhibiting cyclooxygenase activity with the selective inhibitor Celecoxib increased GzmB expression in T cells cocultured with LSECs *in vitro* (**Fig. 6f,g**), indicating a contribution of PGE₂ to LSEC-mediated control of effector CD8 T cell function.

Moreover, a recent publication from Pallett et al. in Nature demonstrated transfer of molecules between CD8 T cells and macrophages in the liver⁶, which prompted us to look into transfer of molecules during cell-cell interaction between CD8 T cells and LSECs. When we labelled LSECs with the dye CFSE that binds to intracellular molecules, we detected LSEC-derived CFSE-labelled molecules in T cells (**Fig. 6h,i**), suggesting a transfer of molecules between LSECs and effector T cells. Since LSECs are defined by cAMP-agonism, that is responsible for their differentiation and function⁷, our results may indicate a possible transfer of adenylyl cyclase molecules from LSECs to T cells that might enhance cAMP signalling in T cells. However, we cannot provide formal evidence for involvement of an intercellular molecule exchange and therefore did not include these data into the revised manuscript.

Legend to Figure 6

a,b, Expression of the ectonucleotidases CD73 and CD39 on LSECs after isolations and quantification ($n \geq 4$). **c,d**, GzmB expression by CD8 T cells activated for 3 days *in vitro* followed by 1 h pre-treatment with SCH58261 (100 nM), solvent control or medium control and subsequent cocultured with LSECs for 24 h ($n=4$, one-way ANOVA with Dunnett's multiple comparisons). **e** PGE₂ secretion of LSECs, hepatocytes and DCs over 24 h measured by ELISA ($n \geq 4$, one-way ANOVA with Tukey's multiple comparison). **f,g**, GzmB expression and quantification by CD44⁺CD8⁺ T cells cocultured treated with LSECs with celecoxib (1 μ M), acetylsalicylic acid (100 μ M, ASS) or medium control treatment ($n=5$, one-way ANOVA with Tukey's multiple comparison). **j,k**, CFSE incorporation and quantification by CD8 T cells cocultured with LSECs labelled with CFSE before cocultured ($n=5$, one-way ANOVA with Tukey's multiple comparison). a-i: one out of ≥ 2 independent experiments shown.

Next, we addressed the reviewer's question about which signalling pathways were induced downstream of cAMP in T cells. Two pathways downstream of cAMP signalling were described in T cells, signalling via PKA or the exchange protein directly activated by cAMP EPAC⁸. We addressed both pathways by using selective pharmacological inhibitors. Only inhibition of PKA but not EPAC prevented the downregulation of GzmB expression in T cells cocultured with LSECs (**Fig. 7a-d**). Consequently, only agonism of PKA but not EPAC signalling inhibited T cell function *in vitro* (**Fig. 7e**). We conclude from these results that enhanced cAMP activated PKA-signalling to inhibit T cell effector function.

Legend to Figure 7

a,b, GzmB expression by CD8 T cells cocultured with LSECs for 24 h after 1 h pre-treatment with Rp-8-Br-cAMPS (1 mM, PKA antagonist) or solvent control ($n=4$, unpaired t-test). **c,d**, GzmB expression by CD8 T cells cocultured with LSECs for 24 h after 1 h pre-treatment with ESI-09 (10 μ M, EPAC antagonist), solvent or medium control ($n=4$, one-way ANOVA with Dunnett's multiple comparison). **e**, IFN γ ⁺TNF⁺ CD8 T cells (peptide-stimulated normalised to medium control) after 4 h pre-treatment of *in vitro* activated CD8 T cells with Sp-8-Br-cAMPS (250 μ M, PKA agonist), 8-pCPT-2'-O-Me-cAMP (30 μ M, EPAC agonist), or solvent

control followed by 15 h peptide restimulation (n=3, one-way ANOVA with Dunnett's multiple comparisons). a-e: one out of \geq two independent experiments shown.

Finally, to address the reviewer's comment on distinct mechanisms operative in modulating effector CD8 T cell function through the liver tissue-rheostat, we transferred antigen-specific effector CD8 T cells loaded with an irreversible adenylyl cyclase inhibitor into mice with persistent hepatic infection. Compared to mock-treated CD8 T cells, these adenylyl cyclase inhibitor-treated CD8 T cells had lower levels of phosphorylated PKA and higher levels of GzmB at 72h after transfer (**Fig. 8a-c**). Moreover, in mice with a persistent hepatic infection that received adenylyl cyclase inhibitor-treated CD8 T cells, we detected an increase of serum ALT (**Fig. 8d**), indicating that adenylyl cyclase inhibitor-treated CD8 T cells escaped the liver tissue-rheostat and executed effector function *in vivo*. All these results were integrated into the revised manuscript.

Legend to Figure 8

a, Re-isolation of CD45.1⁺ CD8 T cell after transfer of MDL-12,330A (100 μ M, MDL)- or mock-treated CD45.1⁺ *in vitro*-activated CD8 T cells into mice with resolved Ad-CMV-GOL or persistent Ad-TTR-GOL infection; **b,c** pPKA and GzmB levels and quantification and **d**, increase of serum ALT from d0 to d3 per liver CD45.1⁺ CD8 T cell (n \geq 4, one-way ANOVA with Tukey's multiple comparison). a-d: one out of \geq two independent experiments shown.

Together, these results provide evidence that enhanced adenylyl cyclase and PKA activity was involved in downregulating the effector function of antigen-specific CD8 T cells in the liver during persistent infection.

Referee #2 (Remarks to the Author):

Reviewer: *The immune-tolerogenic environment of the liver has been recognised for decades and defining the mechanisms of this potential could have significant relevance for the treatment of all autoimmune and allergic diseases as well as a better understanding of liver-based malignancies and chronic infections. Susceptibility of T lymphocytes (especially activated T lymphocytes) to enhanced death in the liver has long thought to be an important mechanism of liver-mediated tolerance (<https://pubmed.ncbi.nlm.nih.gov/10807506/>). But that is a different concept to the idea of modulating T cell function in a controlled fashion as proposed here and by others (<https://aasldpubs.onlinelibrary.wiley.com/doi/full/10.1002/hep.21786>). The concept of a liver-specific rheostat for T cell activity is therefore attractive and of interest to many hepatologists and immunologists. [Whether this is of interest to chemists, physicists, and astronomers as well as biologists, is up to the editor of Nature to decide.] However, having introduced the intriguing term 'rheostat' in the title (<https://eepower.com/resistor-guide/resistor-types/rheostat/#>), it is not clearly defined in the paper. The term rheostat has also been used to describe a finely-tuned biological process, such as gene expression (<https://doi.org/10.1016/j.molcel.2006.09.002>) and protein kinase activity, but its use here is not fully developed nor supported by the data (although the data presented in Figure 4 and the graphical abstract is a beginning.)*

Answer: We thank the reviewer for these comments. We have followed this reviewer's suggestion to explain in more detail what is meant by the term tissue-rheostat. We did not mean to obscure the technical term rheostat that is used for describing regulation of electrical resistance, as pointed out by the reviewer. We rather want to refer to the dynamic process of regulating T cell effector function by tissue cells. To this end, we performed a set of experiments, in which we isolated virus-specific CD8 T cells from livers after resolved infection or during persistent infection and transferred these cells into mice with persistent infection or acute resolving infection, respectively (**Fig. 1a**). Fully functional virus-specific effector CD8 T cells lost GzmB expression when transferred into mice with persistent infection (**Fig. 1b,c**). In contrast, dysfunctional virus-specific CD8 T cells isolated at d30 of persistent infection regained GzmB expression when transferred into mice with an acute-resolving infection (**Fig. 1b,c**). This demonstrated that loss of GzmB expression in virus-specific CD8 T cells as surrogate marker for their effector function was reversible.

Legend to Figure 1

a, Experimental scheme illustrating the isolation of liver CXCR6⁺CD45.1⁺ CD8 T cells at d30 p.i. from mice with resolved or persistent infection that were transferred into mice with Ad-CMV-GOL resolving infection or into mice infected with Ad-TTR-GOL developing a persistent infection (d2 p.i., one-way ANOVA with Tukey's multiple comparison, n≥3). **b,c**, GzmB expression by liver CXCR6⁺CD45.1⁺ CD8 T cells from resolved

or persistent infection (d30 p.i.), transferred into mice with acute resolving or persistent infection (d2 p.i.) and analysed at d30 after transfer ($n \geq 3$, one-way ANOVA with Tukey's multiple comparison). b-c: one out of \geq two independent experiments shown.

We have identified that enhanced adenylyl cyclase activity and cAMP-induced PKA signalling are responsible for the loss of GzmB expression and effector CD8 T cell function by LSECs, which is outlined in detail in the answers given below.

***Reviewer:** This paper uses an elegant model of adenovirus driven acute and chronic viral infection of the liver to provide evidence that during persistent infection, viral-specific CD8+T cells establish close contact with liver sinusoidal endothelial cells, which then enforce cAMP/PKA phosphorylation in the T cells thereby stimulating transcription of cAMP response element modulator (CREM) which is associated with loss of T cell function. CREM has previously been shown to correlate with exhaustion in virus-specific CD8+ T lymphocytes in a paper in Blood from 2009 (<https://ashpublications.org/blood/article/113/19/4575/25922/>) as well as CD4+ T cell populations (<https://doi.org/10.1073/pnas.1603738113>). The paper from Sherman's lab at Scripps described in 2016 expression of the protein tyrosine phosphatase, PTPN22 and declared that it contributed to chronic viral infection by increasing the production of IFN- β following infection, resulting in increased expression of CREM and associated exhaustion in CD4 T lymphocytes. In that study, CREM prevented production of IL-2, thereby contributing to T-cell exhaustion and therefore chronic viral infection. The abstract states 'These findings implicate the IFN β /CREM/IL-2 axis in regulating T-lymphocyte function during chronic viral infection'. More recently, CREM expression has been shown to be associated with CD8 exhaustion and poor prognosis in a human GI malignancy (<https://www.ncbi.nlm.nih.gov/pmc/articles/PMC8685542/>). However, there is recent evidence of PTPN22 expression in tissue-resident memory cells in lung cancer (<https://rupress.org/jem/article/216/9/2128/120725/Single-cell-transcriptomic-analysis-of-tissue>) described in a paper where the hypothesis that high expression of PD-1 and other related markers of apparent exhaustion by T memory cells from the lung reflects tissue-residency rather than exhaustion, was explored. Bosch et al do not describe how these earlier findings are challenged and developed here although the liver seems to provide an important novel context.*

Answer: We thank the reviewer for the constructive criticism of our manuscript. We have incorporated these aspects and publications relevant for chronic infection into the revised manuscript, but did not integrate a discussion on T cell dysfunction in cancer, which is beyond the scope of this manuscript.

We performed additional experiments to address the reviewer's comment on the impact of IFN-induced PTPN22 expression. First, we analysed our RNAseq data from liver CD8 T cells for the presence of a type I Interferon signature. GSEA demonstrated that virus-specific CD8 T cells from chronically infected livers did not show evidence for Interferon-induced signalling (**Fig. 2a**). Furthermore, to assess the role of PTPN22 in regulating function in CD8 T cells that are in close contact with LSECs, we used the specific pharmacological inhibitor of PTPN22-IN-1. Inhibition of the tyrosine phosphatase activity of PTPN22 did not rescue CD8 T cells from reduced expression of GzmB after coculture with LSECs (**Fig. 2b,c**). It is important to note that HBV is a stealth virus

and does not induce a type I Interferon response in the liver^{9,10}, in contrast to the prominent role of type I Interferon in LCMV infection¹¹.

Legend to Figure 2

a, Enrichment of type I interferon signalling by antigen-specific CXCR6⁺ T cells from persistent Ad-TTR-GOL infection compared to resolved Ad-CMV-GOL infection (n=3). **b,c**, Gzmb expression and quantification by CD44⁺CD8 T cells cocultured with activated LSECs for 24 h after 1 h pre-treatment of T cells with PTPN22-IN (1.4 μM, PTP22 inhibitor) or solvent control (n≥3, unpaired t-test).

Our data do not suggest that type I Interferon-induced regulatory pathways such as induction of PTPN22 play a critical role in downregulating CD8 T cell effector function during persistent hepatotropic infection.

Reviewer: Bosch et al use their previously described murine model of antigen-specific acute and chronic viral infection of the liver to compare how specific liver T cell populations function in both scenarios. This mouse model uses two adenoviruses with two different promoters to drive viral gene expression and the outcome of infection.

a. a cytomegalovirus promoter (Ad-CMV-GOL) that leads to acute resolved infection of the liver and b. a hepatocyte-specific transthyretin promoter (Ad-TTR-GOL) that drives persistent infection of the liver. Markers expressed by hepatic CXCR6⁺CD8 T cells after resolved infection included Gzmb, CXCR3, CCR2 and CCR5, and contrasted with enhanced expression of co-inhibitory receptors such as Pcd1, Lag3 and Tnfrsf9 by CXCR6⁺CD8 T cells during persistent hepatic infection. Increased CREM expression is seen, as described by other groups, in the exhausted viral-specific CD8 T cells but this time during chronic infection of the liver. The possibility that close interaction between CD8 T cells and LSECs might induce CREM expression by the T lymphocytes is novel and particularly intriguing. However, the authors show that CREM expression is not upstream of T cell exhaustion but is rather a consequence of PKA signalling. They demonstrate that treatment of memory CXCR6⁺CD8 T cells with the cAMP-inducing agent forskolin, increased PKA (the upstream activator of CREM) and caused loss of cytotoxic effector function (Fig. 4m), similar to the loss of effector function in liver CXCR6⁺CD8 T cells during persistent infection. But they provide little evidence of any mechanism that might explain activation of PKA in the LSEC-adherent hepatic liver CXCR6⁺CD8. They hypothesise that PKA activation might be induced by prostanoids produced by LSECs which is an intriguing, plausible and exciting concept but not explored in any depth. Interferon-secreting liver-resident DCs might also be important. It is disappointing that the authors do not pursue this line of investigation.

Answer: We thank the reviewer for these comments. We have performed numerous further experiments to address these points.

To characterise in detail the interaction between LSECs and CD8 T cells that we observed to be present *in situ* during persistent infection, we performed coculture experiments where CD8 T cells were in contact with primary mouse LSECs. In this *in vitro* model system, we detected increased phosphorylation of PKA and downregulation of GzmB expression in those effector CD8 T cells in close contact with LSECs (**Fig. 3a-d**). We used this *in vitro* system to address whether PKA activation downstream of cAMP was responsible for controlling effector CD8 T cell function. Using a specific inhibitor of PKA, we now provide evidence that PKA signalling downstream of cAMP controls CD8 T cell function (**Fig. 3e,f**). Furthermore, using a specific PKA agonist, we demonstrate that induction of PKA signalling is sufficient to diminish T cell effector function (**Fig. 3g**). In contrast, the alternative EPAC pathway downstream of cAMP did not affect T cell effector function, as we have seen using a specific agonist and antagonist of EPAC (**Fig. 3g-i**). Together, these data support the notion that PKA signalling is involved in controlling the effector function of T cells that interact closely with LSECs.

Legend Figure 3

a,b, Change in pPKA levels in CD8 T cells cocultured with liver cells (n=4, two-way ANOVA with Tukey's multiple comparison). **c,d**, GzmB expression by T cells in coculture with LSECs and quantification (n=3, unpaired t-test). **e,f** GzmB expression and quantification by CD8 T cells cocultured with activated LSECs for 24 h after 1 h pre-treatment with Rp-8-Br-cAMPS (1 mM, PKA antagonist) or solvent control (n=4, unpaired t-test). **g**, IFN γ +TNF $^+$ CD8 T cells (peptide-stimulated normalised to medium control) after 4 h pre-treatment of activated CD8 T cells with Sp-8-Br-cAMPS (250 μ M, PKA agonist) or solvent control followed by 15 h peptide restimulation (n=3, unpaired t-test). **h,i**, GzmB expression by CD8 T cells cocultured with LSECs for 24 h after 1 h pre-treatment with ESI-09 (10 μ M, EPAC antagonist), solvent or medium control (n=4, one-way ANOVA with Dunnett's multiple comparison). a-i: one out of \geq two independent experiments shown.

We addressed the question raised by this reviewer and also reviewer 1, what might induce increased cAMP signalling in T cells that are in close and extensive contact with LSECs. Purinergic signalling, particularly signalling through adenosine receptors, is a potent inducer of cAMP³. Although LSECs constitutively expressed the ectonucleotidase CD73 (**Fig. 4a,b**), which is relevant for the breakdown of AMP to adenosine and subsequent adenosine receptor signalling, they did

not express CD39 (**Fig. 4a,b**) an ectonucleotidase cleaving extracellular ATP to ADP and AMP⁴. Thus, LSECs cannot generate adenosine receptor signalling from extracellular ATP, suggesting that adenosine receptor signalling was not involved in LSEC-induced regulation of T cell function. Consequently, when we blocked adenosine receptor signalling with a specific pharmacological inhibitor, we did not detect an increase in GzmB expression by T cells in coculture with LSECs (**Fig. 4c,d**). Together, these results did not support a role of adenosine receptor signalling in LSEC-induced control of CD8 T cell effector function.

In contrast, we found that LSECs produced large amounts of PGE₂ (**Fig. 4e**), a known inducer of cAMP⁵. Inhibiting cyclooxygenase activity with the selective inhibitor Celecoxib increased GzmB expression in T cells cocultured with LSECs *in vitro* (**Fig. 4f,g**), indicating a contribution of PGE₂ to LSEC-mediated control of effector CD8 T cell function. Of note, the effect of celecoxib required high concentrations and was rapidly lost upon removal of the drug (**Fig. 4h**), indicating that an *in vivo* treatment would be challenging and unlikely to yield complete inhibition, especially in the liver where drugs are metabolised by hepatocytes.

Moreover, a recent publication from Pallett et al. in Nature demonstrated the transfer of molecules between CD8 T cells and macrophages in the liver⁶, which prompted us to look into transfer of molecules during cell-cell interaction between CD8 T cells and LSECs. When we labelled LSECs with the dye CFSE that binds to intracellular molecules, we detected LSEC-derived CFSE-labelled molecules in T cells (**Fig. 4i,j**), suggesting a transfer of molecules between LSECs and effector T cells. Since LSECs are defined by cAMP-agonism responsible for their differentiation and function⁷, our results may indicate that either adenylyl cyclase may be directly transferred from LSECs to T cells or that adenylyl cyclase-activating molecules from LSECs acted on T cells to enhance cAMP signalling.

This led us to explore the molecular determinants of LSECs in greater detail and we performed an untargeted lipidome analysis using LC-MS/MS for primary mouse LSECs, naïve CD8 T cells and activated CD8 T cells. We detected a total of 1200 lipids in LSECs and T cells. Principal component analysis clearly separated these cell populations based on their lipidome profiles (**Fig. 4k**), and direct comparison of LSECs with T cells revealed increased levels of phosphatidyl choline (PC) 19:0/18:1 and phosphatidic acid (PA) 18:1/18:2 in LSECs (**Fig. 4l-n**). Phosphatidyl choline and phosphatidic acid are potential precursors of 18:1 lysophosphatidic acid, a ligand of the G protein-coupled receptor LPA receptor 4 increasing adenylyl cyclase activity¹². Further experiments and a more detailed analysis of these lipidome profiles, however, will be needed to explore in detail the role of adenylyl cyclase-activating pathways. This will be a separate project that in our opinion is beyond the scope of this manuscript.

Since we cannot provide any formal evidence for the involvement of an intercellular molecule exchange and we did not include these data into the revised manuscript.

Legend to Figure 4

a,b, Expression of the ectonucleotidase CD39 and CD73 on LSECs and quantification ($n \geq 4$). **c,d**, GzmB expression and quantification by CD8 T cells cocultured with activated LSECs for 24 h after 1 h pre-treatment of T cells with SCH58261 (100 nM, A2AR antagonist), solvent control, or medium ($n=4$, one-way ANOVA with Dunnett's multiple comparison). **e** PGE₂ secretion of LSECs, hepatocytes and DCs over 24 h measured by ELISA ($n \geq 4$, one-way ANOVA with Tukey's multiple comparison). **f,g**, GzmB expression and quantification by CD44⁺CD8⁺ T cells cocultured with activated LSECs for 24 h with celecoxib (1 μ M), acetylsalicylic acid (100 μ M, ASS) or medium control ($n=5$, one-way ANOVA with Tukey's multiple comparison). **h**, GzmB expression by CD44⁺CD8⁺ T cells cocultured with activated LSECs pre-treated with celecoxib (1 μ M) or medium control for 1 h ($n=5$, unpaired t-test). **i,j**, CFSE incorporation and quantification by CD8 T cells in coculture with LSECs labelled with CFSE before coculture ($n=5$, one-way ANOVA with Tukey's multiple comparison). **k**, Principal component analysis of lipidome analysis by LC-MS/MS from naive T cells, activated T cells and LSECs ($n=3$). **l**, Comparison of LSEC and T cell lipidomes generated by LC-MS-MS from primary mouse cell populations ($n=3$). **m,n**, Phosphatidyl choline (C19:0/C18:1) and Phosphatidic acid (C18:1/18:2) detected in naive T cells, effector T cells and LSECs by LC-MS/MS ($n=3$). a-j: one out of ≥ 2 independent experiments shown.

We next followed up on the importance of adenylyl cyclase activity for inhibition of effector T cell function. Inhibition of adenylyl cyclase in CD8 T cells resulted in a rescue from downregulation of GzmB expression in CD8 T cells in coculture with LSECs (**Fig. 5a**). To demonstrate the importance

of adenylyl cyclase activity in T cells *in situ*, we treated antigen-specific CD8 T cells *in vitro* with an irreversible adenylyl cyclase inhibitor and then transferred these cells into mice with persistent or resolved liver infection. Adenylyl cyclase inhibitor-treated CD8 T cells had lower levels of pPKA and increased levels of GzmB and executed their effector functions *in vivo* as demonstrated by increased liver damage measured by augmented serum ALT levels (**Fig. 5b-e**). In conclusion, these new data now shed further light on the mechanism of the adenylyl cyclase/cAMP/PKA-induced regulation of T cell function by LSECs.

Legend to Figure 5

a, GzmB expression by CD8 T cells cocultured with activated LSECs for 24 h after 1 h pre-treatment of T cells with MDL-12330A (100 μ M, MDL), solvent control, or medium (n=4, one-way ANOVA with Dunnett's multiple comparison). **b**, Re-isolation of CD45.1⁺ CD8 T cell after transfer of MDL-12,330A (100 μ M, MDL)- or mock-treated CD45.1⁺ *in vitro*-activated CD8 T cells into mice with resolved Ad-CMV-GOL or persistent Ad-TTR-GOL infection. **c,d**, pPKA and GzmB levels and quantification by CD45.1^{+/+} CD8 T cells activated *in vitro* for 3 d followed by 1 h pre-treatment with MDL or mock before transfer into Ad-TTR-GOL or Ad-CMV-GOL infected mice for 3 d (n=4, one-way ANOVA with Tukey's multiple comparison). **e**, Increase of sALT per CD45.1^{+/+} CD8 T cell after transfer of MDL-treated or mock-treated CD45.1^{+/+} *in vitro* activated CD8 T cells into mice with resolved Ad-CMV-GOL or persistent Ad-TTR-GOL infection (n \geq 4, one-way ANOVA with Tukey's multiple comparison). a-e: one out of \geq two independent experiments shown.

We next addressed the reviewer's question on the role of interferon-secreting DCs. As already mentioned in answer 1, we did not detect an interferon signature in hepatic antigen-specific CD8 T cells from persistently infected mice (*see Fig. 2a*), which does not support the notion that these T cells had contact with interferon-secreting DCs. To further investigate the interaction between liver dendritic cells and CD8 T cells in the liver *in situ* in the context of persistent viral infection, we performed additional confocal microscopy experiments (**Fig. 6a**). Here, antigen-specific CD8 T cells had a mean distance of 173,1 μ m to CD11c⁺CD103⁺MHC-II⁺ liver dendritic cells (**Fig. 6b**). In contrast, the mean distance of LSECs to CD8 T cells was 1.15 μ m (**Fig. 6b**). Furthermore, liver-resident dendritic cells, identified by being CD11c⁺MHC-II⁺CD103⁺XCR1⁺CD24⁺Ly6C^{neg}, were not increased in frequency and did not show any sign of increased activation as determined by higher

expression levels of CD86 or MHC-II (**Fig. 6c-e**), which did not provide any hint for direct involvement of liver-resident DCs in the silencing of effector T cell function. Together, these results did not support - but did also not formally exclude - a contribution of interferon-secreting liver DCs to local regulation of effector CD8 T cell function.

Legend to Figure 6

a, 3D rendered surfaces of confocal microscopy imaging of CD103⁺CD11c⁺MHCII⁺ dendritic cells and CD45.1⁺ antigen-specific T cells in livers of mice with resolved 10⁷ IU Ad-HBV or persistent 10⁸ IU Ad-HBV infection (n=5). **b**, Shortest distance of CD45.1⁺ CD3⁺ T cells to CD103⁺CD11c⁺MHCII⁺ dendritic cells or CD146⁺ LSECs in livers of mice with persistent infection (n=5, unpaired t-test). **c**, Frequencies of liver non-parenchymal cell populations from mice with resolved 10⁷ IU Ad-HBV or persistent 10⁸ IU Ad-HBV infection (n=5, two-way ANOVA with Tukey's multiple comparison). **d,e**, CD86 expression and quantification by CD11c⁺MHC-II⁺CD103⁺XCR1⁺CD24⁺Ly6C^{neg} dendritic cells from mice with resolved 10⁷ IU Ad-HBV or persistent 10⁸ IU Ad-HBV infection (n=5, unpaired t-test).

Reviewer: CXCR6 is a marker of liver residency. Is there something about CXCR6 expression (and the expression of other markers that are indicative of tissue residency like DR) that alters liver lymphocyte activation and makes liver lymphocytes more susceptible to cAMP/PKA signalling, CREM activation and ultimate exhaustion or apparent exhaustion? Do tissue-resident populations of lymphocytes have different mechanisms of exhaustion to circulating populations? Are CXCR6 liver NK cells equivalently susceptible?

Answer: We thank the reviewer for these comments, which we addressed by performing a series of experiments.

First, we used homozygous CXCR6^{GFP/GFP} knock-out reporter mice that lack functional CXCR6 on their cell surface, as the CXCR6 gene is replaced by GFP¹³. In these mice, we did not detect any changes in the frequencies or absolute cell counts of antigen-specific CXCR6^{GFP/GFP} CD8 T cells in mice with persistent hepatic infection or resolved infection compared to wildtype littermates

(Fig. 7a,b). Furthermore, during persistent infection, we did not detect any changes in the expression of relevant markers, i.e., GzmB, and pPKA, in liver CXCR6^{GFP/GFP} CD8 T cells in homozygous CXCR6^{GFP/GFP} knock-out mice compared to CXCR6^{WT/WT} liver CD8 T cells in control mice (Fig. 7c). Moreover, the overall immune response to infection was not affected in homozygous CXCR6^{GFP/GFP} knock-out mice, i.e., neither clearance of viral infection nor persistence of infection different in CXCR6^{GFP/GFP} knock-out mice (Fig. 7d). Together, these results suggest no role of CXCR6 in the interaction of CD8 T cells with the liver tissue-rheostat.

We next addressed whether circulating antigen-specific CD8 T cells were regulated differently than those T cells found in the liver during persistent viral infection. Circulating antigen-specific CD8 T cells from persistently infected mice were also characterised by increased PKA phosphorylation compared to CD8 T cells from mice that had resolved liver infection (Fig. 7e,f), which suggested that antigen-specific CD8 T cells in the liver and the circulation showed similar regulation during persistent liver infection.

We thank the reviewer for suggesting to expand our investigation's scope to CXCR6⁺ NK cells in the liver. We did not find any difference in the production of cytokines *in situ* between CXCR6⁺ NK cells in livers after resolved infection or in livers during persistent infection (Fig. 7g). Furthermore, there was no difference in PKA phosphorylation between CXCR6-competent vs CXCR6^{GFP/GFP} NK or NKT cells in the liver during persistent viral infection (Fig. 7h). Together, these results demonstrate that the liver tissue-rheostat did not affect CXCR6⁺ NK and NKT cells. These results further support the notion that the tissue-rheostat selectively affects antigen-specific CD8 T cells recognising their antigen in the liver. Again, we thank the reviewer for bringing up these important points, which helped us further improve the manuscript.

Legend to Figure 7

a,b, Frequencies and absolute counts of streptamer⁺ antigen-specific CD8 T cells in CXCR6-KO and wildtype mice on d45 after resolved or during persistent infection (n=5, two-way ANOVA with Sidak's multiple comparison). **c**, Quantification of GzmB and pPKA by antigen-specific CD8 T cells from CXCR6-KO or wildtype littermates on d45 during persistent infection (n=5, unpaired t-test). **d**, Bioluminescence imaging

of CXCR6-KO or wildtype littermate mice infected with acute-resolving Ad-CMV-GOL or persistent Ad-TTR-GOL infection (n=5, two-way ANOVA). **e,f**, pPKA levels and quantification by CXCR6⁺CD45.1⁺CD8 T cells in blood on d45 after infection with resolved or persistent infection (n=3, unpaired t-test). **g**, *In situ* cytokine expression (IFN γ and/or TNF) by NK1.1⁺CXCR6⁺ cells in livers of mice on d45 after infection with 10⁷ or 10⁸ IU Ad-HBV that received Brefeldin A (2.5 mg) 6h before analysis (n=5, unpaired t-test). **h**, pPKA expression by liver NK and NKT cells of CXCR6-KO or wildtype littermate mice on d45 after infection with 10⁷ or 10⁸ IU Ad-HBV (n=5, two-way ANOVA with Sidak's multiple comparison). a-d: one out of \geq two independent experiments shown.

Reviewer: More clearly presented data defining exhaustion of hepatic T cells and the inhibition of their function in the two models are required. 'and hepatic accumulation of activated CD8 T cells that...'. Are CD8+ hepatic lymphocytes preferentially effected? What about liver-resident CD4 T cells? Liver-resident NK cells? It is not clear how the human data contribute to this core concept of the study 'circulating HBcore-specific CD8 T cells did not express CXCR6...'. Metabolic activity, inevitably driven by the microenvironment, is a key determinant of lymphocyte activation status, function, and phenotypic marker expression.

Answer: We apologise for not having clearly presented the data in Figures 1 to 3 in the previous manuscript. Following this reviewer's comment on the manuscript's overall structure, we have thoroughly revised the previous Figures 1 to 3. In the revised manuscript, we now present the data on the characteristics of CD8 T cell dysfunction during persistent hepatic infection in a much more focused manner: *Figure 1* displays functionality and cAMP/CREM-dependent regulation of antigen-specific CD8 T cells in the liver during persistent infection in the adenoviral model. *Figure 2* illustrates the effector function and cAMP/CREM-dependent regulation of HBV-specific liver CD8 T cells in persistent HBV infection. *Figure 3* identifies cAMP/CREM-dependent regulation in HBV-specific CD8 T cells from patients with chronic hepatitis B, and *Figure 4* explains the mechanistic determinants of the liver tissue-rheostat influencing CD8 T cell effector function via the adenylyl cyclase/cAMP/PKA axis.

Again, we thank the reviewer for bringing up the important question on the selective regulation of CD8 T cell function by the liver tissue-rheostat. We have performed further experiments to investigate the change in phosphorylation of PKA in polyclonal liver CD4 T cells after resolved or during chronic viral infection. There was no substantial difference in PKA phosphorylation in polyclonal hepatic CD69⁺CXCR6⁺CD44⁺CD62L^{neg} CD4 T cells from livers after resolved or during chronic infection (**Fig. 8a**). Furthermore, there was no change in frequencies of polyclonal liver CD69⁺CXCR6⁺CD44⁺CD62L^{neg} CD4 T cells (**Fig. 8b**). We further extended our analysis to liver NK cells, in which we neither detected significant changes in cytokine expression *in situ* nor PKA phosphorylation after resolved or during persistent infection (**Fig. 8c,d**). Together, these results confirmed that the liver tissue-rheostat selectively acted on virus-specific CD8 T cells recognizing their antigen during persistent infection of the liver. All these data are now shown in the revised manuscript.

The reviewer further refers to the relevance of our findings in circulating human HBV-specific CD8 T cells for the core concept of our study. We apologise for not having been sufficiently clear in the initial manuscript, where we used the term “tissue residency signature” to describe virus-specific CXCR6⁺ CD8 T cells during persistent infection. In the revised manuscript, we now clearly emphasise that these dysfunctional virus-specific CXCR6⁺ CD8 T cells during persistent viral infection are not tissue-resident and do not have a T cell-intrinsic tissue-residency program defined by transcriptional activity of Hobit/Blimp1. Thus, although dysfunctional CXCR6⁺ CD8 T cells during persistent infection share expression of markers like CXCR6, CD69, and PD1 with tissue-resident memory T cells, they are not truly tissue-resident T cells. In contrast, the CXCR6⁺ CD8 T cells after resolved infection showed gene regulation by Hobit/Blimp1, but did not show evidence for cAMP/CREM signalling, indicating that distinct mechanisms governed a tissue residency state after resolved infection compared to impact of the liver tissue-rheostat on T cells in the liver during persistent infection. Thus, the lack of CXCR6 expression and absence of a tissue-residency signature in dysfunctional HBV-specific CD8 T cells circulating in humans is not a contradiction to the core concept of our study. Instead, detecting the distinctive cAMP/CREM signature in circulating HBV-specific CD8 T cells points to their previous encounter with the liver tissue-rheostat. We have detected the cAMP/CREM signature in circulating HBV-specific CD8 T cells from five further patients with chronic hepatitis B compared to HBV-specific CD8 T cells from two patients who resolved hepatitis B (**Fig. 8e**). Notably, this cAMP/CREM signature was not detected in non-HBV-specific CD8 T cells from the same patients (**Fig. 8f**). These data are now shown in the revised manuscript.

Legend to Figure 8

a, Quantification of PKA phosphorylation by liver CXCR6⁺CD44⁺CD4 T cells at d45 after infection with resolved or persistent infection (n≥3, unpaired t-test). **b**, Frequencies of CXCR6⁺CD69⁺CD4 T cells in livers of mice on d45 after infection with 10⁷ or 10⁸ IU Ad-HBV (n= 5, unpaired t-test). **c**, *In situ* cytokine (IFN γ and/or TNF) expression by CXCR6⁺ NK cells in livers of mice on d45 after infection with 10⁷ or 10⁸ IU Ad-HBV that received 2.5 mg Brefeldin A 6h before analysis (n=5, unpaired t-test). **d**, pPKA expression in CXCR6⁺ NKT cells or NK cells in livers of mice on d45 after infection with 10⁷ or 10⁸ IU Ad-HBV (n=5, two-way ANOVA with Tukey’s multiple comparison). **e,f** GSEA for a cAMP/CREM signalling signature in HBcore-

specific CD8 T cells in patients with chronic hepatitis B compared to individuals with resolved hepatitis B, or non-HBV-specific CD8 T cells in patients with chronic hepatitis B patients (f) (n=5).

Reviewer: *The striking differences in T cell numbers and LSEC density in the persistent versus resolved liver infections (Fig 4b) – presumably there’s also associated stromal cell activation and fibrosis – are indicative of major differences in microenvironment likely to impact T cell metabolism. While metabolomics may be beyond the scope of this research programme, additional probing of the NGS data may well be informative. The recent paper by Fisher et al on ‘Modelling intercellular communication in tissues using spatial graphs of cells’ (<https://www.nature.com/articles/s41587-022-01467-z>) may be informative’.*

Answer: We thank the reviewer for this comment raised similarly by reviewer 1. We have now included a detailed description of the clinical parameters of the patients with chronic hepatitis B who donated blood for the analyses shown in this manuscript. All patients with chronic hepatitis B had a low fibroscan score below 10, had no elevation of bilirubin and low levels of liver enzymes consistent with a Child A low liver damage score (**Fig. 9a,b**). In the preclinical models of persistent viral infection, we also did not detect liver fibrosis or substantial liver damage (**Fig. 9c-e**). Although this does not exclude an effect of different hepatic microenvironments in chronic hepatitis B patients and preclinical models of persistent viral infection on virus-specific CD8 T cells, the similarities concerning low liver damage, low level of inflammation, and a normal hepatic microarchitecture rather suggest rather similar conditions in preclinical models of hepatotropic infection and in patients with chronic hepatitis B.

Legend to Figure 9

a,b, Viral and host parameters of chronic hepatitis B patients (cHBV) or resolved hepatitis B (rHBV), whose HBV-specific CD8 T cells were subjected to single-cell RNAseq analysis (a) or for of SMART-Seq2 analysis

(b). **c,d**, serum ALT levels in mice during acute-resolving or persistent hepatotropic infection compared to uninfected mice ($n \geq 4$, two-way ANOVA with Tukey's multiple comparison). **e**, H/E staining of liver sections from mice with resolved or persistent infection on d45 p.i. ($n \geq 4$). c-e: one out of two independent experiments is shown.

We followed the reviewer's suggestion to look at metabolic features of virus-specific T cells isolated from liver during persistent infection, which is extremely difficult since these cells are so scarce. Unfortunately, we do not have a full representation of all liver cells by RNAseq, because the scarcity of dysfunctional virus-specific CD8 T cells in liver tissue would require sequencing of millions of cells to find a sufficient number of dysfunctional T cells for analysis. We, therefore, did not use a node-centric expression model to visualise cell-cell interactions as described by Fischer et al.¹⁴, but probed our RNAseq data for metabolic changes, as the reviewer suggested. Gene set enrichment analysis for RNAseq data from these dysfunctional CD8 T cells compared to fully functional CD8 T cells revealed increased expression of fatty acid beta oxidation-associated genes (**Fig. 10**). Of note, we did not find increased expression of genes associated with glycolysis (**Fig. 10**), which is a feature of full activation of effector CD8 T cells. These results suggest that energy metabolism in dysfunctional CXCR6⁺ CD8 T cells was not fundamentally different from resting CD8 T cells after resolved infection.

Legend to Figure 10

GSEA for defined metabolic signatures between CXCR6⁺CD45.1⁺ CD8 T cells from livers of mice with resolved vs persistent infection ($n=3$).

Reviewer: *What is it about the model that the effects are only seen with adenovirus and not with other viruses?*

Answer: In this manuscript, we characterise the immune response to viruses that target hepatocytes, such as Hepatitis B Virus (HBV). We, therefore, use recombinant adenoviruses that are hepatotropic and thereby mirror the hepatocyte targeting feature of HBV as viral gene shuttle for delivery into hepatocytes. HBV does not infect mouse hepatocytes, but adenoviral gene transfer can achieve hepatocyte transduction and HBV replication in hepatocytes *in vivo*. Our data demonstrate that a cAMP/CREM signature is also detected in HBV-specific CD8 T cells in mice that express HBV genes as a transgene in the liver², demonstrating that adenoviral gene transfer is not a prerequisite for induction of a cAMP/CREM signature in virus-specific CD8 T cells (**Fig. 11a**).

Moreover, we detect the cAMP/CREM signature also in HBV-specific CD8 T cells in patients with chronic hepatitis B, which together suggest that hepatocyte-specific infection is associated with the development of this gene signature.

Of note, we did not detect a cAMP/CREM signature in LCMV-specific CXCR6+ CD8 T cells from the liver in systemic LCMV infection (**Fig. 11b**). During persistent LCMV infection, T cells recognise their antigen on many different cell populations and not selectively on hepatocytes. To provide further evidence that the cAMP/CREM signature is only found in chronic viral infection where CD8 T cells recognise their antigen on hepatocytes, we analysed a publicly available dataset of single cell RNAseq of virus-specific CD8 T cells from patients with persistent HIV infection, comparing elite controller to patients with persistent progressive infection¹⁵. Consistent with our concept, these HIV-specific CD8 T cells did not have a cAMP/CREM signature (**Fig. 11b**). These results are now shown in the revised manuscript.

Legend to Figure 11

a, GSEA for a cAMP/CREM signature in dysfunctional HB_{core} CD8 T cells recognizing their antigen on hepatocytes in transgenic MUP-core mice, in which the HB_{core} protein is expressed in hepatocytes controlled by the major urinary protein (MUP) promoter compared to rLCMV-core infected mice with ubiquitous core expression. **b**, GSEA for a cAMP/CREM signature in CD8 T cells from patients with persistent HIV infection compared to elite controller patients (n=4).

Other comments

Reviewer: *This paper seems to have been assembled in haste, without adequate curation and editing. Nine figures of extended experiments and data (in addition to the figures provided for the paper) are indeed important for conveying the amount of work that has gone into this study but still seem excessive for a Nature paper which presumably is aiming to communicate a single 'paradigm shifting' concept. Should the reader have to work so hard to extract the key points from the terabytes of data provided to identify the paradigm shift?*

Answer: We have followed the reviewer's comments and have restructured the manuscript and removed all non-essential information to make it easier for the reader to understand the new concept of a liver-tissue rheostat. While incorporating all the data from the revision, the manuscript now has 4 Main Figures and 7 Extended Data Figures. The comments from all three reviewers helped us to focus on the most relevant experiments that are incorporated into the revised manuscript.

Reviewer: *Typographical errors are additional indication of the haste*

Line 60 'explaining their loss of effector.'[sic]

Answer: We have corrected this error.

Reviewer: *Line 99 ...'only emerged only' [sic]*

Answer: We have corrected this error.

Reviewer: *Line 100: 'Thus, this model system reflects the scarcity of virus-specific CD8 T cells during chronic HBV infection and hepatic accumulation of activated CD8 T cells that fail to clear virus-infected hepatocytes.' This is a key observation but where is the figure? If previously published, it needs to be referenced.*

Answer: We have corrected this error and now provide references for this statement (page 3, line 77).

Reviewer: *Line 138: 'Few genes were mutually exclusively expressed and characterized liver CXCR6+CD8 T cells after resolved infection, such as GzmB, Itga1 and Gadd45b, or liver CXCR6+CD8 T cells during persistent infection, such as Crem and the TGF β -dependent transcription factors Tgif1 and Ski' Awkward sentence.*

Answer: We have taken out this sentence that was difficult to understand.

Reviewer: *Line 172: 'in killing of hepatocytes'.*

Answer: We thank the reviewer for this correction. We have carefully revised the entire revised manuscript.

Reviewer: *The discussion is merely a brief summary of the main findings of the paper; no attempt is made to place the data into the context of the title or previous discoveries.*

Answer: We have now included a discussion taking into consideration the relevant aspects mentioned by the three reviewers.

Reviewer: *Some key references seem to be missing [see above].*

Answer: We have added the references mentioned by the reviewer.

Referee #3 (Remarks to the Author):

Title: “A PKA-associated liver-tissue rheostat curbs T cell receptor signalling and effector function of virus-specific CD8 T cells in chronic viral hepatitis” by Percy Knolle and colleagues. In the present manuscript by Percy Knolle and colleagues, the authors intended to understand the molecular mechanisms underlying exhaustion of virus-specific CD8 T cells during persistent hepatotropic hepatitis B virus (HBV) infection.

Validity

Data presented in this paper are thoroughly analyzed and statistical testing is performed where indicated. Data are not over-interpreted, although additional experiments should be considered to further understand the cAMP-steered pathways involved and to substantiate the HBV- and liver specificity of their findings. In general, there is no concern about the validity of the findings presented in this manuscript.

Answer: We thank the reviewer for this comment. We have performed a series of further experiments (outlined in detail below) to further understand the molecular mechanisms of cAMP-steered pathways in regulating CD8 T cell function during persistent hepatotropic infection.

Significance

The presented study is highly significant for the field of viral immune evasion mechanisms and especially HBV and chronic viral hepatitis. Supportive for this finding are also older publications, in which the cAMP and PKA pathways have already been shown to be implicated in exhausted T cells of human immunodeficiency virus (HIV)-infected patients. Also, T cells from HIV-infected patients show elevated levels of cAMP and hyperactivation of PKA. A comparable mechanism also contributes to T cell dysfunction in a subset of patients with common variable immunodeficiency, and to an anergy-like phenotype of T cells observed in a murine model termed MAIDS.

Answer: We thank the reviewer for these comments. We have followed the advice to look more closely at virus-specific CD8 T cells during chronic HIV infection. We used a publicly available dataset that contained single cell RNAseq data for HIV-specific CD8 T cells from patients with chronic infection and from elite controllers of HIV infection¹⁵. Using these data, we did not find a cAMP/CREM signature in HIV-specific CD8 T cells during chronic HIV infection (**Fig. 1**). We included this figure in the revised manuscript.

Legend to Figure 1

GSEA for a cAMP/CREM signature in CD8 T cells from patients with persistent HIV infection compared to elite controller patients (n=4).

We did not find any publicly available datasets for the MAIDS model and, therefore, could not perform similar experiments for virus-specific CD8 T cells in this retroviral infection model. We included these publications into the discussion.

Data and methodology

Data presented in this study are of high quality, experimental repeats have been performed as required.

Answer: We thank the reviewer for acknowledging the high quality of our data.

Analytical approach

More emphasis should be put on understanding the exact molecular mechanism of how cAMP is mediating these effects.

Answer: We thank this reviewer for his positive evaluation of our findings. As suggested by the reviewer, we have performed a series of experiments to provide more mechanistic and molecular understanding how cAMP is mediating inhibition of effector T cell function.

Suggested improvements

Major criticism

1. In Figure 1 and the corresponding Extended Data Figure 1 the authors provide a canonical tissue-residency gene signature of liver CXCR6+ CD8 T cells, indicative of the involvement of cAMP-mediated signal transduction. Can this gene signature also be found in CD8+ T cells of HBV-infected patients? Is this canonical tissue-residency gene signature differentially expressed in HBcoreCD8 T cells, that completely lost cytokine expression when compared to liver CD8 T cells in general?

Answer: We thank the reviewer for bringing up this important issue. We apologise for not having been sufficiently clear on the issue of tissue residency. We detect the cAMP/CREM signature as a clear difference between dysfunctional rheostat-affected CXCR6+ T cells in the liver during persistent infection that are not tissue-resident memory T cells, and *bona fide* liver-resident memory CXCR6+ CD8 T cells, which evolve after resolved infection and are not affected by the liver tissue-rheostat. This is supported by our data showing that only functional CXCR6+ CD8 T cells after resolved infection have a Hobit/Blimp1 signature involved in the cell-intrinsic regulation of tissue residency (**Fig. 2a**). In contrast, only dysfunctional antigen-specific CXCR6+ CD8 T cells from livers with persistent infection had an enrichment for the cAMP/CREM signature (**Fig. 2b**). This all points to an exclusive presence of a Hobit/Blimp1 signature in liver-resident memory CD8 T cells after resolved infection compared to the selective presence of the cAMP/CREM signature in dysfunctional CXCR6+CD8 T cells.

Thus, there is no evidence for tissue-resident memory CXCR6+ CD8 T cells having a cAMP/CREM signature. Conversely, there is no evidence for dysfunctional CXCR6+ CD8 T cells having a tissue

residency signature. However, dysfunctional T cells share many similarities in the expression of adhesion molecules involved in tissue residency.

Legend to Figure 2

a,b, Gene set enrichment analysis (GSEA) in liver CD45.1⁺CXCR6⁺CD8 T cells from resolved (left) and persistent (right) infection for Hobit/Blimp1-dependent genes and a cAMP/Crem signature (n=3).

To provide more evidence for the presence of the cAMP/CREM signature in human circulating HBcore-specific CD8 T cells, we included five more patients with chronic hepatitis B and included further controls, i.e. non-HBV-specific (bulk) CD8 T cells from the same chronic hepatitis B patients as well as HBcore-specific CD8 T cells from patients who had cleared the infection. In patients with chronic hepatitis B, we detected circulating HBcore-specific CD8 T cells that showed this cAMP/CREM signature but not in patients who had cleared HBV infection (**Fig. 3a**). Moreover, we did not detect a cAMP/CREM signature in non-HBV-specific CD8 T cells from the same patients (**Fig. 3b**). These data are now included in the revised manuscript.

To address the question of whether circulating HBcore-specific CD8 T cells in patients had a tissue residency signature, we first created a human tissue-residency signature using a scRNAseq analysis of T cells from human liver explants. We identified a T cell cluster (cluster L7) defined by Hobit-dependent genes (**Fig. 3c,d**). When comparing circulating HBcore-specific CD8 T cells from patients with chronic hepatitis B to these results, we did not identify similarities with the Hobit-defined cluster of liver T cells (**Fig. 3e**). Likewise, HBcore-specific CD8 T cells in patients who had cleared HBV infection did not show a Hobit-defined tissue-resident memory T cell signature (**Fig. 3f**). Thus, our results suggest that circulating HBcore-specific CD8 T cells from patients with chronic hepatitis B have a distinctive cAMP/CREM signature but not a tissue-resident memory T cell signature. Since we do not want to lose focus and risk a confusion of tissue-resident T cells and tissue-resident memory T cells, we have not included the data from Fig. 3c-f in the revised manuscript.

As for liver lymphocytes in general, we isolated liver CD4 T cells, liver NKT cells and liver NK cells from mice with persistent hepatotropic infection. In none of these liver lymphocyte populations we detected increased phosphorylated PKA levels by flow cytometry (**Fig. 3g**). Finally, we compared antigen-specific CD8 T cells during persistent hepatotropic infection with polyclonal CD8 T cells from the same liver. These polyclonal liver CD8 T cells had lower pPKA levels compared to the antigen-specific CD8 T cells (**Fig. 3h,i**). When comparing polyclonal liver CD8 T

cells from persistent infection to acute resolved infection, we also did not detect differences in pPKA levels (**Fig. 3h,i**), further demonstrating that changes in pPKA levels were specific for antigen-specific CD8 T cells recognizing their antigen on infected hepatocytes during persistent infection.

Together, these results indicate that only virus-specific CD8 T cells during persistent hepatotropic infection acquired the distinctive cAMP/CREM signature. All these data have been integrated into the revised manuscript.

Legend to Figure 3

a,b, GSEA for a cAMP/CREM signature in HLA-A2 restricted HBcore-specific from HLA-A2+ patients with chronic hepatitis B (cHBV, n=5) compared to HLA-A2+ patients with resolved hepatitis B (rHBV, n=2) (a),

or in non-HBV-specific CD8 T cells (b). **c**, UMAP clusters from single cell RNAseq of CD8 T cells isolated from human liver explants (n=2). **d**, Feature plot showing selected genes characterising liver CD8 T cell UMAP clusters from scRNAseq of human CD8 T cells isolated from liver explants (from c). **e**, Similarity analysis of peripheral blood HBV-specific CD8 T cells from patients with chronic HBV infection with human liver CD8 T cell clusters (n=3). **f**, GSEA of DEGs between HBcore-specific CD8 T cells in chronic HBV (cHBV, n=5) vs resolved HBV (rHBV, n=2) patients with a tissue-resident memory T cell signature¹⁶. **g**, pS114 PKA (pPKA) levels by liver CXCR6⁺ NK, NKT or CD4 T cells at d45 after infection with 10⁷ or 10⁸ IU Ad-HBV (two-way ANOVA with Tukey's multiple comparison, n=5). **h,i**, pS114 PKA (pPKA) levels in liver CD45.1⁺CD8 T cells compared to CD45.1^{neg}CD8 T cells on d45 p.i. and quantification (two-way ANOVA with Tukey's multiple comparison, n=5).

Reviewer: 2. Cyclic AMP-mediated signal transduction was also shown to be involved in dysfunction of T cells in other viral infections. For example, T cells from human immunodeficiency virus (HIV)-infected patients show elevated levels of cAMP and hyperactivation of PKA. Targeting of the cAMP-PKA type I pathway by selective antagonists reverses T cell dysfunction in HIV T cells ex vivo^{1,2}. A similar mechanism contributes to T cell dysfunction in a subset of patients with common variable immunodeficiency³, and to promote an anergy-like T cell phenotype in a murine immunodeficiency model termed MAIDS⁴. Hence, the authors should be encouraged to analyze to what extent enrichment of these genes can also be found in already published data sets of dysfunctional T cells from other viral infections like HIV in order to substantiate HBV infection- and tissue-specificity of this immune evasion program.

Answer: We thank the reviewer for giving us the opportunity to address the difference between persistent HBV infection to other chronic viral infections in humans such as HIV. As we have pointed out in Figure 1 above, we did not find a cAMP/CREM signature in HIV-specific CD8 T cells in patients with persistent HIV infection. Since HIV does not infect hepatocytes and HIV-specific CD8 T cells do not engage with hepatocytes in a cognate fashion, these data support our notion that the cAMP/CREM signature is a feature of the liver tissue-rheostat that only engages with virus-specific CD8 T cells recognizing their antigen on infected hepatocytes. However, it remains unresolved in our analysis of how different microenvironments during chronic HIV infection may influence cAMP levels in T cells.

Reviewer: 3. In Fig. 4e,f and Extended Data Fig. 9f the authors provide evidence of enhanced PKA activity in CXCR6⁺ CD8 T cells during persistent infection. To substantiate the involvement of PKA the authors should be encouraged to design experiments e.g. comparable to the adoptive transfer experiment described and pretreat the adoptively transferred cells with a PKA inhibitor like RP-cAMPS5 or any other PKA inhibitor like H89 and compare the phenotype of those cells to endogenous non-treated CD8⁺ T cells.

Answer: We followed the reviewer's suggestion and used the PKA inhibitor Rp-8-Br-cAMPS to block PKA activity in virus-specific CD8 T cells. Since we found it impossible to use this competitive and reversible inhibitor for adoptive transfer experiments that last for 72h, we used cocultures of T cells with LSECs as a surrogate for the liver tissue-rheostat – since we found that during persistent infection, the contact surface between LSECs and antigen-specific T cells was extensive and their distance very short compared to other cells in the liver like dendritic cells (**Fig.**

4a,b). This suggested that close contact with LSECs caused enhanced PKA activation. Indeed, in cocultures with LSECs, T cells showed increased PKA phosphorylation and within 24h downregulation of GzmB expression. Since this showed the downregulation of T cell effector functions in T cell – LSEC cocultures, we used this model to study the involvement of PKA in more detail. The PKA inhibitor Rp-8-Br-cAMPS prevented the loss of GzmB expression in CD8 T cells in coculture with LSECs (**Fig. 4c**). Conversely, when we activated PKA with an agonist (Sp-8-Br-cAMPS), such treated T cells lost their ability to respond to cognate stimulation with cytokine production (**Fig. 4d**). These results provide evidence for the contribution of PKA activation in T cells through interaction with LSECs and demonstrate that selective PKA activation is sufficient to impair CD8 T cell effector function. These data are included in the new Figure 4 of the revised manuscript.

Legend to Figure 4

a, 3D-rendered surfaces of confocal microscopy imaging of CD103⁺CD11c⁺MHCII⁺ dendritic cells and CD45.1⁺ antigen-specific T cells in livers of mice with resolved 10⁷ IU Ad-HBV or persistent 10⁸ IU Ad-HBV infection, bar = 50μm (n= 5). **b**, Shortest distance of CD45.1⁺ CD3⁺ T cells to CD103⁺CD11c⁺MHCII⁺ dendritic cells or CD146⁺ LSECs in livers of mice with persistent infection (n= 5, unpaired t-test). **c**, GzmB expression by CD8 T cells cocultured with activated LSECs for 24 h after 1 h pre-treatment with Rp-8-Br-cAMPS (1 mM, PKA antagonist) or solvent control (n=4, unpaired t-test). **d**, IFN_γ⁺TNF⁺ CD8 T cells (peptide-stimulated normalised to medium control) after 4 h pre-treatment of activated CD8 T cells with Sp-8-Br-cAMPS (250 μM, PKA agonist) or solvent control followed by 15 h peptide restimulation (n=3, unpaired t-test). a-i: one out of ≥ two independent experiments shown.

Reviewer: 4. In the same vein, have the authors analyzed involvement of other cAMP-dependent but PKA-independent mechanisms like EPACs6,7, which have been shown to be involved in the TCR-dependent activation of T cells.

Answer: We followed the reviewer's suggestion and used the EPAC-inhibitor ESI-09. We did not detect any effect of the EPAC-inhibitor ESI-09 on the expression level of GzmB in CD8 T cells in coculture with LSECs *in vitro* (**Fig. 5a,b**). More importantly, agonistic stimulation of EPAC signalling using 8-CPT-2-Me-cAMP in T cells did not regulate their Interferon production in response to cognate stimulation (**Fig. 5c**). Thus, we did not detect evidence for the involvement of EPAC in regulating the effector function of T cells through LSECs. These data are included in Figure 4 of the revised manuscript.

Legend to Figure 5

a,b, GzmB expression by CD8 T cells cocultured with LSECs for 24 h after 1 h pre-treatment with ESI-09 (10 μ M, EPAC antagonist), solvent, or medium control and quantification (n=4, one-way ANOVA with Dunnett's multiple comparison). **c**, IFN γ ⁺TNF⁺ CD8 T cells (peptide-stimulated normalised to medium control) after 4 h pre-treatment of *in vitro* activated CD8 T cells with 8-CPT-2Me-cAMP (30 μ M, EPAC agonist), or solvent control followed by 15 h peptide restimulation (n=3, unpaired t-test). a-c: one out of \geq two independent experiments shown.

Reviewer 5. In order to underline the importance of cAMP in the observed T cell phenotype the authors conducted experiments involving Forskolin treatment to increase cAMP. However, the authors should also be encouraged to use adenylyl cyclase inhibitors like MDL-128,9 to demonstrate the importance of de novo cAMP production *in vitro* as well as *in vivo*.

Answer: We thank the reviewer for this suggestion! We addressed the relevance of adenylyl cyclase in T cells *in vitro* as well as *in vivo*. To this end, we used a non-reversible inhibitor of adenylyl cyclase activity (MDL-12330A) that allowed us to perform *in vivo* experiments using adoptively transferred T cells, which had been pre-treated with MDL-12330A. First, pre-treatment of T cells with MDL-12330A prevented their reduction in GzmB expression following contact with LSECs *in vitro* (**Fig. 6a**), which suggested a role for adenylyl cyclase in controlling effector CD8 T cell function by LSECs. To provide evidence for a role of adenylyl cyclase in CD8 T cells *in vivo* in the liver, we adoptively transferred virus-specific CD45.1⁺CD8 T cells pre-treated with MDL-12330A or receiving a mock treatment into mice with persistent liver infection. Clearly, virus-specific CD8 T cells adoptively transferred into mice with persistent infection showed increased pPKA and decreased GzmB levels compared to virus-specific CD8 T cells transferred into mice with resolved infection (**Fig. 6b,c**). Strikingly, after treatment with the adenylyl cyclase inhibitor, MDL-12330A, virus-specific CD8 T cells transferred into mice with persistent hepatotropic infection showed lower levels of pPKA and higher levels of GzmB (**Fig. 6b,c**). Moreover, we detected increased serum ALT levels as an indicator of liver damage and T cell effector function only when MDL-12330A-treated virus-specific CD8 T cells were adoptively transferred (**Fig. 6d**). When calculating the relative efficacy per virus-specific CD8 T cell found in the liver to cause liver damage by killing virus-infected hepatocytes (measured by increased serum ALT levels), we found a pronounced increase in this relative cytotoxic activity of MDL-12330A-treated virus-specific CD8 T cells (**Fig. 6e**). Taken together, these results demonstrate a crucial role of adenylyl cyclase in mediating the inhibition of T cell effector function by LSECs *in*

in vitro and *in vivo* in the context of persistent virus infection of the liver. Of note, these experiments with adenylyl cyclase inhibition demonstrated that *de novo* generation of cAMP in CD8 T cells was critical for the tissue-rheostat function to impair effector T cell function.

Legend to Figure 6

a, GzmB expression by CD8 T cells in coculture with activated LSECs for 24h after 1h pre-treatment of T cells with MDL-12330A (100 μ M, MDL), solvent control, or medium and T cells without LSEC contact (n=4, one-way ANOVA with Dunnett's multiple comparison). **b,c**, pPKA and GzmB levels and quantification by CD45.1^{+/+} CD8 T cells activated *in vitro* for 3d followed by 1h pre-treatment with MDL (100 μ M) or mock before transfer into Ad-TTR-GOL infected or Ad-CMV-GOL infected mice for 3d (n=4, one-way ANOVA with Dunnett's multiple comparison). **d**, sALT before and 3d after adoptive transfer of MDL-treated or mock-treated CD45.1^{+/+} *in vitro* activated CD45.1⁺ CTLs into mice with resolved Ad-CMV-GOL or persistent Ad-TTR-GOL infection (n \geq 4, two-way ANOVA with Sidak's multiple comparison). **e**, Increase of sALT per CD45.1^{+/+} CD8 T cell after transfer of MDL-treated or mock-treated CD45.1^{+/+} *in vitro* activated T cells into mice with resolved Ad-CMV-GOL or persistent Ad-TTR-GOL infection (n \geq 4, one-way ANOVA with Tukey's multiple comparison). a-e: one out of \geq two independent experiments shown.

Reviewer 6. As a possible underlying molecular mechanism, the authors suggested that LSEC-derived prostanoids might stimulate *de novo* production of cAMP in T cells and induce the observed phenotype. If this holds true, COX-2 inhibitors or prostanoid antagonists should revert the observed T cell phenotype and, in best case, prevent chronification after infection with HBV. Hence, the authors should be encouraged to use e.g. COX-2 inhibitors in the models described.

Answer: We followed the reviewer's suggestion and performed additional experiments. When we blocked COX-2 activity in LSECs using celecoxib, we observed a significant increase in GzmB expression in T cells in coculture with LSECs (**Fig. 7a,b**). However, high μ M concentrations of celecoxib and continuous presence of the inhibitor in cocultures with CD8 T cells and LSECs were required to increased GzmB levels. If LSECs were pre-treated with celecoxib and then cocultured with CD8 T cells, almost no inhibitory effect on GzmB expression was observed any more (**Fig. 7c**). This suggested that the effect of celecoxib to efficiently block prostanoid production *in vivo* would be very difficult to control and render the interpretation of *in vivo* results also very difficult.

Legend to Figure 7

a,b, GzmB expression and quantification by CD44⁺CD8⁺ T cells treated with celecoxib (1 μM), acetylsalicylic acid (100 μM, ASS) or medium control treatment during coculture with LSECs (n=5, one-way ANOVA with Tukey's multiple comparison). **c**, GzmB expression by CD44⁺CD8⁺ T cells cocultured with activated LSECs pre-treated with celecoxib (1 μM) or medium control for 1h (n=5, unpaired t-test).

Reviewer 7. *In the same sense, have the authors considered other cAMP-inducing mechanisms like adenosine-mediated inhibition of T cell function via receptors A2B and A2B10 or transfer via gap junctions.*

Answer: We followed the reviewer's suggestion and performed numerous additional experiments to provide more insight into how cAMP was increased in T cells. First, we addressed whether signalling through adenosine receptors was involved in the loss of T cell effector function. Although LSECs constitutively expressed the ectonucleotidase CD73 (**Fig. 8a,b**), which is relevant for the breakdown of AMP to adenosine and subsequent adenosine receptor signalling, they did not express CD39 (**Fig. 8a,b**) an ectonucleotidase cleaving extracellular ATP to ADP and AMP⁴. Thus, LSECs cannot stimulate adenosine receptor signalling from extracellular ATP, suggesting that adenosine receptor signalling was not involved in LSEC-induced regulation of T cell function. Indeed, when we blocked adenosine receptor signalling with a specific pharmacological inhibitor, we did not detect an increase in GzmB expression by T cells in coculture with LSECs (**Fig. 8c,d**). Intercellular transfer of cAMP through gap junctions, which was mentioned by the reviewer and has previously been reported to occur between regulator T cells and effector T cells¹⁷, may constitute a further mechanism how increased cAMP levels might be generated. However, when we blocked adenylyl cyclase activity in effector T cells, we detected that these T cells escaped from LSEC-induced inhibition of GzmB expression (*see Fig. 6*). This did not argue for a direct cell-to-cell transfer of cAMP but instead argued for a role of increased adenylyl cyclase activity in T cells. We therefore looked closer at the interaction between LSECs and effector T cells. Surprisingly, we detected LSEC-derived CFSE-labelled molecules on T cells (**Fig 8e,f**), suggesting a transfer of molecules between LSECs and effector T cells. Since LSECs are defined by cAMP-agonism responsible for their differentiation and function⁷, our results point towards a possible transfer of AC molecules from LSECs to T cells that might enhance cAMP signalling. Of note, a similar exchange of molecules has been reported previously for myeloid cells and CD8 T cells in the liver, where transfer of CD14 from myeloid cells to CD8 T cells conferred functional responsiveness of T cells to stimulation by LPS⁶. In principal, it is also possible that molecules activating adenylyl

cyclase are transferred from LSECs to T cells. To this end, we have performed an untargeted approach using mass spectrometry (LC-MS/MS). This revealed a total of 1200 lipids expressed in LSECs and CD8 T cells. Principal component analysis clearly separated these cell populations based on their lipidome profiles (**Fig. 8g**), and direct comparison of LSECs with T cells revealed increased levels of phosphatidyl choline (PC) 19:0/18:1 and phosphatidic acid (PA) 18:1/18:2 in LSECs (**Fig. 8h-j**). Phosphatidyl choline and phosphatidic acid are potential precursors of 18:1 lysophosphatidic acid, a ligand of the G protein-coupled receptor LPA receptor 4 increasing adenylyl cyclase activity¹². Further experiments and a more detailed analysis of these lipidome profiles, however, will be needed to explore in detail the role of adenylyl cyclase-activating pathways. This will be a separate project that in our opinion is beyond the scope of this manuscript.

Since we cannot provide any formal evidence for the involvement of an intercellular molecule exchange and we did not include these data into the revised manuscript.

Legend to Figure 8

a,b, Expression of the ectonucleotidase CD39 and CD73 on LSECs and quantification ($n \geq 4$). **c,d**, GzmB expression by CD8 T cells activated for 3 days *in vitro* followed by 1 h pre-treatment with SCH58261 (100 nM), solvent control or medium control and subsequent coculture with LSECs for 24 h ($n=4$, one-way ANOVA with Dunnett's multiple comparisons) **e,f**, CFSE incorporation and quantification by CD8 T cells in coculture with LSECs labelled with CFSE before coculture ($n=5$, one-way ANOVA with Tukey's multiple comparison). **g**, Principal component analysis of lipidome analysis by LC-MS/MS from naïve T cells,

activated T cells and LSECs (n=3). **h**, Comparison of LSEC and T cell lipidomes generated by LC-MS-MS from primary mouse cell populations (n=3). **i,j**, Phosphatidyl choline (C19:0/C18:1) and Phosphatidic acid (C18:1/18:2) detected in naïve T cells, effector T cells and LSECs by LC-MS/MS (n=3)a-f: one out \geq of two independent experiments shown.

Minor criticism:

8. In Fig. 1l and Extended Data Fig. 2i the authors suggest involvement of Tox. Are T cells from Tox-deficient mice refractory to the described mechanism?

Answer: We followed the reviewer's suggestion and examined the role of Tox. To this end, we used Tox-deficient CD8 T cells from *Mx^{cre};Rosa26-STOP-eYFP* P14 TCR-transgenic mice¹⁸. We treated Tox-KO or wildtype littermate CD8 T cells with forskolin and characterised their response to cognate stimulation. Tox-KO T cells were equally sensitive as Tox-competent T cells to forskolin-induced inhibition of T cell effector function (**Fig. 9**). Thus, the absence of Tox did not rescue T cells from the inhibitory effects of cAMP signalling.

Legend to Figure 9

Frequencies of cytokine (IFN γ and TNF)-expressing activated Tox-KO or Tox-WT CD8 T cells after 4 h treatment with Fsk (50 μ M) or solvent control followed by 15 h stimulation with peptide (n=8, two-way ANOVA with Tukey's multiple comparison).

Reviewer: Clarity and context

The text is written in a clear and accessible way. Some typos are still present.

Answer: We have very carefully revised the manuscript and have eliminated all typos.

Reviewer: References

Answer: We have included the essential publications in the revised manuscript.

References cited in the point-by-point reply

- 1 Guidotti, L. G. *et al.* Immunosurveillance of the liver by intravascular effector CD8(+) T cells. *Cell* **161**, 486-500 (2015). <https://doi.org:10.1016/j.cell.2015.03.005>
- 2 Benechet, A. P. *et al.* Dynamics and genomic landscape of CD8(+) T cells undergoing hepatic priming. *Nature* **574**, 200-205 (2019). <https://doi.org:10.1038/s41586-019-1620-6>
- 3 Sorrentino, C. *et al.* Adenosine A2A Receptor Stimulation Inhibits TCR-Induced Notch1 Activation in CD8+T-Cells. *Front Immunol* **10**, 162 (2019). <https://doi.org:10.3389/fimmu.2019.00162>
- 4 Regateiro, F. S., Cobbold, S. P. & Waldmann, H. CD73 and adenosine generation in the creation of regulatory microenvironments. *Clin Exp Immunol* **171**, 1-7 (2013). <https://doi.org:10.1111/j.1365-2249.2012.04623.x>
- 5 Brudvik, K. W. & Taskén, K. Modulation of T cell immune functions by the prostaglandin E(2) - cAMP pathway in chronic inflammatory states. *Br J Pharmacol* **166**, 411-419 (2012). <https://doi.org:10.1111/j.1476-5381.2011.01800.x>
- 6 Pallett, L. J. *et al.* Tissue CD14(+)CD8(+) T cells reprogrammed by myeloid cells and modulated by LPS. *Nature* **614**, 334-342 (2023). <https://doi.org:10.1038/s41586-022-05645-6>
- 7 Gage, B. K. *et al.* Generation of Functional Liver Sinusoidal Endothelial Cells from Human Pluripotent Stem-Cell-Derived Venous Angioblasts. *Cell Stem Cell* **27**, 254-269.e259 (2020). <https://doi.org:10.1016/j.stem.2020.06.007>
- 8 Wehbi, V. L. & Taskén, K. Molecular Mechanisms for cAMP-Mediated Immunoregulation in T cells - Role of Anchored Protein Kinase A Signaling Units. *Front Immunol* **7**, 222 (2016). <https://doi.org:10.3389/fimmu.2016.00222>
- 9 Wieland, S., Thimme, R., Purcell, R. H. & Chisari, F. V. Genomic analysis of the host response to hepatitis B virus infection. *Proc Natl Acad Sci U S A* **101**, 6669-6674 (2004). <https://doi.org:10.1073/pnas.0401771101>
- 10 Mutz, P. *et al.* HBV Bypasses the Innate Immune Response and Does Not Protect HCV From Antiviral Activity of Interferon. *Gastroenterology* **154**, 1791-1804 e1722 (2018). <https://doi.org:10.1053/j.gastro.2018.01.044>

- 11 Wilson, E. B. *et al.* Blockade of chronic type I interferon signaling to control persistent LCMV infection. *Science* **340**, 202-207 (2013).
<https://doi.org:10.1126/science.1235208>
- 12 Riaz, A., Huang, Y. & Johansson, S. G-Protein-Coupled Lysophosphatidic Acid Receptors and Their Regulation of AKT Signaling. *Int J Mol Sci* **17**, 215 (2016).
<https://doi.org:10.3390/ijms17020215>
- 13 Unutmaz, D. *et al.* The primate lentiviral receptor Bonzo/STRL33 is coordinately regulated with CCR5 and its expression pattern is conserved between human and mouse. *J Immunol* **165**, 3284-3292 (2000).
<https://doi.org:10.4049/jimmunol.165.6.3284>
- 14 Fischer, D. S., Schaar, A. C. & Theis, F. J. Modeling intercellular communication in tissues using spatial graphs of cells. *Nature biotechnology* **41**, 332-336 (2023).
<https://doi.org:10.1038/s41587-022-01467-z>
- 15 Nguyen, S. *et al.* Elite control of HIV is associated with distinct functional and transcriptional signatures in lymphoid tissue CD8(+) T cells. *Science translational medicine* **11** (2019). <https://doi.org:10.1126/scitranslmed.aax4077>
- 16 Zhao, J. *et al.* Single-cell RNA sequencing reveals the heterogeneity of liver-resident immune cells in human. *Cell Discov* **6**, 22 (2020). <https://doi.org:10.1038/s41421-020-0157-z>
- 17 Bopp, T. *et al.* Cyclic adenosine monophosphate is a key component of regulatory T cell-mediated suppression. *J Exp Med* **204**, 1303-1310 (2007).
<https://doi.org:jem.20062129> [pii]10.1084/jem.20062129
- 18 Alfei, F. *et al.* TOX reinforces the phenotype and longevity of exhausted T cells in chronic viral infection. *Nature* (2019). <https://doi.org:10.1038/s41586-019-1326-9>

Reviewer Reports on the First Revision:

Referees' comments:

Referee #1 (Remarks to the Author):

The authors try to answer my comments concerning the limited quantity of human data by providing the phenotype of T cells in the liver of 3 CHB patients in comparison to 3 uninfected livers. They also isolated HBV-specific CD8 T cells from 3 additional patients with CHB and provide data that confirmed the presence of a cAMP/CREM signature in these virus-specific T cells from CHB patients.

The new results are again technically well executed and overall the paper provides a very convincing story about how virus-specific T cell activity can be regulated in a liver with persistent viral antigens presentation, but overall, my skepticism that this manuscript provides relevant information concerning natural "chronic hepatitis B" remains.

I would say, that overall the manuscript does not show any data about intrahepatic and peripheral HBV-specific T cells in the context of "hepatitis" caused by HBV.

The intrahepatic human data are practically absent. The authors show only that in the liver of 3 patients with CHB there are more CXCR6+ T cells than healthy controls. The main point of this paper is that dysfunctional "CREM+" T cells accumulated in the liver. I would also say that the control should be a liver with pathology different than HBV and not only a healthy livers.

The data of dysfunctional HBV-specific T cells are specific only for core (and there are now extended evidence that CD8 T cells specific for different HBV antigens have different dysfunctional profile (i.e. Schuch, A. et al. Phenotypic and functional differences of HBV core-specific versus HBV polymerase-specific CD8+ T cells in chronically HBV-infected patients with low viral load. Gut gutjnl-2018-316641 (2019)) and these HBV core specific CD8 T cells were studied in patients with no hepatitis and low HBV-DNA. These patients represent a selected patient population called HBeAg neg chronic HBV infection, or in the past "inactive carriers". In other words, the human data provided remains limited.

Referee #2 (Remarks to the Author):

UNAVAILABLE FOR RE-REVIEW.

Referee #3 (Remarks to the Author):

Title: "A PKA-associated liver-tissue rheostat curbs T cell receptor signalling and effector function of virus-specific CD8 T cells in chronic viral hepatitis ", by Percy Knolle and colleagues.

In the revised manuscript by Percy Knolle and colleagues, the authors have responded to all criticisms by including additional data that further support the main hypothesis of the present manuscript.

For example, the authors have included additional analyses of CD8+ T cells from HBV-infected patients and show that the observed cAMP/CREM signature is also detectable in CD8+ T cells isolated from these patients. To demonstrate the specificity of this mechanism, the authors analyzed publicly available datasets of HIV-specific CD8+ T cells. Here, the authors demonstrated that the observed cAMP/CREM signature is not present in HIV-specific CD8+ T cells, highlighting the specificity of the mechanism for the proposed liver tissue rheostat.

Mechanistically, the authors provided additional evidence that LSEC-produced PGE2 effects, but not adenosine-mediated effects, contribute to the observed phenotype in HBcore-specific CD8+ T cells. Disruption of this pathway by a COX2 inhibitor led to increased GzmB expression in T cells, which partially reversed the observed phenotype. Most importantly, the authors now demonstrate in a series of additional experiments that the observed phenotype results from PKA activity and not EPAC-mediated signal transduction, underscoring a true PKA-associated liver-tissue rheostat. Overall, the manuscript is greatly improved and I would like to recommend it for publication in Nature.

Author Rebuttals to First Revision:

Point-by-point reply

Reviewer 1: The authors try to answer my comments concerning the limited quantity of human data by providing the phenotype of T cells in the liver of 3 CHB patients in comparison to 3 uninfected livers. They also isolated HBV-specific CD8 T cells from 3 additional patients with CHB and provide data that confirmed the presence of a cAMP/CREM signature in these virus-specific T cells from CHB patients.

The new results are again technically well executed and overall the paper provides a very convincing story about how virus-specific T cell activity can be regulated in a liver with persistent viral antigens presentation, but overall, my skepticism that this manuscript provides relevant information concerning natural “chronic hepatitis B” remains.

I would say, that overall the manuscript does not show any data about intrahepatic and peripheral HBV-specific T cells in the context of “hepatitis” caused by HBV.

The intrahepatic human data are practically absent. The authors show only that in the liver of 3 patients with CHB there are more CXCR6+ T cells than healthy controls. The main point of this paper is that dysfunctional “CREM+” T cells accumulated in the liver. I would also say that the control should be a liver with pathology different than HBV and not only healthy livers.

The data of dysfunctional HBV-specific T cells are specific only for core (and there are now extended evidence that CD8 T cells specific for different HBV antigens have different dysfunctional profile (i.e. Schuch, A. et al. Phenotypic and functional differences of HBV core-specific versus HBV polymerase-specific CD8+ T cells in chronically HBV-infected patients with low viral load. Gut 2019;69:3166-3174 (2019)) and these HBV core specific CD8 T cells were studied in patients with no hepatitis and low HBV-DNA. These patients represent a selected patient population called HBeAg neg chronic HBV infection, or in the past “inactive carriers”.

In other words, the human data provided remains limited.

Answer:

We agree with the reviewer that it is not possible to formally address by immunohistochemistry of liver tissue from patients with chronic hepatitis B the question of whether these CXCR6+ T cells are HBV-specific. Therefore, we have down-tuned the statements on these findings in the revised manuscript. Control liver tissues are from the tumour-free area of liver biopsies of patients with liver cancer.

To follow the reviewer’s main comment on the lack of data from intrahepatic HBV-specific CD8 T cells, we have started a collaboration with the team of Prof. Georg Lauer (Harvard University), Prof. Adam Gehring (University of Toronto) and Prof. Peter Jansen (University of Rotterdam) who recently conducted a clinical study to evaluate intrahepatic HBV-specific CD8 T cells. Fine needle aspirates were obtained from 21 patients with different phases of chronic HBV

infection, i.e., chronic hepatitis B, HBeAg^{neg}/anti-HBe⁺ patients with chronic HBV infection, and patients with functional cure from chronic HBV infection. From these fine needle aspirates, 977 intrahepatic virus-specific CD8 T cells were isolated and subjected to scRNAseq and analysis for the presence of a CREM transcriptional signature.

In the new Figure 3e of the revised manuscript, we now provide evidence for the presence of intrahepatic HBcore-specific CXCR6⁺ CD8 T cells that have a CREM signature in contrast to CXCR6^{neg} CD8 T cells that lack a CREM signature. This demonstrates that HBV-specific CD8 T cells that express CXCR6 and have a CREM signature exist in the liver of patients with chronic HBV infection and that they are present during hepatitis. In HBeAg^{neg}/anti-HBe⁺ patients with chronic HBV infection, who can be considered inactive carriers – as the reviewer has pointed out – we also detected a CREM signature in intrahepatic HBcore-specific CXCR6⁺ CD8 T cells but not in CXCR6^{neg} CD8 T cells (new Figure 3e in the revised manuscript). Importantly, in patients with a functional cure, i.e., with immune control of HBV infection who have a very low viral antigen load in the liver, we did not detect a CREM signature anymore in the intrahepatic HBV-specific CD8 T cells (new Figure 3e in the revised manuscript). Thus, the new human HBV data show higher CREM activity in HBV-specific CD8 T cells in patients with active hepatitis and less CREM activity with increasing viral control and absence of liver damage.

We further added scRNAseq data from HBcore-specific CD8 T cells from a second cohort of four HLA-A2⁺ patients with chronic HBV infection. The results from this analysis further confirm the presence of a CREM signature in HBcore-specific CD8 T cells in peripheral blood. These results are now shown in the new Extended Data Figure 4c-e. Together, we now have a total of 10 patients with chronic HBV infection, in whom we detect HBV-specific CD8 T cells with a CREM signature in the circulation.

Together, these data demonstrate that the CREM signature found in virus-specific CD8 T cells during persistent infection with hepatotropic viruses and, in particular, HBV infection in preclinical disease models is pronouncedly present in HBV-specific CD8 T cells in the liver and in the circulation of chronic hepatitis B patients.

Reviewer 3: In the revised manuscript by Percy Knolle and colleagues, the authors have responded to all criticisms by including additional data that further support the main hypothesis of the present manuscript.

For example, the authors have included additional analyses of CD8+ T cells from HBV-infected patients and show that the observed cAMP/CREM signature is also detectable in CD8+ T cells isolated from these patients. To demonstrate the specificity of this mechanism, the authors analyzed publicly available datasets of HIV-specific CD8+ T cells. Here, the authors demonstrated that the observed cAMP/CREM signature is not present in HIV-specific CD8+ T cells, highlighting the specificity of the mechanism for the proposed liver tissue rheostat.

Mechanistically, the authors provided additional evidence that LSEC-produced PGE2 effects, but not adenosine-mediated effects, contribute to the observed phenotype in HBcore-specific CD8+ T cells. Disruption of this pathway by a COX2 inhibitor led to increased GzmB expression in T cells, which partially reversed the observed phenotype. Most importantly, the authors now demonstrate in a series of additional experiments that the observed phenotype results from PKA activity and not EPAC-mediated signal transduction, underscoring a true PKA-associated liver-tissue rheostat. Overall, the manuscript is greatly improved and I would like to recommend it for publication in Nature.

Answer: We thank this reviewer for his comments and very positive evaluation of our revised manuscript!

Reviewer Reports on the Second Revision:

Referees' comments:

Referee #1 (Remarks to the Author):

In the revised manuscript, the authors have responded to the criticisms related to the scarcity of HBV-specific CD8 T cells data during natural HBV infection. Importantly they now show scRNA seq data of $n \approx 1000$ HBV-specific CD8 T cells isolated from liver of CHB patients (Figure 3e). As the authors wrote, these data provided “evidence for the presence of intrahepatic HBcore-specific CXCR6+ CD8 T cells that have a CREM signature in contrast to CXCR6neg CD8 T cells that lack a CREM signature “. I think the addition of these novel data is important to support the main conclusion of the paper: the presence of a CREM signature in virus-specific CD8 T cells during persistent infection in an animal model can also be detected in the setting of natural HBV infection. However to clarify the importance of these novel data in the context of CHB infection, I have a request that I think the authors could easily address since they should already be in possession of the data from the siRNA seq. Figure 3e displays a difference in CREM expression in CXCR6pos versus CXCR6neg intrahepatic HBcore-specific CD8 T cells. There is, however, no indication of the proportion of these two phenotypes in the analyzed total HBcore-specific CD8 T cells, unlike in the peripheral blood where the authors reported approximately 25% of HBcore-specific CD8 T cells having high Crem activity (Extended data fig4). The addition of the proportion of CXCR6+ CD8 T cells within HBV-specific CD8 T cells can provide important information to understand the impact of CREM signature on the intrahepatic HBV-specific CD8 T cells response. This information should be also linked with figure 3d that show the presence of CXCR-6+ T cells in the liver. At the moment, to justify this figure the authors wrote: “ To address the question of whether *HBV-specific CD8 T cells in the liver of chronic hepatitis B patients* show a CREM signature, we investigated tissue from liver biopsies by immunohistochemistry. We found an enrichment of CXCR6+CD3+ T cells in the livers of chronic hepatitis B patients (5-15%, mean 5%) when compared to uninfected controls (<1%) (Fig. 3d), but *this approach did not allow us to identify HBV-specific CD8 T cells*.”

Perhaps this figure 3d should now be used in terms of the frequency of CXCR6+ T cells within the CHB liver, and data should be linked with the scRNA seq data of intrahepatic HBcore-specific CD8 T cells to better clarify the impact of CREM+ virus-specific CD8 T cells in the HBV infected liver. If the proportion of CXCR-6+ cells within HBV-specific CD8 T cells is low, I think the authors should tune down the conclusion. In the abstract, the authors wrote that “ Circulating and intrahepatic virus-specific CD8 T cells in CHB similarly showed enhanced CREM expression and activity”. Such a sentence seems to imply that “ all (or the majority) of HBV-specific CD8 T cells present in CHB patients have such signature. If this is occurring only in a small proportion, such info should be added to the abstract.

Referee #4 (Remarks to the Author):

The authors herein utilize a persistent murine model of hepatotropic targeting infection using

Adenovirus delivery in addition to chronically HBV-infected human subjects to propose a novel explanation for the signaling cues potentiating CD8⁺ T cell dysfunction during persistent antigen presence in the liver. The authors have performed significant new analyses to address previous reviewers' concerns. Examination of chronic LCMV infection in light of their findings provides a useful comparison detailing the dichotomy by which CD8⁺ T cell dysfunction does not come as a "one size fits all" model, regarding both its origins of TCR-dependence and its potential for reinvigoration.

Reading this for the first time, there are just a couple of points that I would like to see addressed in a final version.

The authors indeed do a tremendous job of deciphering both the direct culprits inducing this CREM signature, LSECs, as well as its mechanistic drivers, via PGE production and its ensuing COX-2 sensitivity, and investigating whether these CXCR6⁺ CTLs are also present in human subjects infected with HBV. Although, it is surprising that these HBV-specific T cells maintain their CREM signature circulating in the periphery where antigen encounter with LSECs are removed.

While the authors make a compelling case for the CREM signatures of CD8⁺ T cells associated with persistent HBV antigen presence, this extension as a generalized phenomenon applicable to "hepatitis" is not substantiated by its (a) lack of appreciable hepatitis-related sequelae observed in the murine model and (b) its omission of investigating other human viruses associated with hepatitis. Furthermore, the absence of ALT elevation in addition to the lack of histological hallmarks, serve to support the notion that the authors should perhaps remove the term "hepatitis" from the title.

As cited in the text (PMID 29729204) for the basis for Adenovirus-associated HBV infection, the "resolving" model would be well supported by showing its peak HBV copies early in acute infection in extended data 3D, rather than in the post-clearance phase at day 45. Furthermore, while robust replication of the "resolving" model was not clearly depicted, it is not surprising that the CD8⁺ T cell signatures associated with what was possibly a transient transduction were starkly contrasted with a persistent antigen expression. That being said, important pathways were gleaned from interrogating T cells with increased interaction with LSECs in this report.

The immunological findings of this work are nonetheless novel, relevant, and should be of general interest to a wide audience. The LSEC-derived signals inducing the CREM signature of dysfunctional CD8⁺ T cells, its mechanistic explanation, and its translation upheld in human subjects solidify the reversible nature of this phenomenon as an invaluable contribution to understanding organ-specific T cell dysfunction in chronic infection as potentially variable events across a broad spectrum of tropisms. I recommend the authors to soften the current claims from being relayed as applicable to all chronic infections inducing hepatitis, especially that the Adenovirus delivery model being used here is simply providing long-term presence of antigen as opposed to a bona fide HBV viral infection and replication in their mouse model. Once depicted as such, this work would convey an important message worthy of publication in Nature.

Referee #1 (Remarks to the Author):

In the revised manuscript, the authors have responded to the criticisms related to the scarcity of HBV-specific CD8 T cells data during natural HBV infection. Importantly they now show scRNA seq data of $n = \sim 1000$ HBV-specific CD8 T cells isolated from liver of CHB patients (Figure 3e). As the authors wrote, these data provided “evidence for the presence of intrahepatic HBcore-specific CXCR6+ CD8 T cells that have a CREM signature in contrast to CXCR6neg CD8 T cells that lack a CREM signature “. I think the addition of these novel data is important to support the main conclusion of the paper: the presence of a CREM signature in virus-specific CD8 T cells during persistent infection in an animal model can also be detected in the setting of natural HBV infection.

Answer: We thank the referee for this positive evaluation of the additional data on intrahepatic HBV-specific CD8 T cells we provided in the revised manuscript.

However to clarify the importance of these novel data in the context of CHB infection, I have a request that I think the authors could easily address since they should already be in possession of the data from the siRNA seq. Figure 3e displays a difference in CREM expression in CXCR6pos versus CXCR6neg intrahepatic HBcore-specific CD8 T cells. There is, however, no indication of the proportion of these two phenotypes in the analyzed total HBcore-specific CD8 T cells, unlike in the peripheral blood where the authors reported approximately 25% of HBcore-specific CD8 T cells having high Crem activity (Extended data fig4). The addition of the proportion of CXCR6+ CD8 T cells within HBV-specific CD8 T cells can provide important information to understand the impact of CREM signature on the intrahepatic HBV-specific CD8 T cells response.

This information should be also linked with figure 3d that show the presence of CXCR-6+ T cells in the liver. At the moment, to justify this figure the authors wrote: “ To address the question of whether *HBV-specific CD8 T cells in the liver of chronic hepatitis B patients* show a CREM signature, we investigated tissue from liver biopsies by immunohistochemistry. We found an enrichment of CXCR6+CD3+ T cells in the livers of chronic hepatitis B patients (5-15%, mean 5%) when compared to uninfected controls (<1%) (Fig. 3d), but *this approach did not allow us to identify HBV-specific CD8 T cells*.”

Perhaps this figure 3d should now be used in terms of the frequency of CXCR6+ T cells within the CHB liver, and data should be linked with the scRNA seq data of intrahepatic HBcore-specific CD8 T cells to better clarify the impact of CREM+ virus-specific CD8 T cells in the HBV infected liver. If the proportion of CXCR-6+ cells within HBV-specific CD8 T cells is low, I think the authors should tune down the conclusion. In the abstract, the authors wrote that “ Circulating and intrahepatic virus-

specific CD8 T cells in CHB similarly showed enhanced CREM expression and activity". Such a sentence seems to imply that " all (or the majority) of HBV-specific CD8 T cells present in CHB patients have such signature. If this is occurring only in a small proportion, such info should be added to the abstract.

Answer: As the referee suggested to bring the data on the CXCR6⁺ T cells detected by immunohistochemistry into perspective with the scRNAseq data, we performed additional experiments. We now include a further 5 liver biopsies from patients with chronic hepatitis B, where we find CXCR6⁺ T cells. The results demonstrate that CXCR6⁺ T cells were detected with a frequency between 2.7 and 15.4 % in patients (n=16) with chronic hepatitis B. As we pointed out, we cannot provide any information on the antigen-specificity of CXCR6⁺ T cells detected by immunohistochemistry and carefully reworded our statements as suggested by the reviewer.

The text in the results section now reads as follows:

"We found CXCR6⁺CD3⁺ T cells in the livers of chronic hepatitis B patients (n=11), with a frequency of 2.7% to 15.4% of all T cells, whereas no CXCR6⁺CD3⁺ T cells were detected in livers from patients without HBV infection (n=5) (Fig. 3d). However, immunohistochemistry did not allow us to detect whether HBV-specific CD8 T cells were among the CXCR6⁺ T cells."

As the referee suggested, we added the information on the proportion of the hepatic HBV-specific CXCR6⁺ CD8 T cells among all HBV-specific CD8 T cells to the text in the results section and in the figure. The frequency of CXCR6⁺ CD8 T cells among all HBV-specific CD8 T cells in the liver varies between 11.9 and 22.7% in patients with chronic hepatitis, chronic HBV infection and in patients with functional cure. The results section has been modified to follow the request to link the detection of CXCR6⁺ T cells by immunohistochemistry to the results from scRNAseq analysis.

The text now reads as follows:

"Frequencies of CXCR6 expressing CD8 T cells were in the range of 11.9 to 22.7 % of hepatic HBV-specific CD8 T cells (Fig. 3e), which is in line with the low proportions of CXCR6⁺ T cells detected by immunohistochemistry."

As suggested by the reviewer, we included the information on the proportion of CXCR6⁺ HBV-specific CD8 T cells in the abstract to clarify that not all HBV-specific CD8 T cells in the liver express CXCR6 and have a CREM signature.

The text in the revised abstract reads as follows:

“In chronic hepatitis B patients, circulating and intrahepatic HBV-specific CXCR6⁺ CD8 T cells with enhanced CREM expression and activity were detected at a frequency of 12-22% of HBV-specific CD8 T cells.”

The proportions of CXCR6⁺ T cells detected by immunohistochemistry are lower than the proportions of CXCR6⁺ HBV-specific CD8 T cells isolated with the help of multimer staining from fine needle aspirates from the livers of chronic hepatitis B patients, which is expected since it is unlikely that most T cells in the liver of patients with chronic hepatitis B are HBV-specific. Together, scRNAseq and immunohistochemistry both demonstrate the presence of a sizeable population of HBV-specific CD8 T cells that express CXCR6.

Referee #4 (Remarks to the Author):

The authors herein utilize a persistent murine model of hepatotropic targeting infection using Adenovirus delivery in addition to chronically HBV-infected human subjects to propose a novel explanation for the signaling cues potentiating CD8+ T cell dysfunction during persistent antigen presence in the liver. The authors have performed significant new analyses to address previous reviewers' concerns. Examination of chronic LCMV infection in light of their findings provides a useful comparison detailing the dichotomy by which CD8+ T cell dysfunction does not come as a "one size fits all" model, regarding both its origins of TCR-dependence and its potential for reinvigoration.

Reading this for the first time, there are just a couple of points that I would like to see addressed in a final version.

The authors indeed do a tremendous job of deciphering both the direct culprits inducing this CREM signature, LSECs, as well as its mechanistic drivers, via PGE production and its ensuing COX-2 sensitivity, and investigating whether these CXCR6+ CTLs are also present in human subjects infected with HBV. Although, it is surprising that these HBV-specific T cells maintain their CREM signature circulating in the periphery where antigen encounter with LSECs are removed.

Answer: We thank the reviewer for this positive evaluation of our manuscript.

While the authors make a compelling case for the CREM signatures of CD8+ T cells associated with persistent HBV antigen presence, this extension as a generalized phenomenon applicable to "hepatitis" is not substantiated by its (a) lack of appreciable hepatitis-related sequelae observed in the murine model and (b) its omission of investigating other human viruses associated with hepatitis. Furthermore, the absence of ALT elevation in addition to the lack of histological hallmarks, serve to support the notion that the authors should perhaps remove the term "hepatitis" from the title.

Answer: We have followed the reviewer's suggestion and removed the word "hepatitis" from the title and replaced it with "chronic hepatitis B virus infection". Throughout the manuscript, we make it clear that the findings reported here apply to HBV infection and cannot be generalised for all viral hepatitis infections.

As cited in the text (PMID 29729204) for the basis for Adenovirus-associated HBV infection, the "resolving" model would be well supported by showing its peak HBV copies early in acute infection in

extended data 3D, rather than in the post-clearance phase at day 45. Furthermore, while robust replication of the “resolving” model was not clearly depicted, it is not surprising that the CD8+ T cell signatures associated with what was possibly a transient transduction were starkly contrasted with a persistent antigen expression. That being said, important pathways were gleaned from interrogating T cells with increased interaction with LSECs in this report.

Answer: We followed the referee’s suggestion and performed additional experiments to evaluate HBV copy numbers at an earlier time point after Ad-HBV transduction, i.e. d8 p.i. The results are shown in the revised Extended Data Figure 3d and clearly demonstrate a drop in HBV copies by 21-fold comparing d8 to d45 ($p > 0.0001$) after transduction of hepatocytes with Ad-HBV when the immune response controls the infection. In contrast, there is only a modest reduction by 4-fold from d8 to d45 ($p = 0.0021$) after transduction with 1×10^8 i.u. of Ad-HBV. We have included this information in the results section.

We also performed a Southern blot analysis to demonstrate HBV replication after Ad-HBV transduction of mouse livers. While HBV replication is clearly detectable at the dose of 1×10^8 i.u. of Ad-HBV, this cannot be detected at the lower dose due to the low sensitivity of the Southern blot. We include the Southern blot analysis of the mouse livers for the reviewer's information below.

The text in the revised results section now reads as follows:

“We, therefore, established a preclinical in vivo model where hepatocytes were cleared by virus-specific immunity after transduction with 10^7 IU Ad-HBV, resulting in a >20-fold reduction in HBV copies to almost undetectable levels in the liver from d8 to d45 after transduction. After transduction with 10^8 IU Ad-HBV, continuous HBV gene expression in hepatocytes developed, revealed by high serum HBeAg levels, a 4-fold reduction in HBV copies and persistence of HBeAg^{pos} hepatocytes in liver tissue (Fig. 2a, Extended Data Fig. 3d-f).”

The immunological findings of this work are nonetheless novel, relevant, and should be of general interest to a wide audience. The LSEC-derived signals inducing the CREM signature of dysfunctional CD8+ T cells, its mechanistic explanation, and its translation upheld in human subjects solidify the reversible nature of this phenomenon as an invaluable contribution to understanding organ-specific T cell dysfunction in chronic infection as potentially variable events across a broad spectrum of tropisms. I recommend the authors to soften the current claims from being relayed as applicable to all chronic infections inducing hepatitis, especially that the Adenovirus delivery model being used here is simply providing long-term presence of antigen as opposed to a bona fide HBV viral infection and replication in their mouse model. Once depicted as such, this work would convey an important message worthy of publication in Nature.

Answer: We have followed the referee's suggestion and have carefully reworded our statements throughout the manuscript. We now make it very clear that the Ad-HBV model is a means to deliver HBV genomes that express viral antigens under the control of the endogenous viral promoters into hepatocytes and does not entirely reflect the situation of HBV infection in humans, where HBV can infect hepatocytes, replicate and spread. We have, therefore, replaced the term "infection" with "transduction" and "replication" with "viral gene expression" throughout the manuscript. This terminology now reflects the limitations of currently available *in vivo* model systems in mice to study the T cell response against HBV and complies with the request of the referee to soften the claims of our findings.

Reviewer Reports on the Third Revision:

Referees' comments:

Referee #1 (Remarks to the Author):

Thanks for accepting the last suggestions. Congratulations for the important work.

Referee #4 (Remarks to the Author):

In this revised version, the authors have significantly strengthened their statements and markedly improved their interpretation of their data. The effort put forward by the authors to significantly revise this timely data and important topic related to the understanding of the CREM signatures of CD8+ T cells associated with persistent HBV antigen presence is commendable.